# KNAPSACK SCHEMA LINKING AGENT FOR LLM-BASED TEXT-TO-SQL GENERATION

## ABSTRACT

Generating SQLs according to user queries (text-to-SQL) is a long-standing sequential challenge, where the accuracy of the initial schema linking significantly impacts the subsequent SQL generation performance. However, existing models often focus more on SQL generation and less on the schema linking task, leading to potential missing or redundant schema linking and suboptimal SQL generation performance. The underlying reason is that schema linking is not a simple selection problem but a **Knapsack problem**, which should consider both the *value* of the schema linking in terms of missing important information and the *weight* of the schema linking in terms of providing redundant information. Motivated by this, we provide two tailored SL benchmarks and two tailored metrics to train SL agents and to evaluate the missing and redundant schema linking. In this paper, we propose the **Knapsack Schema Linking Agent (KaSLA)**, which can link the most valuable and least redundant schema element subsets for both tables and columns. KaSLA introduces an importance score function to predict each schema element's importance score, and then utilizes the importance score to estimate the value and the weight of each schema. Then, by estimating the capacity, the maximum weight the knapsack can hold, of a given user query from historical SQL records, KaSLA employs efficient dynamic programming to select the most valuable schema element set within the estimated capacity. Extensive experiments on two benchmark datasets demonstrate the superior performance of KaSLA over 12 state-of-the-art baselines. Especially on the popular and challenging BIRD benchmark, KaSLA can outperform the baselines by over 5.72%.

## 1 INTRODUCTION

With the advent of large language models (LLMs) (Achiam et al., 2023; Dubey et al., 2024), LLM-based text-to-SQL is emerging as the next-generation interface for database users (Hong et al., 2024b). Typically, text-to-SQL frameworks employ a two-step process: first linking user queries to database schema, then generating the corresponding SQL statement. However, the accurate SQL statement requires the linking to the correct schema elements (tables and columns) meaning that fatal errors in earlier stages inevitably propagate to later ones. For instance, inaccurate schema linking—such as overlooking crucial tables or linking to irrelevant tables or columns—will lead to incorrect SQL statements. As illustrated in Figure 1, current text-to-SQL models still exhibit

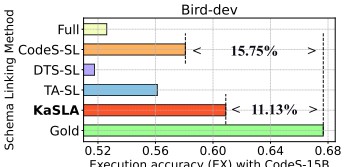

Figure 1: The performance comparison of the schema linking methods with a same text-to-SQL backbone on BIRD-dev.

a significant performance gap (15.75%) in SQL generation compared with feeding gold standard schema linking. Consequently, improving the accuracy of schema linking remains a critical challenge with substantial research value, offering considerable potential for advancement in the field.

Recent state-of-the-art text-to-SQL models tend to focus primarily on final SQL generation while employing relatively simplistic schema linking strategies. DIN-SQL (Pourreza & Rafiei, 2023) pioneered the use of LLMs to generate schema linking by inputting the full schema. DAIL-SQL (Gao et al., 2024) extended this approach by leveraging historical query-SQL pairs as evidence to improve generative schema linking. DELLM (Hong et al., 2024a) introduced a specialized data expert LLM

Figure 2: Comparison of previous selection schema linking framework and knapsack schema linking framework. We use table linking as an example. Selection models usually lead to either missing or redundant items, while knapsack schema linking solves this issue by maximizing the total value of objects while adhering to the total weight constraint.

to provide additional knowledge for schema linking. DTS-SQL (Pourreza & Rafiei, 2024) fine-tunes two LLMs for table linking and SQL generation, respectively. TA-SQL (Qu et al., 2024) generates a dummy SQL and then utilizes LLM to abstract the linked schema from it for the subsequent SQL generation. In contrast to the above approach, which relies exclusively on the generative capabilities of LLMs, CodeS (Li et al., 2024b) implemented a recall-based strategy to include semantically matching schema elements in the input for SQL generation models, albeit at the cost of potentially introducing un-matching elements. However, as illustrated in Figure 1, both generative and recalled schema linking strategies have a significant gap against the gold schema linking.

The significant performance gap can be attributed to three key problems:

- **Oversimplified Selection Modeling**: Existing text-to-SQL models do not impose constraints on the schema linking process. As shown in Figure 3, current schema linking processes suffer from either high missing rates, high redundancy rates, or both. Such missing and redundant elements negatively impact the overall SQL generation performance.

- **Missing & Redundancy Seesaw Problem**: There exists a seesaw phenomenon between missing and redundant elements in schema linking. Generative strategies tend to have high missing rates and low redundancy rates, while recall-based models exhibit the opposite behavior. Reducing both missing and redundant schema elements simultaneously poses a significant challenge.

- **Lack of Benchmarks and Metrics**: The field currently lacks a formal definition of the schema linking problem and standardized evaluation metrics for assessing missing and redundant elements. This absence hinders consistent evaluation and comparison of different approaches, impeding progress in the field.

From these observations, we highlight that schema linking is not a simple selection problem but a Knapsack problem, which requires maximizing total value (low missing rate) while satisfying total weight constraints (low redundant rate). To address this, we design two tailored schema linking benchmarks and introduce two schema linking metrics: **schema missing rate** and **schema redundancy rate** to assess schema linking performance. Based on these benchmarks, we propose the **Knapsack Schema Linking Agent (KaSLA)**, a dynamic programming agent that links the most valuable and least redundant element sets according to user queries. KaSLA can be applied to both table and column linking. It introduces a novel nomination-guaranteed score function to predict the importance score of each element, which simultaneously assigns high scores to highly confident elements while ensuring all elements receive basic importance scores to prevent miss-

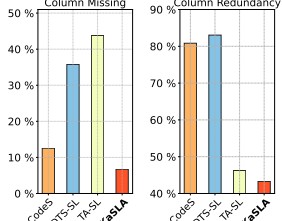

Figure 3: The comparison of schema linking models on missing and redundancy on BIRD-dev.

ing. These importance scores are then used to estimate the value and weight of each element. By estimating the capacity of a given user query from historical SQL records, KaSLA employs efficient dynamic programming to select the most valuable element set within the predicted capacity. In summary, our contributions are as follows:

- We formally define the schema linking problem as a Knapsack problem. Motivated by this formulation, we propose the Knapsack Schema Linking Agent (KaSLA) to dynamically select the least missing and redundant schema elements and enhance mainstream text-to-SQL models.

- We introduce two well-organized benchmarks for schema linking training and evaluation, along with two metrics. These contributions facilitate future research in schema linking and text-to-SQL tasks. These benchmarks are anonymously published [1].

- KaSLA offers a novel nomination-guaranteed score function to overcome the missing & redundancy seesaw problems, and then estimate the value, weight, and capacity. This approach provides a fundamental solution for improving schema linking and text-to-SQL performance. Codes are public access available [2].

- Extensive experiments demonstrate that KaSLA can enhance schema linking accuracy by reducing missing and redundant information, as well as improve the final text-to-SQL generation accuracy. Particularly on the highly challenging BIRD benchmarks, KaSLA surpasses baselines by more than **5.72%** .

## 2 PRELIMINARIES

**Notations.** Given a natural language query $\boldsymbol{q}$ and a database schema $\mathcal{S} = \{s_1, \cdots, s_{|\mathcal{S}|}\}$, where $s_i$ represents either a table or a column in the database, conveying the structured representation of the corresponding database. The database schema $\mathcal{S}$ can be further divided into two subsets: the set of tables $\mathcal{T} = \{t_1, \cdots, t_{|\mathcal{T}|}\}$ and the set of columns $\mathcal{C} = \{c_1, \cdots, c_{|\mathcal{C}|}\}$, such that $\mathcal{S} = \mathcal{T} \cup \mathcal{C}$. For each table $t \in \mathcal{T}$, its corresponding set of columns is denoted as $\mathcal{C}_t = \{c_1, \cdots, c_{|\mathcal{C}_t|}\}$.

**Knapsack Problem of Schema Linking.** As discussed in the introduction, schema linking performance is negatively impacted by both missing and redundant elements, indicating that schema linking is not a simple selection problem. For example, directly feeding all schemas into the text-to-SQL models can capture all possible schemas but will suffer from heavy redundancy. On the other hand, when the total number of schemas is limited, there is a higher possibility of missing important ones.

Motivated by this, schema linking can be defined as a knapsack problem (Fréville, 2004), where the objective is to maximize the information contained in the schema linking set while controlling the total weight of the set. Formally, we have the following definition:

**Definition 2.1** (*Knapsack Schema Linking Problem*). For any given query $q$, let $C_q$ represent the total weight capacity of schemas for this query. For each schema $s$, let $V_{s,q}$ denote the value of the schema element given user query $q$, and let $W_{s,q}$ denote the weight of the schema given the query.

The optimal schema linking can be achieved by solving the following optimization problem:

$$\boldsymbol{S}^* = \arg\max_{\boldsymbol{S} \subseteq \mathcal{S}} \sum_{s \in \boldsymbol{S}} V_{s,q}, \quad \text{subject to} \quad \sum_{s \in \boldsymbol{S}} W_{s,q} \le C_q. \tag{1}$$

In this formulation, the objective is to maximize the total value of the selected schema elements while ensuring that the total weight of the selected elements does not exceed the capacity $C_q$. By solving this knapsack problem, we can obtain the optimal schema linking set that balances the trade-off between including relevant information and minimizing redundancy.

To solve the Knapsack Schema Linking Problem, we need to estimate the value $V_{s,q}$ and weight $W_{s,q}$ of each schema element given the user query, as well as determine the capacity $C_q$ for each query. In the following sections, we will first introduce the schema linking benchmark and then introduce our proposed KaSLA.

## 3 SCHEMA LINKING BENCHMARKING

In this section, we first describe the construction of our benchmark dataset and then propose two specially designed metrics for schema linking evaluation.

---

[1] https://anonymous.4open.science/r/iclr2025-SL-benchmark/

[2] https://anonymous.4open.science/r/iclr2025KaSLA/

## 3.1 BENCHMARK CONSTRUCTION

Our schema linking benchmark dataset is designed to facilitate the training and evaluation of schema linking models, aiming to inspire further research in developing increasingly powerful schema linking techniques. Each instance in the benchmark comprises a query, a full schema, and the corresponding ground truth schema linking. Formally, the benchmark dataset can be expressed as $\mathcal{B} = \{(q_i, \mathcal{S}, \boldsymbol{S}_i)\}_{i=1}^{|\mathcal{B}|}$, where $q_i$ represents the user query, $\mathcal{S}$ denotes the full schema, and $\boldsymbol{S}_i$ is the ground truth schema linking result for the $i$-th sample.

We have compiled two benchmarks based on the widely used Spider (Yu et al., 2018) and BIRD (Li et al., 2023c) datasets, each containing over ten thousand records.

## 3.2 SCHEMA LINKING METRICS

**Limitations of Previous Studies** Previous text-to-SQL research usually overlooked the evaluation of schema linking (Pourreza & Rafiei, 2023). Other works mentioned schema liking performance still use classification metrics such as AUC (Li et al., 2023a; 2024b), Recall, Precision, and F1 scores (Qu et al., 2024). However, any missing of the necessary element will result in wrong SQL generation. Besides, Recall, Precision, and F1 cannot quantitatively evaluate the actual linking results. AUC is also not suitable because schema linking is an imbalanced classification task. To this end, we propose two metrics that evaluate the missing and redundancy rates that directly impact the SQL generation performance rather than traditional metrics. We provide a detailed discussion of the limitations of traditional metrics and compare them with the proposed metrics in the Appendix G.

**Schema Missing Rate** Since the missing of even one necessary schema element will highly impact the SQL generation performance, the $\mathcal{R}_{miss}$ is designed to be a strict metric. Specifically, for each instance, if the ground truth schema linking is not included in the predicted schema linking, this instance gets a fail score of 1. Conversely, if the ground truth schema linking is totally included, the instance receives a success score of 0, indicating no missing. Formally, given the query $q$, the predicted schema linking result $\widehat{\boldsymbol{S}}_q$, and the ground truth schema linking result $\boldsymbol{S}_q$, the Schema Missing Rate $\mathcal{R}_{miss}$ can be defined as:

$$\mathcal{R}_{miss} = \frac{1}{|\mathcal{B}|} \sum_{(q,\mathcal{S},\boldsymbol{S}) \in \mathcal{B}} \mathbb{1}(\boldsymbol{S}_q \nsubseteq \widehat{\boldsymbol{S}}_q), \tag{2}$$

where $\mathbb{1}(\cdot)$ is an indicator function that returns 1 if the condition inside is satisfied and 0 otherwise. As defined above, any missing element in $\widehat{\boldsymbol{S}}_q$ will result in a failure for that instance.

**Schema Redundancy Rate** Different from the missing condition, where even minor missing will result in wrong SQL statements, for redundancy, the LLMs have certain but not much anti-interference ability, where a slight redundancy will not impact the SQL statement. However, as shown in Figure 1, heavy redundancy, such as inputting full schema, will have far lower SQL generation performance compared with inputting schema linkings. Motivated by this, Schema Redundancy Rate ($\mathcal{R}_{redun}$) describes how many portions of schemas are redundant, where the formal definition is:

$$\mathcal{R}_{redun} = \frac{1}{|\mathcal{B}|} \sum_{(q,\mathcal{S},\boldsymbol{S}) \in \mathcal{B}} \underbrace{(\frac{|\widehat{\boldsymbol{S}}_q \setminus \boldsymbol{S}_q|}{|\widehat{\boldsymbol{S}}_q|})^{\sigma}}_{\text{Redundancy Rate}}, \quad \underbrace{\sigma = \mathbb{1}(\boldsymbol{S}_q \subseteq \widehat{\boldsymbol{S}}_q)}_{\text{Non-missing Indicator}}. \tag{3}$$

Here, $|\widehat{\boldsymbol{S}}_q \setminus \boldsymbol{S}_q|$ is the number of redundant elements which is predicted but not present in the ground truth. Note that we only consider the redundancy rate of the non-missed prediction, and the redundancy of the missed prediction will directly be treated as 1, which means a failure. Since if the prediction is null set $\Phi$, the redundancy will be unreasonably computed as 0. Based on this formulation, a lower $\mathcal{R}_{redun}$ indicates better schema linking performance with fewer redundancy.

By employing both $\mathcal{R}_{miss}$ and $\mathcal{R}_{redun}$, we provide a more comprehensive and unbiased evaluation of schema linking models, ensuring that both completeness and redundancy are adequately assessed. This approach offers a nuanced understanding of model performance, directly applicable to the task of SQL generation.

# 4 KNAPSACK SCHEMA LINKING AGENT (KASLA)

We first introduce the definition and training process of the nomination-guaranteed score function. Then we discuss the estimation of key knapsack factors using the importance score. Finally, we present the full hierarchical KaSLA schema linking process.

## 4.1 NOMINATION-GUARANTEED SCORE FUNCTION

**Overview** The key factor of KaSLA is evaluating the importance score for each given schema element. We introduce a hybrid nomination-guaranteed score function to accomplish this task. The nomination part identifies high-confidence elements using a heuristic approach, while the guaranteed part provides a basic importance score for each element. Formally, the nomination-guaranteed function can be expressed as:

$$I_{s,q} = \min(\mathcal{I}(s \mid q, \mathcal{S}), 1), \quad \mathcal{I}(s \mid q, \mathcal{S}) = \mathcal{I}_{nomi}(s \mid q, \mathcal{S}) + \alpha \mathcal{I}_{guar}(s \mid q, \mathcal{S}), \tag{4}$$

where $\mathcal{I}_{nomi}(s \mid q, \mathcal{S})$ returns a $\{0, 1\}$ value that directly nominates elements, $\mathcal{I}_{guar}(s \mid q, \mathcal{S})$ assigns a soft $[0, 1]$ value to each element, and $\alpha$ controls the contribution of the guaranteed score. We set an upper bound of 1 for $I_{s,q}$ to avoid introducing overly strong contrasting relationships between the potentially matched elements.

**Nomination Scoring Model** The nomination scoring model effectively returns a subset given the full schema, formally expressed as $\widehat{\boldsymbol{S}}_{nomi} \subseteq \mathcal{S}$ given the input $(q, \mathcal{S})$. Thus, $\mathcal{I}_{nomi}(s \mid q, \mathcal{S}) = 1$ if $s \in \widehat{\boldsymbol{S}}_{nomi}$ and $\mathcal{I}_{nomi}(s \mid q, \mathcal{S}) = 0$ if $s \notin \widehat{\boldsymbol{S}}_{nomi}$.

To best meet these requirements, we employ Large Language Models (LLMs) (Lozhkov et al., 2024) as the nomination model. Specifically, we utilize StarCoder2, fine-tuning it on our benchmark using the ground truth schema linking result $\boldsymbol{S}$ as the target for schema prediction. The loss function is formulated as:

$$\mathcal{L}_{nomi} = - \sum_{(q, \mathcal{S}, \boldsymbol{S}) \in |\mathcal{B}|} \log P(\boldsymbol{S} \mid q, \mathcal{S}). \tag{5}$$

**Guaranteed Scoring Model** The guaranteed scoring model $\mathcal{I}_{guar}(s \mid q, \mathcal{S}) = P(s \in \boldsymbol{S})$ assigns an importance value to any given element, representing the predicted probability of $s \in \boldsymbol{S}$.

We design the guaranteed scoring model from a semantic perspective. First, we use a RoBERTa-Large model $f_{emb}(\cdot)$ to obtain semantic embeddings of all schema elements $f_{emb}(s)$, where $s \in \mathcal{S}$, and the query embedding $f_{emb}(q)$. Then, following the approach of Li et al. (2023a; 2024b), we employ a cross-attention network $f_{att}(\cdot)$ to jointly embed the semantic embeddings of tables and their columns for each element $s \in \mathcal{S}$. The importance score is then predicted as $(\mathbf{W} \cdot f_{att}(f_{emb}(s)) + \mathbf{b}) \cdot f_{emb}(q)$, where $\mathbf{W}$ and $\mathbf{b}$ are learnable parameters.

We train this guaranteed scoring model using our constructed benchmark. Specifically, we use the presence of $s$ in $\boldsymbol{S}$ as a training objective. The detailed loss function is defined as:

$$\mathcal{L}_{guar} = \sum_{(q, \mathcal{S}, \boldsymbol{S}) \in |\mathcal{B}|} \sum_{s \in \mathcal{S}} \text{FL}(\mathbb{1}(s \in \boldsymbol{S}), \mathcal{I}_{guar}(s \mid q, \mathcal{S})), \tag{6}$$

where $\text{FL}(\cdot)$ is the focal loss function (Ross & Dollár, 2017), designed to focus learning on hard negative samples.

## 4.2 KNAPSACK FACTOR ESTIMATION

As formulated in Eq. (1), for a given schema element $s$ and query $q$, KaSLA employs $V_{s,q}$ and $W_{s,q}$ to represent the value and weight of $s$, respectively. For query $q$, KaSLA utilizes $C_q$ to denote the total weight capacity. Unlike traditional knapsack problems Fréville (2004), KaSLA lacks ground truth values for these factors. This section proposes three estimation functions to predict the aforementioned factors, enabling KaSLA to select the most valuable element subsets while adhering to the total weight capacity constraint.

**Value Estimation**   Schemas with higher importance scores should be assigned higher values for inclusion in KaSLA. Consequently, any positive correlation function can map the importance score $I_{s,q}$ to the value estimation $\widehat{V}_{s,q}$. For simplicity, we employ an identity transformation as the most straightforward mapping function:

$$\widehat{V}_{s,q} = f_V(s, q) = I_{s,q}. \tag{7}$$

While more sophisticated value estimation functions could be designed, this simple yet effective solution proves powerful in practice.

**Weight Estimation**   For weight estimation, elements with higher importance should be assigned lower weights than those with lower importance scores. Thus, the weight estimation employs a negative correlation function to map importance scores to weight estimations. Specifically, we define the element weight $\widehat{W}_{s,q}$ using a reciprocal power function of the importance $I_{s,q}$ to represent the likelihood that introducing an element will add redundancy to the final schema linking results.

For schema element $s \in \mathcal{S}$, the prediction function of $W$ is formulated as:

$$\widehat{W}_{s,q} = f_W(s, q) = \lfloor (I_{s,q} - \mathbb{E}_{I \geq \tau}(\mathcal{S}, q) + 1)^{-1} \rfloor, \tag{8}$$

where $\mathbb{E}_{I \geq \tau}(\mathcal{S}, q)$ is the expectation of element importance scores greater than a hyperparameter $\tau$:

$$\mathbb{E}_{I \geq \tau}(\mathcal{S}, q) = \mathbb{E}\left[ \{ I_{s,q} \mid I_{s,q} \geq \tau \}_{s \in \mathcal{S}} \right]. \tag{9}$$

The rounding operation in $f_W(s, q)$ is designed to reduce the time and space complexity of subsequent Knapsack optimization. We use $\mathbb{E}_{I \geq \tau}(\mathcal{S}, q)$ to denote the average importance of elements with high importance scores. $f_W(s, q)$ computes the reciprocal of the difference between each element's importance $I_{s,q}$ and an expected high importance score. A large negative difference indicates that the element deviates significantly from other highly important ones, suggesting a higher likelihood of redundancy.

**Capacity Estimation**   To constrain redundancy in the schema linking process, we define weight capacity $C$ as the maximum allowable weight for KaSLA. We begin by selecting a top-$K_C$ similar demonstration set $D$ from the training dataset, based on the similarity between the user queries in the training dataset and the given query $q$. For each sample $d$ in $D$, we define a schema element set that includes all ground truth elements as well as all elements with an estimated importance greater than the ground truth ones. We denote this set as $\text{Knap}(d)$ to represent an assumed full knapsack:

$$\text{Knap}(d) = \boldsymbol{S}_d + \{ s \mid I_{s,q_d} \geq \min\{ I_{z,q_d} \}_{z \in \boldsymbol{S}_d} \}_{s \in \mathcal{S}_d \setminus \boldsymbol{S}_d}, \tag{10}$$

where $\boldsymbol{S}_d$ is the ground truth linking set, and $\mathcal{S}_d$ is the original schema of $d$. We then calculate the sum of the predicted weights of all elements in $\text{Knap}(d)$ to represent the assumed capacity of $d$. Accordingly, we define the prediction function of $C$ as follows:

$$\widehat{C}_q = f_C(q) = \gamma \cdot \max\{ \sum_{s \in \text{Knap}(d)} \widehat{W}_{s,q} \}_{d \in D}, \tag{11}$$

where $\gamma > 0$ is a hyperparameter. The maximum capacity among all samples in the most similar demonstrations of $q$ provides a robust prediction of the capacity.

### 4.3   HIERARCHICAL KASLA SCHEMA LINKING

Our KaSLA operates in a hierarchical manner to effectively address schema linking challenges. The process unfolds in two stages: first, KaSLA performs table linking, and subsequently, for each selected table, it simultaneously accomplishes column linking. This hierarchical approach enables KaSLA to efficiently reduce the dimensionality of the schema space, thereby enhancing its capability to handle large-scale and complex schema linking and text-to-SQL generation tasks encountered in real-world applications.

**KaSLA Agent Details** The core of KaSLA's schema linking process is formulated as an optimization problem, which we solve using a tailored integer 0-1 knapsack dynamic programming algorithm. Given the estimated values $\widehat{V}_{s,q}$ and weights $\widehat{W}_{s,q}$ for all elements $s$, and the estimated capacity $\widehat{C}_q$ of the query $q$, we aim to solve:

$$\boldsymbol{S}^* = \arg\max_{\boldsymbol{S} \subseteq \mathcal{S}} \sum_{s \in \boldsymbol{S}} \widehat{V}_{s,q}, \quad \text{subject to} \quad \sum_{s \in \boldsymbol{S}} \widehat{W}_{s,q} \leq \widehat{C}_q. \tag{12}$$

This optimization problem is efficiently solved using dynamic programming, leveraging the discrete nature of the weights and capacity. The algorithm systematically builds an optimal solution by considering all possible element combinations within the capacity constraint, ensuring that KaSLA selects the most valuable elements while respecting the query's capacity limit. This approach not only guarantees an optimal solution but also provides insights into the trade-offs between element value and computational capacity, allowing for adaptable and efficient schema linking across diverse query complexities.

**Hierarchical KaSLA Strategy** The hierarchical nature of KaSLA allows for a more efficient and scalable approach to schema linking. This strategy is implemented in two distinct phases: table linking and column linking. The full algorithm is provided in Appendix D Algorithm 1 and the complexity analysis is provided in Appendix E.

TABLE LINKING PHASE In the first phase, KaSLA focuses on identifying the relevant tables for the given query. Let $\boldsymbol{T}$ be the set of all available tables and $q$ be the input query. The table linking optimization problem is formulated as:

$$\boldsymbol{T}^* = \arg\max_{\boldsymbol{T} \subseteq \mathcal{T}} \sum_{t \in \boldsymbol{T}} \widehat{V}_{t,q}, \quad \text{subject to} \quad \sum_{t \in \boldsymbol{T}} \widehat{W}_{t,q} \leq \widehat{C}_q^t, \tag{13}$$

where $\widehat{V}_{t,q}$ and $\widehat{W}_{t,q}$ are the estimated value and weight of table $t$ for query $q$, respectively, and $\widehat{C}_q^t$ is the estimated capacity for tables in query $q$.

COLUMN LINKING PHASE Following the table linking, KaSLA proceeds to select relevant columns for each chosen table. For each table $t \in \boldsymbol{T}^*$, let $\boldsymbol{C}_t$ be the set of columns in table $t$. The column linking optimization problem for each table is defined as:

$$\boldsymbol{C}_t^* = \arg\max_{\boldsymbol{T}' \subseteq \boldsymbol{T}_t} \sum_{c \in \boldsymbol{T}'} \widehat{V}_{c,q,t}, \quad \text{subject to} \quad \sum_{c \in \boldsymbol{T}'} \widehat{W}_{c,q,t} \leq \widehat{C}_{q,t}^c, \tag{14}$$

where $\widehat{V}_{c,q,t}$ and $\widehat{W}_{c,q,t}$ are the estimated value and weight of column $c$ in table $t$ for query $q$, respectively, and $\widehat{C}_{q,t}^c$ is the estimated capacity for columns of table $t$ in query $q$. The final schema linking result $\boldsymbol{S}^*$ is the union of all selected tables and the corresponding selected columns:

$$\widehat{S}_{\text{KaSLA}} = \boldsymbol{T}^* \cup \bigcup_{t \in \boldsymbol{T}^*} \boldsymbol{C}_t^*. \tag{15}$$

**Final KaSLA Application** In the deployment phase, both during training and inference, KaSLA can integrate with any text-to-SQL model $\mathcal{M}$. This flexibility allows KaSLA to enhance the performance of existing state-of-the-art models while maintaining their underlying strengths. Given an input query $q$, KaSLA's predicted schema linking $\widehat{S}_{\text{KaSLA}}$, and the ground truth schema linking result $\boldsymbol{S}$, the training process of a SQL generation model is formulated as follows:

$$\widehat{y} = \arg\max_{y \in \mathcal{Y}} P_{\mathcal{M}}(y|q, \widehat{S}_{\text{KaSLA}} \cup \boldsymbol{S}), \tag{16}$$

where $\mathcal{Y}$ represents the target SQL queries, and $P_{\mathcal{M}}(y|q, \widehat{S}_{\text{KaSLA}} \cup \boldsymbol{S})$ denotes the probability assigned by model $\mathcal{M}$ to token $y$ in $\mathcal{Y}$, conditioned on the input query $q$ and the unified schema information. During the inference stage, only the schema linking results generated by KaSLA are used:

$$\widehat{\mathcal{Y}} = \mathcal{M}(q, \widehat{S}_{\text{KaSLA}}), \tag{17}$$

where $\widehat{\mathcal{Y}}$ represents the predicted SQL queries.

## 5 EXPERIMENTS

We conducted comprehensive experiments on two public text-to-SQL datasets to evaluate KaSLA and address the following research questions: **RQ1:** Does KaSLA outperform existing text-to-SQL baselines? **RQ2:** Can KaSLA enhance the performance of other text-to-SQL models? **RQ3:** Does KaSLA demonstrate better schema linking performance? **RQ4:** Is KaSLA a solution for real-world text-to-SQL applications without training data? (We addressed RQ4 in Appendix F.)

### 5.1 EXPERIMENT SETUP

We conducted experiments on two well-known large-scale text-to-SQL datasets, BIRD (Li et al., 2023c) and Spider (Yu et al., 2018). We included multiple robust baselines based on LMs, LLMs with in-context learning, and LLMs with fine-tuning. For evaluation, we used the proposed metrics Schema Missing Rate ($\mathcal{R}_{miss}$) and Schema Redundancy Rate ($\mathcal{R}_{redun}$) for schema linking, as well as Execution Accuracy (EX) (Yu et al., 2018; Li et al., 2023c) and Valid Efficiency Score (VES) (Li et al., 2023c) for text-to-SQL. The detailed experimental settings are presented in Appendix B.

### 5.2 MAIN RESULTS OF SQL GENERATION

To address **RQ1**, we conduct experiments to evaluate the SQL generation ability of our KaSLA and other baselines with Execution Accuracy (EX) and Valid Efficiency Score (VES) on the BIRD-dev and Spider-dev datasets and report the results in Table 1.

**KaSLA's superior performance.** Based on the experimental results, KaSLA demonstrates outstanding performance across both BIRD-dev and Spider-dev. KaSLA achieves the highest overall EX (63.75%) and VES (69.68%)on both BIRD-dev and Spider-dev datasets, clearly surpassing all other approaches. While its scores on Spider-dev are 88.01% EX and 86.06% VES, significantly outperforming other models.

**Comparison with GPT-4 ICL baseline.** In comparison to the best In-Context Learning (ICL) baseline, which uses the powerful GPT-4, KaSLA shows remarkable improvements. On BIRD-dev, KaSLA outperforms the best ICL baseline by 6.77% in EX and 5.56% in VES. On the Spider-dev dataset, KaSLA's EX improvement compared to GPT4-based methods is slightly lower, this discrepancy can be attributed to the inherently simpler and more regular schema of the Spider-dev dataset. A simpler schema can be processed well without specially designed linking methods, thereby making advanced systems like KaSLA appear less dominant in terms of improvement gains. Consequently, these improvements on BIRD-dev highlight KaSLA's advanced capabilities, particularly in more intricate database environments.

**Performance across difficulty levels.** When evaluating the models based on the difficulty levels, KaSLA consistently excels across all difficulty levels—easy, medium, and hard. On BIRD-dev, KaSLA achieves significant EX improvements of 4.20%, 18.50%, and 1.58% over the best ICL baselines and 2.54%, 12.55%, and 12.28% over the best ICL baselines on easy, medium, and hard queries, respectively. On Spider-dev, while the overall improvement is smaller, KaSLA still shows performance enhancements across hard and extra hard queries compared with the best SFT baseline.

### 5.3 APPLY KASLA TO EXISTING TEXT-TO-SQL MODELS

To address **RQ2**, we evaluated the impact of integrating KaSLA as a plug-in model into existing text-to-SQL methods and provided the results in Table 2. For our analysis, we specifically incorporated the schema elements linked by KaSLA into prominent text-to-SQL baselines. Through this integration, KaSLA's enhanced schema linking capabilities are utilized as part of the input prompt for SQL generation, promising improvements in both EX and VES.

**Improvements with KaSLA.** As shown in Table 2, integrating KaSLA leads to noticeable improvements. For instance, the combination of CodeS-15B-SFT with KaSLA demonstrates an improvement in overall EX on the BIRD-dev dataset from 58.08% to 60.95%, alongside a boost in VES from 59.87% to 66.11%. Similar enhancements are observed with DAIL-SQL , E-SQL, and CHESS when augmented with KaSLA, notably improving EX and VES across various difficulty levels and datasets.

Table 1: The text-to-SQL performance of our KaSLA and three main types of baselines: regular LMs, In-Context Learning (ICL) with LLMs, and Supervised Fine-Tuning (SFT) with LLMs, with Execution Accuracy (EX) (%) and Valid Efficiency Score (VES) (%) on BIRD-dev and Spider-dev datasets. The numbers in parentheses next to each method (e.g., 23', 24') represent the release year of the respective models or methods. (SC) refers DAIL-SQL with self-consistency.

| Type | Method | Text-to-SQL Model | BIRD-dev | | | | | Spider-dev | | | | | |
|---|---|---|---|---|---|---|---|---|---|---|---|---|---|
| | | | EX | | | | VES | EX | | | | | VES |
| | | | Easy | Medium | Hard | Total | Total | Easy | Medium | Hard | Extra | Total | Total |
| LMs | ResdSQL (23') | T5-Base | 42.27 | 20.22 | 15.97 | 33.12 | 32.85 | 91.94 | 83.63 | 68.39 | 51.81 | 77.95 | 77.71 |
| | | T5-Large | 46.49 | 27.96 | 22.92 | 38.66 | 40.62 | 93.55 | 85.43 | 72.41 | 53.61 | 80.08 | 79.72 |
| | | T5-3B | 53.51 | 33.33 | 16.67 | 43.94 | 44.42 | 94.76 | 87.67 | 72.99 | 56.02 | 81.82 | 80.89 |
| ICL | C3-SQL (23') | GPT-3.5 | 58.92 | 38.49 | 31.94 | 50.20 | 50.77 | 92.74 | 85.20 | 77.59 | 62.05 | 82.01 | 80.09 |
| | MAC-SQL (23') | GPT-4 | - | - | - | 57.56 | 58.76 | - | - | - | - | 86.75 | - |
| | DIN-SQL (23') | | - | - | - | 50.72 | 58.79 | 92.34 | 87.44 | 76.44 | 62.65 | 82.79 | 81.70 |
| | DAIL-SQL (23') | | 62.49 | 43.44 | 38.19 | 54.43 | 55.74 | 91.53 | 89.24 | 77.01 | 60.24 | 83.08 | 83.11 |
| | DAIL-SQL (SC) | | 63.03 | 45.81 | 43.06 | 55.93 | 57.20 | 91.53 | 90.13 | 75.29 | 62.65 | 83.56 | - |
| | TA-SQL (24') | | 63.14 | 48.82 | 36.81 | 56.32 | - | 93.50 | 90.80 | 77.60 | 64.50 | 85.00 | - |
| | SuperSQL (24') | | 66.92 | 46.67 | 43.75 | 58.60 | 60.62 | 94.35 | 91.26 | 83.33 | 68.67 | 87.04 | 85.92 |
| | Dubo-SQL (24') | | - | - | - | 59.71 | 66.01 | - | - | - | - | - | - |
| SFT | Pure-SL(24') | StarCoder2-15B | 56.65 | 43.23 | 31.94 | 50.26 | 57.09 | 93.15 | 87.89 | 74.71 | 59.04 | 82.30 | 80.63 |
| | DTS-SQL (24') | | 63.03 | 46.02 | 34.72 | 55.22 | 64.17 | 91.94 | 90.58 | 78.74 | 66.87 | 85.11 | 85.49 |
| | TA-SL (24') | | 59.89 | 45.38 | 34.72 | 53.13 | 60.51 | 95.56 | 92.38 | 79.89 | 63.25 | 86.36 | 85.03 |
| | CodeS (24') | | 68.00 | 51.40 | 39.58 | 60.30 | 65.04 | 94.76 | 91.26 | 72.99 | 64.46 | 84.72 | 83.52 |
| | **KaSLA (Ours)** | | **69.73** | **57.85** | **44.44** | **63.75** | **69.68** | **96.77** | **93.27** | 82.76 | 66.27 | **88.01** | **86.06** |
| | %Improv. vs. the best ICL baseline | | +4.20% | +18.50% | +1.58% | +6.77% | +5.56% | +2.56% | +2.20% | -0.68% | -3.49% | +1.11% | +0.16% |
| | %Improv. vs. the best SFT baseline | | +2.54% | +12.55% | +12.28% | +5.72% | +7.13% | +1.27% | +0.96% | +3.59% | +2.81% | +1.91% | +3.04% |

Table 2: The text-to-SQL performance of adding KaSLA as a plug-in model dedicated to schema linking to other text-to-SQL methods on BIRD-dev and Spider-dev.

| Method | Text-to-SQL Model | BIRD-dev | | | | | Spider-dev | | | | | |
|---|---|---|---|---|---|---|---|---|---|---|---|---|
| | | EX | | | | VES | EX | | | | | VES |
| | | Easy | Medium | Hard | Total | Total | Easy | Medium | Hard | Extra | Total | Total |
| CodeS | CodeS-15B-SFT | 65.62 | 49.68 | 36.81 | 58.08 | 59.87 | 95.97 | 89.01 | 75.29 | 62.05 | 84.04 | 81.74 |
| CodeS **+ KaSLA** | | 68.54 | 52.04 | 40.97 | 60.95 | 66.11 | 95.97 | 90.58 | 75.86 | 59.64 | 84.43 | 83.06 |
| DTS-SQL | StarCoder2-15B | 63.03 | 46.02 | 34.72 | 55.22 | 64.17 | 91.94 | 90.58 | 78.74 | 66.87 | 85.11 | 85.49 |
| DTS-SQL **+ KaSLA** | | 65.95 | 49.89 | 37.50 | 58.41 | 65.56 | 96.37 | 92.38 | 79.31 | 69.28 | 87.43 | 85.73 |
| TA-SL | StarCoder2-15B | 59.89 | 45.38 | 34.72 | 53.13 | 60.51 | 95.56 | 92.38 | 79.89 | 63.25 | 86.36 | 85.03 |
| TA-SL **+ KaSLA** | | 65.30 | 49.46 | 38.89 | 58.02 | 65.88 | 95.97 | 92.83 | 80.46 | 64.46 | 86.94 | 85.58 |
| DAIL-SQL | GPT-4 | 62.49 | 43.44 | 38.19 | 54.43 | 55.74 | 91.53 | 89.24 | 77.01 | 60.24 | 83.08 | 83.11 |
| DAIL-SQL **+ KaSLA** | | 64.86 | 49.89 | 38.89 | 57.89 | 59.39 | 95.97 | 90.13 | 84.48 | 66.87 | 86.85 | 85.51 |
| CHESS | GPT-4 | 69.51 | 57.20 | 40.97 | 63.10 | 67.23 | 95.97 | 93.05 | 81.61 | 63.86 | 87.14 | 85.66 |
| CHESS **+ KaSLA** | | 69.84 | 58.28 | 43.75 | 63.89 | 68.72 | 97.18 | 93.50 | 82.18 | 66.87 | 88.20 | 86.12 |
| E-SQL | GPT-4o | 70.70 | 60.43 | 45.83 | 65.25 | 70.01 | 97.18 | 93.72 | 83.33 | 65.66 | 88.30 | 86.38 |
| E-SQL **+ KaSLA** | | 71.14 | 61.29 | 47.22 | 65.91 | 70.86 | 97.58 | 93.72 | 83.91 | 66.87 | 88.68 | 86.82 |

**KaSLA's adaptability and robustness.** These results underscore KaSLA's flexibility and effectiveness as a plug-in model. The performance uplift attributed to KaSLA supports its invaluable role in improving complex text-to-SQL tasks, making it a versatile and powerful addition to any existing system. This adaptability not only confirms KaSLA's robustness but also promotes its universal applicability across different models to achieve superior performance outcomes.

## 5.4 EXPERIMENTAL RESULTS OF SCHEMA LINKING

To answer the **RQ3**, we conduct the experiments and report the benchmark results of schema linking methods on BIRD-dev in Table 3 and report the results on Spider-dev in Appendix I Table 12. We utilize the proposed evaluation metrics: Schema Redundancy Rate ($\mathcal{R}_{miss}$) and Schema Redundancy Rate ($\mathcal{R}_{redun}$). We also involve a combined metric ($\mathcal{R}_{correct} = 1 - (\mathcal{R}_{miss} + \mathcal{R}_{redun})/2$) to evaluate the comprehensive ability of schema linking model to avoid missing and redundancy.

**KaSLA's superior performance.** The experimental results demonstrate that our KaSLA method exhibits significantly lower missing rates and redundancy rates, resulting in higher overall correctness rates, consistently achieving a superior performance compared to several other schema linking methods. For instance, on the BIRD-dev dataset, KaSLA attains an impressive $\mathcal{R}_{correct}$ of 85.99% (table) and 75.97% (column), far surpassing other methods such as CodeS-SL or DTS-SL.

Table 3: Schema linking benchmarks with $R_{miss}$ and $R_{redun}$ on BIRD-dev. We define a comprehensive metric $\mathcal{R}_{correct} = 1 - (\mathcal{R}_{miss} + \mathcal{R}_{redun})/2$ to evaluate the ability of schema linking model to avoid element missing and redundancy.

| Dataset | Metric | | Schema linking Method | | | | | | | | | | KaSLA (Ours) |
|---|---|---|---|---|---|---|---|---|---|---|---|---|---|
| | | | CodeS-SL | | | 10shot ICL-SL | | DIN-SL | | DTS-SL | Pure-SL | TA-SL | |
| | | | RoBERTa | | SGPT -1.3B | GPT-3.5 | GPT-4 | GPT-3.5 | GPT-4 | StarCoder2-15B | | | |
| | | | Base | Large | | | | | | | | | |
| BIRD-dev | $\mathcal{R}_{miss} \downarrow$ | table | 1.56 | **0.85** | 3.59 | 33.57 | 31.23 | 49.61 | 36.83 | 35.01 | 35.01 | 21.32 | 5.21 |
| | | colmn | 15.00 | 12.53 | 15.94 | 93.84 | 91.49 | 96.18 | 93.64 | 35.77 | 57.54 | 43.80 | **5.83** |
| | $\mathcal{R}_{redun} \downarrow$ | table | 57.43 | 56.97 | 58.48 | 42.81 | 35.74 | 52.31 | 39.78 | 39.71 | 39.71 | 25.77 | **22.80** |
| | | colmn | 81.50 | 80.83 | 82.13 | 95.29 | 92.18 | 96.49 | 94.13 | 83.04 | 61.75 | 46.25 | **42.24** |
| | $\mathcal{R}_{correct} \uparrow$ | table | 70.50 | 71.09 | 68.97 | 61.81 | 66.52 | 49.04 | 61.69 | 62.64 | 62.64 | 76.46 | **85.99** |
| | | colmn | 51.75 | 53.32 | 50.96 | 5.44 | 8.16 | 3.66 | 6.12 | 40.60 | 40.36 | 54.97 | **75.97** |

**Balancing metrics for effectiveness.** Despite some methods achieving lower $R_{miss}$ or $R_{redun}$ individually, they fail to strike a balance between the two metrics. For example, CodeS-SL with a RoBERTa-Large achieves the lowest $R_{miss}$ of 0.85% for table linking on BIRD-dev, yet its $R_{redun}$ is quite high at 56.97%, leading to a lower $R_{correct}$. In contrast, KaSLA not only maintains a low $R_{miss}$ but also substantially reduces $R_{redun}$, thus demonstrating its comprehensive capability in schema linking. This ability to effectively avoid both element missing and redundancy verifies KaSLA's robustness and efficacy, making it a reliable choice for schema linking tasks in complex datasets.

## 6 LIMITATION

A natural limitation of KaSLA is the inference delay. KaSLA uses a fine-tuned StarCoder2-15B to generate nomination scores for the entire schema, aiming for improved semantic understanding, but this results in additional time consumption during inference. We analyzed the average processing time per instance during inference for each component of KaSLA on the BIRD and Spider datasets, as detailed in Table 4. We found that the nomination model with StarCoder2-15B contributes to this inference delay, while other components, such as the guaranteed model with RoBERTa-Large and the hierarchical knapsack optimization using dynamic programming, do not significantly increase time consumption. In the future, we aim to reduce the time usage of the nomination model when processing the entire schema, while maintaining accuracy.

Table 4: Inference time cost per instance of each component in KaSLA

| Dataset | Schema linking model | Nomination model with StarCoder2-15B | Guaranteed model with RoBERTa-Large | Factor estimation and dynamic programming | Text-to-SQL | Total |
|---|---|---|---|---|---|---|
| BIRD-dev | Full Schema | / | / | / | 14.35 s | 14.35 s |
| | KaSLA | 12.87 s | 0.12 s | < 0.01 s | 8.80 s | 21.80 s |
| Spider-dev | Full Schema | / | / | / | 9.69 s | 9.69 s |
| | KaSLA | 9.17 s | 0.06 s | < 0.01 s | 7.89 s | 17.13 s |

## 7 CONCLUSION

This paper introduced the Knapsack Schema Linking Agent (KaSLA), a novel approach to address the schema linking challenges in text-to-SQL tasks. By framing schema linking as a Knapsack problem, KaSLA effectively balances the trade-off between missing and redundant schema linkages, significantly enhancing the accuracy of SQL generation. Our proposed benchmarks and metrics provide a new standard for evaluating schema linking performance, fostering further advancements in the field. Extensive experiments demonstrate KaSLA's superiority over existing methods.

These findings highlight KaSLA's potential to revolutionize schema linking and advance the broader capabilities of text-to-SQL systems. By effectively reducing both missing and redundant information, KaSLA not only improves schema linking accuracy but also enhances the overall performance of SQL generation. The innovation of the nomination-guaranteed score function plays a crucial role in overcoming the missing and redundancy seesaw problem, offering a robust solution that can be integrated into mainstream text-to-SQL models. Our contributions pave the way for more precise and efficient database interactions, underscoring the transformative impact of KaSLA in the field.

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

## A  RELATED WORK

Text-to-SQL studies have witnessed significant advancements and development over the years in natural language processing (NLP) research (Wang et al., 2022; Qin et al., 2022). The techniques involved in text-to-SQL implementation have undergone a long period of evolution (Hong et al., 2024b). Early methods in text-to-SQL research largely focused on template and rule-based human

engineering (Li & Jagadish, 2014; Mahmud et al., 2015; Yu et al., 2021). Subsequently, with the emergence of deep learning (Katsogiannis-Meimarakis & Koutrika, 2023) and pre-trained language models (PLMs) (Liu et al., 2019; Devlin et al., 2019), text-to-SQL has further advanced along with these developments (Yin et al., 2020; Choi et al., 2021; Li et al., 2023b). Most recently, as LLMs have gained prevalence in both research-oriented papers and industrial projects, text-to-SQL systems integrating LLMs are now a research hotspot in the NLP and database communities. Our study follows an LLM-based text-to-SQL paradigm, consisting of in-context learning and fine-tuning techniques.

## A.1 IN-CONTEXT LEARNING-BASED TEXT-TO-SQL

At the inception of LLMs, in-context learning (ICL) and prompt engineering emerged as a core method driving advancements in the study of LLMs (Reynolds & McDonell, 2021). Early efforts in LLM-based text-to-SQL studied the effectiveness of ICL with different prompt designs on various LLMs (Liu et al., 2023). The natural language understanding (NLU) capabilities empowered by numerous training corpora enable LLMs to perform well in SQL generation with simple prompt engineering (Rajkumar et al., 2022). With the success of well-designed ICL methods in other NLU tasks (Wei et al., 2022; 2021), the integration into the text-to-SQL task has also achieved solid improvement (Tai et al., 2023). Chain-of-Thought (CoT) prompting (Wei et al., 2022) shows great potential in natural language reasoning tasks. DIN-SQL (Pourreza & Rafiei, 2023) proposed a CoT-based decomposed ICL framework with self-correction; ACT-SQL () designed a method to generate automatic CoT exemplars to enhance SQL reasoning. To optimize the prompt towards better quality, DAIL-SQL (Gao et al., 2024) introduces a few-shot sampling strategy, providing related samples for LLMs to learn from, and the Knowledge-to-SQL (Hong et al., 2024a) framework is proposed to generate helpful knowledge to assist SQL generation. Self-consistency (Wang et al., 2023) for LLM-based text-to-SQL ensures accuracy based on the execution result. The C3 (Dong et al., 2023) framework conducts a majority vote of consistency; SQL-PaLM (Sun et al., 2023) introduces an error-filtering-aided consistency decoding based on execution for better SQL generation. Multi-stage decomposition is also popular in text-to-SQL studies. MAC-SQL (Wang et al., 2024) employs a multi-agent collaboration framework to generate and refine SQL; TA-SQL (Qu et al., 2024) introduces a two-stage generation framework incorporating schema linking and logical synthesis. ICL-based methods are the mainstream paradigm of LLM-based text-to-SQL, which have made significant progress and continue to be widely studied in the most current work. CHASE-SQL (Pourreza et al., 2024) proposes an in-context learning prompt for question decomposition and involves a novel online synthetic example generation method to adapt LLMs to test datasets. Our proposed method also follows the ICL paradigm for text-to-SQL. Firstly, we generate a schema linking and then use adaptive evaluation to improve its quality, providing clear guidance to understand the user query and the corresponding database schema. Then, the LLMs incorporate the provided schema linking to generate accurate SQL queries.

## A.2 FINE-TUNING-BASED TEXT-TO-SQL

Fine-tuning is an intuitive and widely recognized technique for enabling LLMs to perform specific downstream tasks (Wei et al., 2021). The fine-tuning methods for aligning LLMs with instructional tasks are gradually evolving towards better performance and greater effectiveness (Rafailov et al., 2023). Even though code-specific LLMs are trained on massive programming scenarios, they still struggle when facing challenging user queries and complex database environments (Gan et al., 2021; Deng et al., 2021; Li et al., 2023c). As a straightforward method, supervised fine-tuning (SFT) (Wei et al., 2021) is utilized to adapt open-source LLMs for SQL generation, which elicits a solid improvement (Gao et al., 2024). For well-designed methods, CodeS (Li et al., 2024b) proposes a two-stage training framework. First, a backbone code-LLM is pre-trained on an incremental training corpus, followed by SFT on bi-directional augmented query-SQL pairs, achieving impressive performance with open-source LLMs. Instead of fine-tuning a single model, DTS-SQL (Pourreza & Rafiei, 2024) fine-tunes two LLMs separately for schema linking generation and SQL generation. This two-stage generation process elicits higher accuracy. Our work involves fine-tuning the LLMs and the PLMs based on the schema linking task, then incorporating the generated schema linking to assist in accurate SQL generation.

### A.3 SCHEMA LINKING

Schema linking is a crucial step in text-to-SQL tasks, involving the identification and association of natural language query elements with corresponding database schema components, such as tables and columns. This process significantly influences the overall performance of SQL generation and has been the focus of extensive research.

With the advent of Large Language Models (LLMs), schema linking has seen considerable innovation and potential. DIN-SQL (Pourreza & Rafiei, 2023) marked a significant advancement by using LLMs to generate schema linkages, utilizing the entire schema as input to leverage the LLMs' contextual understanding. Building on this, DAIL-SQL (Gao et al., 2024) incorporated historical query-SQL pairs as evidence, refining schema linking by utilizing past interactions. DELLM (Hong et al., 2024a) extended these capabilities with a specialized data expert LLM, providing additional knowledge to improve schema linking in complex scenarios. Similarly, E-SQL (Caferoğlu & Ulusoy, 2024) introduced a query enrichment method that incorporates relevant database elements to enrich user queries rather than simplifying the full schema. DTS-SQL (Pourreza & Rafiei, 2024) fine-tuned LLMs specifically for table linking and SQL generation, ensuring task-focused training at each process stage. TA-SQL (Qu et al., 2024) introduced the approach of generating a dummy SQL to abstract linked schemas via an LLM, allowing for iterative refinement before the final SQL generation. Conversely, CodeS (Li et al., 2024b) implemented a retrieval-based strategy, recalling semantically matching schema elements as inputs for SQL generation models. CHESS (Talaei et al., 2024) treated column linking as a binary classification task to eliminate obviously irrelevant columns and then linked tables and columns from the filtered schema. To address the challenge of missing elements, RSL-SQL (Cao et al., 2024) proposed bidirectional schema linking, with forward schema linking identifying potential matching elements from the full schema and backward schema linking extracting elements from preliminary SQL generated based on these potential matches. Distillery (Maamari et al., 2024) explored schema linking performance with extremely large LLMs like GPT-4o and Llama 3.1-405b, finding that these models can effectively process full schema without a schema linking model, while moderately-sized models, like Llama 3.1-8B, still rely on schema linking.

Effective schema linking demands no missing elements and minimal redundant elements; any missing schema element can cause SQL generation to fail directly, while redundant elements can confuse the process and lead to incorrect outputs. This challenge aligns with the knapsack problem—maximizing value while minimizing weight within constraints. Inspired by this, we formulated schema linking as a knapsack problem and proposed KaSLA, which aims to link the most relevant and least redundant schema element sets according to user queries.

## B DETAILED EXPERIMENT SETUP

In this section, we provide the experiment setup in the view of datasets, evaluation metrics, and baselines. We also report the implementation details of the proposed KaSLA and the used input format of LLMs in Appendix D.3.

**Datasets and Evaluation Metrics.** We conducted experiments on two well-known large-scale text-to-SQL datasets, BIRD (Li et al., 2023c) and Spider (Yu et al., 2018). Both datasets feature human-annotated queries and SQLs, complex database elements, and challenging cross-domain scenarios meticulously. The statistics for BIRD and Spider are reported in Appendix C Table 5. For evaluation, we evaluated our approach and baselines on both schema linking and text-to-SQL tasks. For schema linking evaluation, we employed the metrics introduced in Section 3.2: Schema Missing Rate ($\mathcal{R}_{miss}$) and Schema Redundancy Rate ($\mathcal{R}_{redun}$). For text-to-SQL evaluation, we measured performance using two well-established execution-based metrics: Execution Accuracy (EX) (Yu et al., 2018; Li et al., 2023c) and Valid Efficiency Score (VES) (Li et al., 2023c).

**Baselines.** To ensure a comprehensive and credible evaluation, we included multiple robust baselines from three categories: *(i) LMs baselines:* ResdSQL (Li et al., 2023a); *(ii) LLMs + in-context learning baselines:* C3-SQL (Dong et al., 2023), DIN-SQL (Pourreza & Rafiei, 2023), MAC-SQL (Wang et al., 2024), DAIL-SQL (Gao et al., 2024), SuperSQL (Li et al., 2024a), Dubo-SQL (Thorpe et al., 2024), TA-SQL (Qu et al., 2024), E-SQL (Caferoğlu & Ulusoy, 2024), CHESS (Talaei et al., 2024); and *(iii) LLMs + fine-tuning baselines:* DTS-SQL (Pourreza & Rafiei, 2024),

CodeS (Li et al., 2024b). We also involve PureSL, which means directly fine-tuning an LLM for schema linking in our benchmarks without any tailored design. For DIN-SQL, TA-SQL, CodeS, and DTS-SQL that include a schema linking process in their framework, we denoted them as DIN-SL, TA-SL, CodeS-SL, and DTS-SL. We reported the overall text-to-SQL performance and schema linking performance, respectively.

## C STATISTICS OF DATASET

We provide the statistics for the BIRD and Spider datasets in Table 5, and the imbalanced proportions of matching versus non-matching schema elements are illustrated in Figure 4.

As illustrated in Figure 4, schema linking is inherently an imbalanced classification task, where a small subset of elements matches the natural language query compared to the numerous non-matching ones. Previous research has primarily utilized metrics such as AUC (Li et al., 2023a; 2024b) for evaluating schema linking. In this imbalanced classification scenario, a schema linking model can achieve high AUC scores by predicting mostly irrelevant schema elements, leading to results replete with redundant elements. Additionally, missing any matching element directly results in incorrect SQL generation despite causing only minor fluctuations in AUC.

Such biased evaluations lead to sub-optimal outcomes and hinder further advancements in schema linking and SQL generation. This underscores the significance and potential value of our proposed Schema Missing Rate ($\mathcal{R}_{miss}$) and Schema Redundancy Rate ($\mathcal{R}_{redun}$), which provide more accurate and meaningful evaluations in the context of imbalanced schema element distributions.

| Dataset | | N | #DB | #Table / DB | | #Column / DB | | #Table / N | | #Column / N | |
|---|---|---|---|---|---|---|---|---|---|---|---|
| | | | | Avg. | Max | Avg. | Max | Avg. | Max | Avg. | Max |
| BIRD | train | 9428 | 69 | 7.57 | 65 | 51.29 | 455 | 2.00 | 6 | 4.47 | 16 |
| | dev | 1534 | 11 | 6.82 | 13 | 72.55 | 199 | 1.93 | 4 | 4.44 | 12 |
| Spider | train | 8659 | 146 | 5.43 | 26 | 27.79 | 352 | 1.71 | 6 | 3.35 | 13 |
| | dev | 1034 | 20 | 4.00 | 11 | 21.95 | 56 | 1.51 | 4 | 2.78 | 8 |

Table 5: The statistics of datasets. 'N' is the total number of samples in dataset. '#' denotes 'The number of'. 'Avg.' and 'Max' represent the average value and the maximum value, respectively. For example, 'Avg. #Table / DB' means the average number of tables per database, 'Max. #Column / N' means the maximum number of matching columns for instances in the dataset.

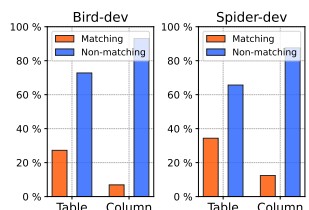

Figure 4: The imbalanced proportion of matching schema elements with the non-matching ones.

---

**Algorithm 1:** Hierarchical KaSLA Strategy

**Input:** user query $q$, schema $\boldsymbol{S}$

**Output:** schema linking results $\widehat{\boldsymbol{S}}_{\text{KaSLA}}$, including a table linking set $\boldsymbol{T}^*$ and the corresponding column linking set $\boldsymbol{C}_t^*$ for $t \in \boldsymbol{T}^*$

$\mathcal{I}_{\boldsymbol{T}} = \left\{ \widehat{I}_{t,q} \leftarrow f_I(t,q) \right\}_{t \in \boldsymbol{T}}, \mathcal{V}_{\boldsymbol{T}} = \left\{ \widehat{V}_{t,q} \leftarrow f_V(t,q) \right\}_{t \in \boldsymbol{T}}, \mathcal{W}_{\boldsymbol{T}} = \left\{ \widehat{W}_{t,q} \leftarrow f_W(t,q)) \right\}_{t \in \boldsymbol{T}}$

$\widehat{C}_q^t \leftarrow f_C(q)$ for table linking, $\widehat{C}_{q,t}^c \leftarrow f_C(q)$ for column linking

$\boldsymbol{T}^* \leftarrow \text{KDP}(\mathcal{V}_{\boldsymbol{T}}, \mathcal{W}_{\boldsymbol{T}}, \widehat{C}_q^t)$             // We present KDP in Algorithm 2

**for** $t \in \boldsymbol{T}^*$ **do**

    $\mathcal{I}_{\boldsymbol{C}_t} = \left\{ \widehat{I}_{c,q,t} \leftarrow f_I(c,q) \right\}_{c \in \boldsymbol{C}_t}, \mathcal{V}_{\boldsymbol{C}_t} = \left\{ \widehat{V}_{c,q,t} \leftarrow f_V(c,q) \right\}_{c \in \boldsymbol{C}_t},$

    $\mathcal{W}_{\boldsymbol{C}_t} = \left\{ \widehat{W}_{c,q,t} \leftarrow f_W(c,q) \right\}_{c \in \boldsymbol{C}_t}$

    $\boldsymbol{C}_t^* \leftarrow \text{KDP}(\mathcal{V}_{\boldsymbol{C}_t}, \mathcal{W}_{\boldsymbol{C}_t}, \widehat{C}_{q,t}^c)$

**return** $\widehat{\boldsymbol{S}}_{KaSLA} = \boldsymbol{T}^* \cup \bigcup_{t \in \boldsymbol{T}^*} \boldsymbol{C}_t^*$

---

## D IMPLEMENTATION DETAILS.

### D.1 TRAINING AND INFERENCE PROCESS

We implemented the proposed KaSLA and baselines under the following settings: *(i) Close-source LLM-based experiments:* We conducted all in-context learning experiments on GPT-4-turbo. For baselines with publicly available SQL generation results, we directly evaluated them using our evaluation settings. *(ii) Open-source LLM-based experiments:* We utilized LoRA (Hu et al., 2021) for parameter-efficient fine-tuning. In KaSLA, we utilize StarCoder2-15B (Lozhkov et al., 2024) as the base model and use two LoRA networks: one for the nomination scoring model in the importance evaluation component and one for the text-to-SQL model in the final SQL generation. The learning rate was initialized to $1e-4$ with a cosine decay, the batch size was set to 16, and the training epoch was 3. We used the same settings for all fine-tuning baselines. *(iii) Open-source LM-based experiments:* We fine-tuned a RoBERTa-Large (Liu et al., 2019) following (Li et al., 2024b) for the guaranteed scoring model used in the importance prediction component.

For the knapsack factors prediction, we set the following parameters: For top-$K_r$ used in the filter of the recall results, we set it to 5 for tables and 6 for columns, as recommended by Li et al. (2024b). For top-$K_C$ used in demonstration conduction, we set it to 30. We set $\alpha = 1$, $\tau = 0.5$, and $\gamma = 1$ in the prediction of value, weight, and capacity, respectively.

### D.2 PROCEDURE OF KASLA

We present the complete procedure of hierarchical KaSLA strategy in Algorithm 1 and illustrate the details of the dynamic programming algorithm used in KaSLA in Algorithm 2.

---

**Algorithm 2:** Dynamic Programming-based 0-1 Knapsack Problem Optimization

---

**Input:** value set $\mathcal{V}$, corresponding weight set $\mathcal{W}$, capacity $\mathcal{C}$
**Output:** selection set $\hat{\mathcal{H}}$
$n \leftarrow |\mathcal{V}|, \hat{\mathcal{H}} \leftarrow \{\}$
$\mathcal{A} \leftarrow$ array of $(n+1) \times (\mathcal{C}+1)$ initialized to 0
$keep \leftarrow$ array of $(n+1) \times (\mathcal{C}+1)$ initialized to False
**for** $i \leftarrow 1$ *to* $n$ **do**
    **for** $w \leftarrow 0$ *to* $\mathcal{C}$ **do**
        **if** $\mathcal{W}[i-1] \leq w$ **then**
            **if** $(\mathcal{V}[i-1] + \mathcal{A}[i-1][w - \mathcal{W}[i-1]]) > \mathcal{A}[i-1][w]$ **then**
                $\mathcal{A}[i][w] \leftarrow \mathcal{V}[i-1] + \mathcal{A}[i-1][w - \mathcal{W}[i-1]]$
                $keep[i][w] \leftarrow$ True
            **else**
                $\mathcal{A}[i][w] \leftarrow \mathcal{A}[i-1][w]$
        **else**
            $\mathcal{A}[i][w] \leftarrow \mathcal{A}[i-1][w]$
$k \leftarrow \mathcal{C}$ **for** $i \leftarrow n$ *downto* 1 **do**
    **if** $keep[i][k]$ **then**
        $\hat{\mathcal{H}}.\text{add}(i-1)$
        $k \leftarrow k - \mathcal{W}[i-1]$
**return** $\hat{\mathcal{H}}$

---

### D.3 INPUT FORMATS

We provided the input formats used in our KaSLA of generation with LLMs in Table 6.

## E COMPLEXITY ANALYSIS

The efficiency of KaSLA stems from its hierarchical approach, strategic use of dynamic programming, and inherent parallelism. KaSLA's time complexity is determined by two main phases: table linking

Table 6: Input formats of generation with LLMs

| Type | Input format |
|------|-------------|
| Schema | table department,
columns = [ department_id (int \| primary key\|values: 1), name (text\|values: State, Treasury),
          ranking (int \| values: 1, 2)]
table head,
columns = [ head_id (int \| primary key \| values: 1), name (text \| values: Tiger Woods), age (real \| values: 67.0)]
table management,
columns = [ department_id ( int \| primary key \| values: 2), head_id (int \| primary key \| values: 5),
          temporary_acting (text \| values: Yes)]
foreign keys: management.head_id = head.head_id, management.department_id = department.department_id |
| Question | How many heads of the departments are older than 56? |
| Schema linking | List the relevant columns in each table: |
| SQL generation | Generate SQL to solve the above question: |

| Model | SuperSQL | CodeS | KaSLA |
|-------|----------|-------|-------|
| Original BIRD-dev | 58.60 | 60.30 | **63.75** |
| Presented BIRD-dev | 69.90 | 67.58 | **71.92** |
| non-presented BIRD-dev | 49.35 | 51.01 | **88.01** |

| Model | SuperSQL | CodeS | KaSLA | KaSLA (Transfer) | %Improve |
|-------|----------|-------|-------|------------------|----------|
| BIRD-dev | 58.60 | 60.30 | **63.75** | 63.10 | -1.02% |
| Spider-dev | 87.40 | 84.72 | **88.01** | 87.23 | -0.89% |

Table 7: Execution Accuracy (EX) (%) of text-to-SQL models trained on whole BIRD-train but evaluated on presented BIRD-dev and non-presented BIRD-dev.

Table 8: Execution Accuracy (EX) (%) of KaSLA trained on Spider-train but evaluated on BIRD-dev, and vice versa, for cross-scenario transfer ability evaluation.

and column linking within the linked tables. For a database with $n_t$ tables and a table capacity of $C_t$, the table linking phase has a time complexity of $O(n_t C_t)$. KaSLA then links columns within the selected table. Let $n_c$ and $C_c$ represent the number of columns and the column capacity of a selected table, respectively. The time complexity of the column linking phase is $O(n_c C_c)$. Consequently, the total time complexity of KaSLA is $O(n_t C_t + n_t(n_c C_c))$, indicating that the complexity of KaSLA scales relevant with the database dimensions.

# F TRANSFERABILITY OF KaSLA

In real-world scenarios, databases often have different backgrounds and come from various domains, leading to diverse database contents and user queries. This diversity poses challenges to the transferability of schema linking models, which are typically pre-trained on public datasets. We conducted two experiments to evaluate KaSLA's performance in cross-scenario transfer.

We first selected databases from the BIRD dataset's dev set with no similar scenarios to those in the training dataset. We created a non-represented BIRD-dev dataset using the selected databases whose background knowledge differs from the training dataset. The remaining data was treated as the presented BIRD-dev dataset. The non-presented dataset helps in assessing the model's transfer capability. Since there are only two importance scoring models of KaSLA that need to be trained, we pre-trained them on the whole BIRD-train data and evaluated the SQL generation performance on the non-presented dataset and the presented dataset with the same SQL generation model. As shown in Table 7, KaSLA demonstrates a clear advantage over baselines in dealing with both presented and non-presented development data.

We also performed transfers between different datasets, such as the Spider and BIRD datasets. We trained the two importance scoring models on the Spider training dataset and evaluated it on the BIRD dev dataset, and vice versa, to explore its cross-scenario transfer ability. The results are provided in Table 8. We can find that pre-training KaSLA on public datasets yields results that outperform the baselines and are only slightly lower than what domain-specific fine-tuning would achieve. The above results show that KaSLA demonstrates strong cross-scenario transferability.

# G    COMPARISON WITH TRADITIONAL METRICS

Traditional metrics, recall, and precision, are also utilized widely in the schema linking evaluation. However, they have drawbacks that hinder their usage in schema linking and cannot reflect the actual schema linking performance.

## G.1    LIMITATIONS OF TRADITIONAL METRICS

Recall, $R_{recall} = \frac{|\widehat{S}_q \cap S_q|}{|S_q|}$ for each instance, can reflect the degree of an element missing, but such continuous degree can not directly reflect the actual impact of schema linking to SQL generation. Any missing element, whether more or less, will directly result in incorrect SQL generation. However, it may not have a noticeable effect on recall cause recall can only measure a continuous degree instead of a definitive judgment about whether this instance has an element missing or not. For example, assume $\widehat{S}_q = \{t_1, t_2, t_3, t_4\}$ and $S_q = \{t_1, t_2, t_3, t_4, t_5\}$ for table linking. This $\widehat{S}_q$ will result in incorrect SQL generation because $t_5$ is missing. However, $R_{recall}$ will be 80%, which unjustly gives a positive evaluation.

Precision, $R_{precision} = \frac{|\widehat{S}_q \cap S_q|}{|\widehat{S}_q|}$ for each instance, can reflect the degree of redundancy only if there are no missing elements. However, element missing is still common for current schema linking methods. Any missing element will directly result in incorrect SQL generation, but the precision might still give a positive evaluation as long as the number of missing elements is small compared to the total elements in the linking results. For example, assume $\widehat{S}_q = \{t_1, t_2, t_3\}$ and $S_q = \{t_1, t_2, t_3, t_4, t_5\}$ for table linking. With tables $t_4$ and $t_5$ missing, the SQL generation will always be incorrect. However, $R_{precision}$ will be 100%, a highly positive evaluation, which is meaningless.

AUC is also unsuitable because schema linking is an imbalanced classification task. As shown in Figure 4, matching elements are only a small subset of all elements. A schema linking model can achieve a high AUC score by predominantly predicting non-matching elements and ignoring matching elements.

## G.2    ADVANTAGES OF THE PROPOSED METRICS

Compared with Recall, our schema missing rate, $R_{miss}$ in Eq. (2), for each instance, can reflect the actual impact of schema linking results to SQL generation by a strict evaluation. It considers an instance a failure if any element is missing, acknowledging success only when no elements are missing. This aligns with the fact that any missing element leads to incorrect SQL generation.

Compared with Precision, Our schema redundancy rate, $R_{redun}$ in Eq. (3), can provide a meaningful evaluation by only calculating redundancy only when all matching elements are present in the linking results. The main difference between precision and our redundancy rate is the Non-missing indicator. The prediction with element missing will be judged as failure because, in such cases, redundancy is not the sole reason for incorrect SQL generation, thus making its evaluation meaningless.

# H    ABLATION STUDY

We presented the ablation study results for the importance score $I$ with the nomination score $\mathcal{I}_{nomi}$ and the guaranteed score $\mathcal{I}_{guar}$ in Table 9, the results concerning the choice of language model for the guaranteed scoring model are shown in Table 10, and the results for the hyperparameters $\alpha$, $\tau$, and $\gamma$ are provided in Table 11.

In Table 9, we observe that using only the soft score $\mathcal{I}_{guar}$ without the binary score $\mathcal{I}_{nomi}$ results in sub-optimal performance. This occurs because the RoBERTa-Large-based guaranteed model provides only a general representation, which may lack clear differentiation, potentially causing confusion that the StarCoder2-based nomination model addresses. Additionally, relying solely on the binary score $\mathcal{I}_{nomi}$ also leads to sub-optimal outcomes. The Starcoder2-based nomination model tends to confirm elements with high confidence, resulting in distinct scoring differences and possibly missing elements that match with medium confidence. This issue is addressed by the RoBERTa-Large-based

guaranteed model. These findings emphasize the importance of combining the binary score $\mathcal{I}_{nomi}$ and the soft score $\mathcal{I}_{guar}$, allowing them to collaborate for accurate importance score estimation.

In Table 10, we compared various models and sizes for the guaranteed scoring model in SQL generation and found that RoBERTa-Large produced the best results on both datasets. RoBERTa enhances BERT by employing a more comprehensive training process, leading to improved text representation abilities (Min et al., 2023). This allows RoBERTa-Large to provide better generalization and understanding of language nuances, which are crucial for schema linking tasks where capturing subtle semantic relationships is important. Moreover, using fewer parameters (BERT-base or RoBERTa-base) or more parameters (SGPT-1.3B) did not lead to performance improvements, indicating that RoBERTa-Large is an excellent choice for the guaranteed scoring model.

Regarding the range of hyperparameters, $\alpha$, $\tau$, and $\gamma$, Table 11 shows that these hyperparameters only slightly affect KaSLA's performance, yet they offer the potential for optimal results. This highlights the generalization ability of KaSLA.

| Dataset | origin $\mathcal{I}_{nomi}$&$\mathcal{I}_{guar}$ | w/o $\mathcal{I}_{guar}$ | w/o $\mathcal{I}_{nomi}$ |
|---|---|---|---|
| BIRD-dev | **63.75** | 53.13 | 57.63 |
| Spider-dev | **88.01** | 86.36 | 85.40 |

Table 9: Execution Accuracy (EX) (%) of the ablation study about the importance score $I$ with $\mathcal{I}_{nomi}$ and $\mathcal{I}_{guar}$.

| Dataset | BERT -Base | BERT -Large | RoBERTa -Base | RoBERTa -Large | SGPT -1.3B |
|---|---|---|---|---|---|
| BIRD-dev | 61.67 | 61.92 | 62.78 | **63.75** | **63.75** |
| Spider-dev | 86.94 | 87.33 | 87.52 | **88.01** | 87.72 |

Table 10: Execution Accuracy (EX) (%) of the ablation study about the choice of language model for guaranteed scoring model.

Table 11: Execution Accuracy (EX) (%) of the ablation study about the hyperparameters $\alpha$, $\tau$, and $\gamma$.

| $\tau = 0.5, \gamma = 1$ | $\alpha = 0.1$ | $\alpha = 0.5$ | $\alpha = 1$ | $\alpha = 5$ |
|---|---|---|---|---|
| BIRD-dev | 63.10 | 63.69 | **63.75** | 61.15 |
| Spider-dev | 87.72 | **88.10** | 88.01 | 86.27 |
| $\alpha = 1, \gamma = 1$ | $\tau = 0.1$ | $\tau = 0.5$ | $\tau = 0.7$ | $\tau = 0.9$ |
| BIRD-dev | 63.10 | **63.75** | 63.43 | 63.10 |
| Spider-dev | 87.43 | **88.01** | 87.81 | 87.52 |
| $\alpha = 1, \tau = 0.5$ | $\gamma = 0.5$ | $\gamma = 1$ | $\gamma = 1.5$ | $\gamma = 2$ |
| BIRD-dev | 49.02 | **63.75** | 63.10 | 62.84 |
| Spider-dev | 57.25 | **88.01** | 87.72 | 87.14 |

# I    SCHEMA LINKING BENCHMARK ON SPIDER-DEV

We provide the whole schema linking evaluation results with $R_{miss}$, $R_{redun}$ and $R_{correct}$ on BIRD-dev and Spider-dev. Similarly, KaSLA scores a high $R_{correct}$ for column linking on the Spider-dev dataset, outperforming all other methods by a considerable margin.

Table 12: Schema linking benchmarks with $R_{miss}$, $R_{redun}$ and $R_{correct}$ on Spider-dev.

| Dataset | Metric | | Schema linking Method | | | | | | | | | **KaSLA** |
|---|---|---|---|---|---|---|---|---|---|---|---|---|
| | | | CodeS-SL | | 10shot ICL-SL | | DIN-SL | | DTS-SL | Pure-SL | TA-SL | **(Ours)** |
| | | | RoBERTa | SGPT | GPT-3.5 | GPT-4 | GPT-3.5 | GPT-4 | StarCoder2-15B | | | |
| | | | Base | Large | -1.3B | | | | | | | | |
| Spider-dev | $\mathcal{R}_{miss} \downarrow$ | table | 0.00 | **0.00** | 29.00 | 33.57 | 31.23 | 49.61 | 36.83 | 6.29 | 6.29 | 2.42 | 1.26 |
| | | colmn | 0.81 | **0.30** | 2.12 | 93.84 | 91.49 | 96.18 | 93.64 | 6.55 | 15.62 | 9.07 | 2.12 |
| | $\mathcal{R}_{redun} \downarrow$ | table | 55.96 | 55.96 | 56.13 | 42.81 | 35.74 | 52.31 | 39.78 | 9.90 | 9.90 | **4.73** | 6.41 |
| | | colmn | 80.77 | 80.55 | 80.97 | 95.29 | 92.18 | 96.49 | 94.13 | 69.50 | 19.23 | **11.24** | 12.44 |
| | $\mathcal{R}_{correct} \uparrow$ | table | 72.02 | 72.02 | 71.79 | 61.81 | 66.52 | 49.04 | 61.69 | 91.91 | 91.91 | **96.43** | 96.17 |
| | | colmn | 59.21 | 59.57 | 58.46 | 5.44 | 8.16 | 3.66 | 6.12 | 61.97 | 82.57 | 89.84 | **92.72** |

