# OpenReview forum: "Knapsack Schema Linking Agent for LLM-Based Text-to-SQL Generation"
_ICLR.cc/2025/Conference — Submitted to ICLR 2025_

### Official Review · Reviewer_QByZ · 2024-11-04

**Soundness:** 4
**Presentation:** 2
**Contribution:** 3
**Rating:** 6
**Confidence:** 3

**Summary:**

The work focuses on text-to-SQL query conversion. While current research primarily addresses SQL generation, this study is more inclined toward schema-linking tasks. This approach aims to prevent missing or redundant schema linking, resulting in more optimal SQL generation. The authors argue that schema selection is not a simple task but rather resembles a knapsack problem. Additionally, the authors note that the current state-of-the-art (SOTA) in Text-to-SQL relies on simplistic schema linking strategies.

**Strengths:**

This work introduces a novel approach by applying the Knapsack problem to schema linking, effectively addressing the generative versus recall methodologies. The authors highlight the lack of benchmarks and metrics for schema linking, emphasising the need for standard evaluation metrics to assess missing and redundant schemas. They clearly explain why previous metrics are not suitable for evaluating schema linking.

The authors design two tailored schema linking benchmarks and introduce two schema linking metrics: schema missing rate and schema redundancy rate to evaluate schema linking performance.

**Weaknesses:**

- Since this approach is based on dynamic programming, the computation time might be significant. It would be beneficial if the authors addressed this issue in the paper.

- The related work is severely lacking and is in the supplementary section instead of the main paper. The authors make it easy to understand what they have done but it is not clear what the previous approaches were and how exactly the proposed approach is better.

**Questions:**

- In the Guaranteed Scoring Model, the authors used the RoBERTa-Large model to obtain semantic embeddings. It would be helpful if they provided a rationale for choosing this specific model.

---

> ### Author Response · Authors · 2024-11-24
> **Response to W1**
>
> Dear reviewer QByZ,
>
> Thanks very much for your insightful suggestions and concerns. We have provided a detailed response to each concern and made improvements in the revised vision.
>
> > W1: Since this approach is based on dynamic programming, the computation time might be significant. It would be beneficial if the authors addressed this issue in the paper.
>
> Thanks for your comments. The complexity of the dynamic programming (DP) process is very low since we utilize a hierarchical structure to select the schemas. Additionally, the running time analysis shows that the computational time of the DP process is minimal.
>
> - **KaSLA utilizes a hierarchical structure to reduce the time complexity:** The time complexity of the dynamic programming component in KaSLA is $O(n _t C _t + n _t(n _c C _c))$. This relates to the database's table number $n _t$, column number $n _c$ of the table, and the estimated table capacity $C _t$, and column capacity $C _c$. For a traditional 0-1 knapsack problem with $n$ items and a capacity of $C$, its time complexity is $O(nC)$ [1]. In our hierarchical KaSLA strategy, we first optimize the table linking and then optimize the column linking within the linked tables. Thus, the total time complexity is $O(n _t C _t) + n _t O((n _c C _c)) = O(n _t C _t + n _t(n _c C _c))$.
> - **The actual time complexity $O(n _t C _t + n _t(n _c C _c))$ is small because the $n _t, n _t, C _t, C _c$ are usually small.** We averaged these values for BIRD-dev and Spider-dev and displayed them in the table below. The results show that the actual time complexity is small, making our dynamic programming component time-efficient.
>
> *Table W1.a Average schema element number and average capacity of BIRD-dev and Spider-dev*
>
> |            | Average table number $n_t$ | Average column number $n_c$ | Average table capacity $C_t$ | Average column capacity $C_c$ |
> | ---------- | -------------------------- | --------------------------- | ---------------------------- | ----------------------------- |
> | BIRD-dev   | 7.08                       | 75.56                       | 12.58                        | 21.54                         |
> | Spider-dev | 4.41                       | 24.55                       | 6.44                         | 7.09                          |
>
> - Inspired by your question, to understand the tradeoff between KaSLA's time consumption and performance, we provide the inference time on one 80G A100 GPU for the whole framework, both with and without KaSLA, in the table below. we found that the **dynamic programming only takes less than 0.01 seconds to process one instance** in both BIRD-dev and Spider-dev.
>
> *Table W1.b Inference time cost of each component in KaSLA*
>
> | Inference time  / instance | Schema  linking model | Nomination model | Guaranteed model | Factor estimation and dynamic programming | Text-to-SQL |
> | -------------------------- | --------------------- | ---------------- | ---------------- | ----------------------------------------- | ----------- |
> | BIRD-dev                   | Full schema           | \                | \                | \                                         | 14.35 s     |
> |                            | KaSLA                 | 12.87 s          | 0.12 s           | < 0.01 s                                  | 8.80 s      |
> | Spider-dev                 | Full schema           | \                | \                | \                                         | 9.69 s      |
> |                            | KaSLA                 | 9.17 s           | 0.06 s           | < 0.01 s                                  | 7.89 s      |
>
> We also provide the SQL generation accuracy for text-to-SQL, both with and without KaSLA, as follows:
>
> *Table W1.c Text-to-SQL accuracy with full Schema and KaSLA*
>
> | Execution Accuracy (EX) | Text-to-SQL with full schema | Text-to-SQL with KaSLA |
> | ----------------------- | ---------------------------- | ---------------------- |
> | BIRD-dev                | 55.54%                       | 63.75%                 |
> | Spider-dev              | 81.91%                       | 88.01%                 |
>
> As shown in these tables, we can conclude that:
>
> - **KaSLA introduces a significant improvement with a marginal inference delay.** This delay occurs because we use StarCoder2-15B to generate nomination scores for the full schema, aiming for better semantic understanding, which results in additional time consumption. Reducing the inference time for LLM-based nomination models during full schema processing might be a potential research direction.
>
> [1] https://en.wikipedia.org/wiki/Knapsack_problem

---

> ### Author Response · Authors · 2024-11-24
> **Response to W2**
>
> > W2: The related work is severely lacking and is in the supplementary section instead of the main paper. The authors make it easy to understand what they have done but it is not clear what the previous approaches were and how exactly the proposed approach is better.
>
> Thanks for your comments. In the revised manuscript, we have reorganize our paper to provide a more clear presentation of the proposed approach.
>
> - **We have added a new paragraph in the related work section of the paper to primarily discuss schema linking, which has been updated in the revision.** In this paragraph, we examine the current progress in schema linking, including several recent works [2-6]. This discussion helps clarify recent developments in schema linking, highlight the challenges, and explain the impact of our work.
> - **The main impact of our KaSLA compared to baselines is that it addresses both element missing and element redundancy.** Recent schema linking work often seeks to enhance performance by using in-context learning to boost reasoning [2, 6], employing larger LLMs [4], or augmenting contextual information [3, 5]. However, the current work does not emphasize the issues of missing elements and redundancy. Our KaSLA models schema linking as a knapsack optimization problem to reduce redundancy while ensuring no missing elements, better meeting the needs of SQL generation.
>
> We have updated our related work section. However, due to page limitations, we may currently have to place the related work section in the appendix.
>
> [2] Chess: Contextual harnessing for efficient sql synthesis
>
> [3] E-SQL: Direct Schema Linking via Question Enrichment in Text-to-SQL
>
> [4] The Death of Schema Linking? Text-to-SQL in the Age of Well-Reasoned Language Models
>
> [5] RSL-SQL: Robust Schema Linking in Text-to-SQL Generation
>
> [6] CHASE-SQL: Multi-Path Reasoning and Preference Optimized Candidate Selection in Text-to-SQL

---

> ### Author Response · Authors · 2024-11-24
> **Response to Q1**
>
> > Q1: In the Guaranteed Scoring Model, the authors used the RoBERTa-Large model to obtain semantic embeddings. It would be helpful if they provided a rationale for choosing this specific model.
>
> Thanks for raising this important discussion about the choice of language model for guaranteed scoring model. We address this from two perspectives:
>
> - **RoBERTa-Large offers better generalization and understanding of language nuances, which are crucial for schema linking tasks where capturing subtle semantic relationships is important.** RoBERTa improves upon BERT by employing an enhanced and more extensive training process, resulting in improved text representation ability. In the field of LLMs, RoBERTa is also involved in many works that augment LLMs in various downstream tasks [7].
> - **RoBERTa-Large achieves the best performance compared to other language models in our setting.** We compared different models and sizes of the guaranteed scoring model on SQL generation and found that RoBERTa-Large produced the best results on both datasets. Additionally, using fewer parameters (BERT-base or RoBERTa-base) or more parameters (SGPT-1.3B) did not lead to performance improvements. This indicates that RoBERTa-Large is a good choice for the guaranteed scoring model.
>
> | Execution Accuracy (EX) | BERT-Base | BERT-Large | RoBERTa-Base | RoBERTa-Large | SGPT-1.3B |
> | ----------------------- | --------- | ---------- | ------------ | ------------- | --------- |
> | BIRD-dev                | 61.67%    | 61.92%     | 62.78%       | **63.75%**        | **63.75%**    |
> | Spider-dev              | 86.94%    | 87.33%     | 87.52%       | **88.01%**        | 87.72     |
>
> [7] Recent Advances in Natural Language Processing via Large Pre-Trained Language Models: A Survey

---

> ### Author Response · Authors · 2024-11-24
>
> Thanks for your insightful suggestions and questions. We have addressed all additional discussions in our revision. We are eager to continue the discussion if you have further questions or concerns.

---

> ### Author Response · Authors · 2024-12-01
> **Looking forward to your constructive feedback**
>
> Dear reviewer QByZ,
>
> Thanks again for taking the time to review our paper and for providing valuable suggestions and concerns. We have carefully responded to each of your questions. Following your suggestion, we have provided a time complexity analysis of the dynamic programming component, explained the choice of RoBERTa-Large along with the corresponding ablation study, and updated the discussion on related schema linking work in the revision. We hope these responses can address your concerns and thanks for your support.
>
> As the author-reviewer discussion deadline approaches, we would be very grateful if you could share your valuable thoughts with us. If you have any further concerns, we look forward to further discussion with you.

---

> > ### Comment · Reviewer_QByZ · 2024-12-02
> >
> > Thank you for your comments and clarifications. I have updated my soundness scores accordingly.

---

> > > ### Author Response · Authors · 2024-12-02
> > > **Thanks for your reply**
> > >
> > > Thanks for your thoughtful review and feedback! We sincerely appreciate your valuable insights and are grateful for the opportunity to address your concerns. Thank you for your support of our paper.

---

### Official Review · Reviewer_dUxV · 2024-11-04

**Soundness:** 3
**Presentation:** 2
**Contribution:** 2
**Rating:** 5
**Confidence:** 4

**Summary:**

The authors point out, with supporting evidence, that inability to link to a high-recall and high-precision set of schema elements (such as tables and their columns) is a significant contributor to text2sql system errors.  This set of schema elements are provided to the (usually LLM-based) SQL generator/decoder, along with the natural language question/utterance. Loss of recall will generally result in hallucination (where the generated SQL mentions fictitious tables and columns), and loss of precision shows up as interference (leading to existent but incorrect tables and columns being used). The authors then propose a schema retrival method in which recall and precision are deliberately balanced using a knapsack paradigm.  Limited knapsack capacity encourages high precision.  Item (schema element) profits encourage high recall.  The profit of an item is an additive combination of two parts: one is the probability of generation by a fine-tuned LLM, and the other is a late-interaction comparison between the query embedding and a suitable encoding of schema elements.  Meanwhile, the weight of an item is inversely related to an "importance score".  Finally, the knapsack capacity is determined via in-context demonstrations.

**Strengths:**

Strengths:
* Identifies schema retrieval failure as a significant reason for text2sql failures.
* Proposes a schema subset selection problem as a cost-benefit driven knapsack instance.
* Shows consistent end-to-end text2sql improvements on BIRD and Spider.
* Shows a more mixed improvement at schema selection.

**Weaknesses:**

Weaknesses:
* Proposed method has too many moving parts and lacks clarity in its development.
* Writeup quality can be greatly improved (see below).  Nonstandard and unfamiliar terminology is frequently used.
* Instead of inventing new retrieval measures, why not not stick to classic ones like recall and precision?  I do not quite buy that you have to "propose two metrics that evaluate the missing rate and the redundancy rate that directly impact the SQL generation performance, rather than indirect metrics like AUC and F1"  Recall and precision directly impact SQL generation as much as the measures you propose.
* While some improvements result, Knapsack may not be technically the correct paradigm to model schema selection.
* (L364-L373) are confusing.  Why should the SQL generator get access to "ground truth schema $S$" at all?  You would not be able to use it in a real deployment, would you?

**Questions:**

Questions/comments

https://bird-bench.github.io/ shows systems with better performance, but perhaps the authors are interested in a more controlled comparison?  They should explain their position in this matter.

Many writing nits:

What is described as "schema" throughout the paper is more accurately called "schema element", which is usually a table (including views) or a (table, column) pair in case of text2sql.

I do not find it appropriate or necessary to call the proposed method an "agent".

"Nomination-guaranteed" is not a thing.  You cannot use this in an abstract (or anywhere, actually).

Various other terms are used throughout in confusing manners:
redundancy, "semantically related schema",

(L307-L315) Why is $\gamma$ needed as a hyperparameter?

L017 …but can be modeled as a Knapsack…
L019 "redundant" or "incorrect", "excessive"?
L023 "nomination-guaranteed" is non-standard.  Define or use a more self-contained expression.
L025 better to specify early that "capacity" is the maximum item weight the knapsack can support.  This is found on L139 "weight capacity".

Figure 2 What are the schema items $S_1,\ldots,S_4$?  Better to call them "schema items" or "schema elements" and give actual examples of tables and columns.

One can measure a large value of your redundancy measure, but it is not clear how much that damages SQL generation (compared to loss of schema recall).  This is worth a separate study.

Eqn (2) is a measurement of loss of recall.
"Redundancy rate" in eqn (3) is like one minus precision.
It is confusing that redundancy is forced to zero if the recall is not perfect.

The terminology in "nomination-guaranteed" gets even more confusing.

About $\mathcal{I}_ {nomi}(s|q,\mathcal{S})$:
while you have defined in eqn (5) the quantity $\mathcal{L}_ {nomi}$, I could not find an expression for $\mathcal{I}_ {nomi}(s|q,\mathcal{S})$ nearby.  It seems there is no use of this score downstream.  It is just an indicator for what schema elements StarCoder2 will generate.
Authors need to provide exact input and output specifications for finetuning StarCoder2.
Also in eqn (5) how do you deal with the fact that $\mathbf{S}$ may be output in any order?

And then, in L247, an _importance value_ assigned by RoBERTa to each (surviving?) schema element.

L271 "schemas with higher importance scores should be assigned higher values".  Then we see $I_{s,q}$ which comes from eqn (4), combining inclusion by StarCoder2 and scoring by RoBERTa.  This is confusing, because on the latter was called "imporance" earlier.

L281 "schemas with higher importance should be assigned lower weights than those with lower importance scores".  So the importance score figures twice: once in value, and again in weights.  This seems not-elegant.

L488 $\mathcal{R}_{correct}$ springs another surprise.  Given the relatedness to recall and precision, why not just use F1?

Success in Table 3 is more mixed.

---

> ### Author Response · Authors · 2024-11-24
> **Responses to W1, W2**
>
> Dear reviewer dUxV,
>
> Thanks very much for these insightful suggestions and concerns. We will respond to these suggestions and concerns in detail one by one.
>
> > W1 Proposed method has too many moving parts and lacks clarity in its development.
>
> Thanks for highlighting this concern. We will explain the motivation for formulating schema linking as a knapsack problem and the development of KaSLA.
>
> - **Schema linking requires no missing elements and low redundant elements.** Any missing schema element can cause SQL generation to fail directly, while redundant elements can confuse the SQL generation process and lead to incorrect outputs. As shown in the table, missing tables or columns and adding redundant ones significantly negatively impact text-to-SQL performance.
>
> *Table W1.a Randomly drop 1, 2, and 3 columns from all instances' schema linking results.*
>
> | Execution Accuracy (EX) | Drop 3 column | Drop 2 column | Drop 1 column | Origin KaSLA |
> | ----------------------- | ---------------------- | ---------------------- | ---------------------- | ------------ |
> | BIRD-dev                | 27.57%                 | 33.38%                 | 46.09%                 | **63.75%**   |
>
> *Table W1.b Randomly add 10%, 20%, and 30% redundant columns into all instances' schema linking results.*
>
> | Execution Accuracy (EX) | **Add 30% redundanct columns** | **Add 20% redundanct columns** | **Add 10% redundanct columns** | Origin KaSLA |
> | ----------------------- | --------------------------------------- | --------------------------------------- | --------------------------------------- | ------------ |
> | BIRD-dev                | 56.38%                                  | 58.34%                                  | 60.82%                                  | **63.75%**   |
>
> - **The knapsack problem—maximizing value while minimizing weight within a capacity constraint—aligns well with schema linking by addressing both the missing and redundant requirements.** Therefore, we formulate schema linking as a knapsack problem and propose KaSLA. The components of KaSLA align closely with knapsack optimization, incorporating factors such as value, weight, and capacity, along with a dynamic programming-based hierarchical schema linking mechanism. We apply importance scoring estimation to adapt knapsack optimization for schema linking.
> - The table below shows that KaSLA achieves optimal performance by attaining both the lowest column missing ratio and the lowest column redundancy ratio, demonstrating the effectiveness of the proposed knapsack optimization framework.
>
> *Table W1.c Performance on Bird-dev with execution accuracy (EX) for text-to-SQL evaluation and missing ratio and redundancy ratio  for schema linking evaluation*
>
> |                         | Pure-SL | DTS-SL | TA-SL  | CodeS-SL | KaSLA      |
> | ----------------------- | ------- | ------ | ------ | -------- | ---------- |
> | Column missing ratio    | 57.54%  | 35.77% | 43.80% | 12.53%   | **5.83%**  |
> | Column redundancy ratio | 61.75 % | 83.04% | 46.25% | 80.83%   | **42.24%** |
> | Execution Accuracy (EX) | 50.26%  | 55.22% | 53.13% | 60.30%   | **63.75%** |
>
> - **We have released all the code to facilitate development and reproduction:** the datasets used, the training and inference pipeline for importance scoring models, and the implementation of hierarchical knapsack optimization for schema linking. We have systematically organized the code for better usability. In practical deployment or future research, more focus can be placed on adapting estimation methods for importance scoring and knapsack factors.
>
>
>
>
> > W2 Writeup quality can be greatly improved (see below). Nonstandard and unfamiliar terminology is frequently used.
>
> Thanks for the constructive feedback on the quality of the writing and terminology. We have addressed each sub-question in Q2 and incorporated your suggestions into our revision.

---

> ### Author Response · Authors · 2024-11-24
> **Response to W3 (1/3)**
>
> > W3 Instead of inventing new retrieval measures, why not not stick to classic ones like recall and precision? I do not quite buy that you have to "propose two metrics that evaluate the missing rate and the redundancy rate that directly impact the SQL generation performance, rather than indirect metrics like AUC and F1" Recall and precision directly impact SQL generation as much as the measures you propose.
> >
> > Q2.9 Eqn (2) is a measurement of loss of recall. "Redundancy rate" in eqn (3) is like one minus precision. It is confusing that redundancy is forced to zero if the recall is not perfect.
> >
> > Q2.13 L488 Rcorrect springs another surprise. Given the relatedness to recall and precision, why not just use F1?
>
> Thanks for bringing up these insightful discussions about schema linking evaluation. We addressed these concerns by explaining the limitations of classic recall and precision and the advantages of the proposed schema missing rate and schema redundancy rate. We also explained why AUC and F1 are not suitable for schema linking.
>
> **(1) Limitations of precision and recall in Schema**
>
> Precision: $R_{precision}=\frac{1}{|B|} \sum _{(q,S) \in B}\frac{|\hat{S} _q \cap S _q|}{|\hat{S} _q|}$
>
> Recall: $R_{recall} = \frac{1}{|B|} \sum _{(q,S) \in B} \frac{|\hat{S} _q \cap S _q|}{|S _q|}$
>
> ($B$ is the whole dev dataset, $\hat{S}_q$ is the predicted schema linking results, and $S_q$ is the ground truth.)
>
> They are both general evaluation metrics; however, they have drawbacks that hinder their usage in schema linking.
>
> - **The same recall metric values can correspond to different SQL generation accuracies:** In schema linking, missing a single table or column (recall < 100%) can significantly decrease text-to-SQL generation accuracy. Averaging recall cannot indicate how many instances are correct. We provide two examples in the table below.
> - **As shown in the table**, in example-1, the average recall is 80%. However, since 3 out of 5 instances have missing elements, only 2 instances can generate correct SQL. In example-2, the average recall is also 80%, but with missing elements in every instance, not a single instance can generate correct SQL. This demonstrates that average recall does not accurately represent actual schema linking results.
>
> *Table W3.a  Two examples of using recall as a column linking metric for 5 instances—high average recall does not equal high correct linking*
>
> | Recall    | ${Recall} _{1}$ | ${Recall} _{2}$ | ${Recall} _{3}$ | ${Recall} _{4}$ | ${Recall} _{5}$ | Average recall | Correct instance |
> | --------- | --------------- | --------------- | --------------- | --------------- | --------------- | -------------- | ---------------- |
> | Example-1 | 100%            | 100%            | 66.7%           | 66.7%           | 66.7%           | 80%            | 2                |
> | Example-2 | 80%             | 80%             | 80%             | 80%             | 80%             | 80%            | 0                |
>
> - **Higher precision metric values may result in lower SQL generation accuracy:** Precision is not decisive in schema linking tasks. For instance, if the precision is 100% but the recall is below 100%, there will be 0 correct SQL generation. We provide two examples in the table below.
> - **As seen in the table**, in example-3, all 5 instances have no redundant elements in their linking results, yet all instances still have missing elements. Precision is 100%, but no correct SQL can be generated. In example-4, there are no missing elements, but redundancy exists; therefore, precision is low, but all instances can potentially generate correct SQL. This indicates that precision does not clearly correlate with the success of schema linking results.
>
> *Table W3.b  Two examples of using precision and recall as column linking metrics for 5 instances—precision cannot reflect linking results accurately*
>
> | (Precision, Recall) | (${Precision} _{1}$, ${Recall} _{1}$) | (${Precision} _{2}$, ${Recall} _{2}$) | (${Precision} _{3}$, ${Recall} _{3}$) | (${Precision} _{4}$, ${Recall} _{4}$) | (${Precision} _{5}$, ${Recall} _{5}$) | Average precision | Correct instance |
> | ------------------- | ------------------------------------- | ------------------------------------- | ------------------------------------- | ------------------------------------- | ------------------------------------- | ----------------- | ---------------- |
> | Example-3           | (100%, 80%)                           | (100%, 80%)                           | (100%, 80%)                           | (100%, 80%)                           | (100%, 80%)                           | 100%              | 0                |
> | Example-4           | (20%, 100%)                           | (20%, 100%)                           | (20%, 100%)                           | (20%, 100%)                           | (20%, 100%)                           | 20%               | 5                |

---

> ### Author Response · Authors · 2024-11-24
> **Response to W3 (2/3)**
>
> **(2) Necessary to propose missing ratio and redundancy ratio**
>
> Schema missing ratio: $R _{miss} =\frac{1}{|B|} \sum _{(q,S) \in B} \unicode{x1D7D9}(S _q \nsubseteq \widehat{S} _q)$
>
> schema redundancy ratio: $R _{redun} =\frac{1}{|B|} \sum _{(q,S) \in B}  \underbrace{{(\frac{|\hat{S} _q \setminus S _q|}{|\hat{S} _q|})}^{\sigma}} _{\text{redundancy ratio}}, \underbrace{\sigma = \unicode{x1D7D9}(S _q \subseteq \widehat{S} _q)} _{\text{Non-missing Indicator}}$
>
> Due to the above drawbacks, precision and recall cannot be used in schema linking evaluation. Then we propose the missing and redundancy ratios to evaluate the schema linking accurately.
>
> - **Our schema missing ratio can reflect the actual impact of schema linking results to SQL generation by a strict evaluation.** It considers an instance a failure if any element is missing, acknowledging success only when no elements are missing. This aligns with the fact that any missing element leads to incorrect SQL generation.
> - **In Table W3.a,** the two examples show a missing ratio of 60% and 100% (lower is better), accurately reflecting the difficulty in generating correct SQL from these schema linking results.
> - **Lower missing ratio will have higher SQL generation accuracy**: The table below shows the column missing ratio and SQL generation accuracy using the same text-to-SQL model with different schema linking models. The results demonstrate that the missing ratio accurately reflects the accuracy of schema linking and directly affects SQL generation. KaSLA achieves the highest SQL generation accuracy with the lowest missing ratio.
>
> *Table W3.c Performance on Bird-dev with execution accuracy (EX) for text-to-SQL evaluation and missing ratio for schema linking evaluation. The SQL generation model is a fine-tuned StarCoder2-15B*
>
> |                         | Pure-SL | DTS-SL | TA-SL  | CodeS-SL | KaSLA      |
> | ----------------------- | ------- | ------ | ------ | -------- | ---------- |
> | Column missing ratio    | 57.54%  | 35.77% | 43.80% | 12.53%   | **5.83%**  |
> | Execution Accuracy (EX) | 50.26%  | 55.22% | 53.13% | 60.30%   | **63.75%** |
>
> - **Our schema redundancy ratio can provide a meaningful evaluation by calculating redundancy only when all matching elements are present in the linking results.** The main difference between precision and our redundancy ratio is that the prediction with an element missing will be judged as a failure because, in such cases, redundancy is not the sole reason for incorrect SQL generation, thus making its evaluation meaningless.
> - **In Table W4.b3**, the two examples show redundancy ratios of 100% and 20% (lower is better), correctly reflecting that the first schema linking result struggles to generate correct SQL while the second one can.
> - **Lower redundancy and missing ratios will have higher SQL generation accuracy.** Using the same experimental setup, the table below shows the column redundancy ratio and corresponding SQL generation accuracy. The results in Tables W4.c and W4.d indicate that only KaSLA achieves the lowest redundancy and missing ratios, resulting in the highest accuracy of SQL generation.
>
> *Table W3.d Performance on Bird-dev with execution accuracy (EX) for text-to-SQL evaluation and missing ratio for schema linking evaluation. The SQL generation model is a fine-tuned StarCoder2-15B*
>
> |                         | Pure-SL | DTS-SL | TA-SL  | CodeS-SL | KaSLA      |
> | ----------------------- | ------- | ------ | ------ | -------- | ---------- |
> | Column redundancy ratio | 61.75 % | 83.04% | 46.25% | 80.83%   | **42.24%** |
> | Execution Accuracy (EX) | 50.26%  | 55.22% | 53.13% | 60.30%   | **63.75%** |

---

> ### Author Response · Authors · 2024-11-24
> **Response to W3 (3/3)**
>
> **(3)** **The special design of redundancy ratio considering no element missing**
>
> - Redundancy is deemed a failure if elements are missing because a prediction with missing elements is meaningless for redundancy evaluation, as explained in point (2).
>
> **(4) Why AUC is not suitable for schema linking**
>
> - **AUC is unsuitable because schema linking is an imbalanced classification task.** As shown in the table below, matching elements are only a small subset of all elements. A schema linking model can achieve a high AUC score by predominantly predicting non-matching elements and ignoring matching elements.
>
> *Table W3.e The average number of all element / matching elements of each instance in BIRD-dev and Spider-dev*
>
> | Average number per instance | All Table | Matching table | All Column | Matching column |
> | --------------------------- | --------- | -------------- | ---------- | --------------- |
> | BIRD-dev                    | 7.08      | 1.93           | 75.56      | 5.21            |
> | Spider-dev                  | 4.41      | 1.51           | 24.55      | 3.06            |
>
> **(5)** **Why use $R_{correct} = 1 - (R_{miss} + R_{redun})/2$ instead of $R_{F1} = \frac{2 R_{recall} R_{precision}}{R_{recall} + R_{precision}}$**
>
> - $R_{correct}$ in our paper is designed to reflect the combined ability of a schema linking model to accurately predict matching elements while reducing redundancy. As shown in Table 3 of our paper, KaSLA achieves both a low missing rate and low redundancy rate, resulting in the highest $R_{correct}$.
> - **We didn't use F1 score, because recall and precision themselves are not suitable metrics**, as explained above in point (1) and point (2).
>
> We have updated the comparation between the traditional metrics with the proposed schema missing rate and schema redundancy rate in the revision.

---

> ### Author Response · Authors · 2024-11-24
> **Responses to W4, W5**
>
> > W4 While some improvements result, Knapsack may not be technically the correct paradigm to model schema selection.
>
> Thanks for raising this important question. We address it from two angles: the motivation for formulating schema linking as a knapsack problem and the necessity of a capacity constraint.
>
> - **Schema linking requires no missing elements and low redundant elements.** Any missing schema element can cause SQL generation to fail directly, while redundant elements can confuse the SQL generation process and lead to incorrect outputs.   As shown in the table, missing tables or columns and adding redundant ones significantly negatively impact text-to-SQL performance.
> - **These requirements align with the knapsack optimization goal of maximizing value while minimizing weight within a capacity constraint.** Therefore, we formulate it as a knapsack problem to address both requirements.
>
> *Table W4.a Randomly drop 1, 2, 3 columns from all instances' schema linking results.*
>
> | Execution Accuracy (EX) | Drop 3 column | Drop 2 column | Drop 1 column | Origin KaSLA |
> | ----------------------- | ---------------------- | ---------------------- | ---------------------- | ------------ |
> | BIRD-dev                | 27.57%                 | 33.38%                 | 46.09%                 | **63.75%**   |
>
> *Table W4.b Randomly add 10%, 20%, and 30% redundant columns into all instances' schema linking results.*
>
> | Execution Accuracy (EX) | **Add 30% redundanct columns** | **Add 20% redundanct columns** | **Add 10% redundanct columns** | Origin KaSLA |
> | ----------------------- | --------------------------------------- | --------------------------------------- | --------------------------------------- | ------------ |
> | BIRD-dev                | 56.38%                                  | 58.34%                                  | 60.82%                                  | **63.75%**   |
>
>
> - KaSLA achieves optimal performance by attaining both the lowest column missing ratio and the lowest column redundancy ratio, demonstrating the effectiveness of the proposed knapsack optimization framework.
>
> *Table W4.c Performance on Bird-dev with execution accuracy (EX) for text-to-SQL evaluation and missing ratio and redundancy ratio  for schema linking evaluation. The SQL generation model is a fine-tuned StarCoder2-15B*
>
> |                         | Pure-SL | DTS-SL | TA-SL  | CodeS-SL | KaSLA      |
> | ----------------------- | ------- | ------ | ------ | -------- | ---------- |
> | Column missing ratio    | 57.54%  | 35.77% | 43.80% | 12.53%   | **5.83%**  |
> | Column redundancy ratio | 61.75 % | 83.04% | 46.25% | 80.83%   | **42.24%** |
> | Execution Accuracy (EX) | 50.26%  | 55.22% | 53.13% | 60.30%   | **63.75%** |
>
> Our paper aims to introduce the idea that schema linking inherently possesses the characteristics of optimization problems, aiming for an ideal goal under constraints, particularly related to the classic and straightforward knapsack framework. Future research may explore alternative combinatorial optimization methods for schema linking.
>
>
> > W5 (L364-L373) are confusing. Why should the SQL generator get access to "ground truth schema S" at all? You would not be able to use it in a real deployment, would you?
>
> Thank you for pointing this out. we have modified this section and updated a revision to address any confusion caused by wording.
>
> - **The ground truth schema $S$ is used only for training.** In lines 364-373 of the submission version, we discuss the training of the SQL generation model, where the input schema elements combine KaSLA’s linking results and the training dataset’s ground truth linking results $S$. This ensures that the inputs contain all matching elements, as even the optimized KaSLA might not achieve 100% correct schema linking.
> - **During inference on the dev dataset (i.e., real deployment), the input schema elements for the SQL generation model are solely KaSLA’s linking results.** No ground truth information from the dev dataset is available. To avoid confusion, we have added the inference process after the training process to explain the KaSLA deployment process more clearly.

---

> ### Author Response · Authors · 2024-11-24
> **Response to Q1**
>
> > Q1 https://bird-bench.github.io/ shows systems with better performance, but perhaps the authors are interested in a more controlled comparison? They should explain their position in this matter.
>
> Thank you for referencing the BIRD benchmark, which highlights some of the best performances in the text-to-SQL field. We appreciate to discuss KaSLA's position concerning these strong performances, and we address this step by step:
>
> - **Most of the top-ranking do not release their code, and we cannot reproduce their results.** For methods where the paper is available but the code is not, we found it difficult to achieve results similar to those reported by relying solely on the details provided in their papers. Therefore, in our experiments, we primarily considered methods that have both released papers and available code.
> - **We compared CHESS (top 6) and E-SQL (top 15) and found that KaSLA can enhance them.** They reported results on BIRD-dev that exceed those of our KaSLA with the StarCoder2-15B-based SQL generation model. Since KaSLA is a plug-and-play schema linking model, we reproduced E-SQL and CHESS and compared their performance with and without KaSLA. As shown in the table below, integrating KaSLA can further enhance them by improving their schema linking, highlighting KaSLA's plug-and-play nature and robust schema linking capability.
> - **Implement details:** CHESS [1] uses a retrieval-augmented chain-of-thought prompting method for schema linking, while E-SQL [2] employs a filtered schema correction strategy. We enhanced their schema linking methods with KaSLA. Since CHESS uses GPT-4 and a fine-tuned DeepSeek Coder model but did not release the latter's checkpoint, we reproduced each component in CHESS using GPT-4. In line with [2], we used GPT-4o as the base LLM for E-SQL.
>
> *Table Q1.a Text-to-SQL performance of integrating KaSLA into SOTA baselines (CHESS and E-SQL)*
>
> | Execution Accuracy (EX) | KaSLA  | CHESS (GPT-4) |  CHESS (GPT-4) +KaSLA       | E-SQL (GPT-4o) |   E-SQL (GPT-4o) +KaSLA       |
> | ----------------------- | ------ | ------------- | ---------- | -------------- | ---------- |
> | BIRD-dev                | 63.75% | 63.10%        | 63.89%     | 65.25%         | 65.91%     |
> | Spider-dev              | 88.01% | 87.14%        | 88.20%     | 88.30%         | 88.68%     |
>
> [1] CHESS: Contextual Harnessing for Efficient SQL Synthesis
>
> [2] E-SQL: Direct Schema Linking via Question Enrichment in Text-to-SQL

---

> ### Author Response · Authors · 2024-11-24
> **Rresponse to Q2 (1/3)**
>
> > Q2 Many writing nits:
>
> - Thanks for pointing out these suggestions to enhance the quality of our paper. We have carefully considered each of your suggestions and updated a revision . Below are the specific changes we have addressed:
>
> > Q2.1 What is described as "schema" throughout the paper is more accurately called "schema element", which is usually a table (including views) or a (table, column) pair in case of text2sql.
>
> - We have made adjustments throughout the paper to ensure that "schema element" is used where appropriate to distinguish it from the full database schema.
>
> > Q2.2 I do not find it appropriate or necessary to call the proposed method an "agent".
>
> - We refer to our model as an "agent" because its plug-and-play nature. Our KaSLA can be integrated into any text-to-SQL framework as a schema linking agent or serve as an enhancement to existing schema linking methods.
>
> > Q2.3 "Nomination-guaranteed" is not a thing. You cannot use this in an abstract (or anywhere, actually).
>
> - Thank you for highlighting this. As you noted, using non-standard terminology in the abstract can cause confusion. We have revised the abstract accordingly.
> - Within the main text, we describe the StarCoder2-based scoring model as a "Nomination model" because it typically generates binary score $ I_{\text{nomi}} $ with high-confidence, akin to nominating select a few candidates from a pool. The RoBERTa-based scoring model is termed the "guaranteed model" since it provides a soft score $ I_{\text{guar}} $ for each element, offering only a general representation.
>
> > Q2.4 (L307-L315) Why is γ needed as a hyperparameter?
>
> - **We employ $ \gamma $ in the estimation function of capacity: $\widehat{C} _q = f _{C}(q) = \gamma \cdot \max {\lbrace \sum _{s \in \text{Knap}(d)} \widehat{W} _{s,q} \rbrace} _{d \in D}$, $\gamma>0$, to adjust the scaling of the estimated capacity.** In practical applications, scaling to a larger capacity can include more potential matching elements in the linking results, but it may also overly relax constraints, leading to redundancy. We use $ \gamma $ to manage this trade-off.
> - We simply apply $ \gamma = 1 $ in our experiments, that is, use the normal capacity scale. we provide the Execution Accuracy (EX) results with $ \gamma = \in \{0.5, 1, 1.5, 2\}$ in table below. The results show that $ \gamma$ can slightly affect KaSLA's performance and offer the potential for the most optimal results. This demonstrates the generalization ability of KaSLA.
>
> *Table Q2.4.a Ablation study of $ \gamma $*
>
> | $ \gamma = \in \{0.5, 1, 1.5, 2 \}$ | 0.5    | 1          | 1.5      | 2      |
> | -------------------------------- | ------ | ---------- | ------ | ------ |
> | BIRD-dev                         | 49.02% | **63.75%** | 63.10% | 62.84% |
> | Spider-dev                       | 57.25% | **88.01%** | 87.72% | 87.14% |
>
>
> > Q2.5 Various other terms are used throughout in confusing manners: redundancy, "semantically related schema".
> >
> > Q2.6 L017 …but can be modeled as a Knapsack… L019 "redundant" or "incorrect", "excessive"? L023 "nomination-guaranteed" is non-standard. Define or use a more self-contained expression.  L025 better to specify early that "capacity" is the maximum item weight the knapsack can support. This is found on L139 "weight capacity".
>
> Thanks for highlighting these terms. We have revised the manuscript to ensure these terms are used consistently and clearly. We have made the following revisions in the revision to prevent any confusion:
>
> - Use "matching element" and "un-matching element" to describe the linking relationship between schema elements and the query. Use "missing element" and "redundant element" to identify errors in the linking results. Remove unclear terms like "semantically related schema," "incorrect," and "excessive."
> - Move the first mention of "capacity" to the start of the method explanation to clarify that "capacity" means the maximum weight the knapsack can hold.
> - Add detailed definitions for the "nomination scoring model" and "guaranteed scoring model" to clear up any confusion caused by these names.
>
>
> > Q2.7 Figure 2 What are the schema items S1,…,S4? Better to call them "schema items" or "schema elements" and give actual examples of tables and columns.
>
> - Thanks for your suggestion. Using "schema elements" instead of "S1,…,S4" indeed helps present our methodology more clearly. We used "S1,…,S4" due to the space constraints in the submission version.
> - Following your recommendation, we have redrawn Figure 2 to use "schema elements" and include actual examples of tables and columns, enriching the figure's informative value.

---

> > ### Comment · Reviewer_dUxV · 2024-11-27
> > **Thanks for taking the trouble to improve the writeup!**
> >
> > I appreciate the effort you have invested to improve the presentation.

---

> > > ### Author Response · Authors · 2024-11-29
> > >
> > > Dear reviewer dUxV,
> > >
> > > Thanks a lot for your valuable feedback and for discussing your concerns with us further. We appreciate your time and your recognition of some of our responses that addressed your concerns, which clarified the limitations of recall and precision, highlighted the advantages of our proposed $R_{miss}$ and $R_{redun}$, and improved our writing by incorporating your suggestions. We appreciate the opportunity to continue this discussion, and thanks for your response and any further questions.
> > >
> > > The knapsack-like selection strategy based on standard recall and precision that you introduced is very interesting and insightful. We will explain it in detail as follows:
> > >
> > > > It would be nice if you could demonstrate that if standard recall and precision were used to guide a knapsack-like selection strategy, as close as possible to your method, they result in suboptimal choices.
> > >
> > > Thank you so much for sharing your insight. We have some detailed thoughts we'd like to discuss with you:
> > >
> > > - **We would like to clarify that $R_{miss}$ and $R_{redun}$ are currently used to evaluate the performance of schema linking and are not directly used to guide the schema linking strategy in KaSLA.** Through $R_{miss}$ and $R_{redun}$, we discovered that the current schema linking model struggles to address both missing and redundancy issues. This inspired us to introduce a capacity constraint into the schema linking process. Consequently, we designed KaSLA, a knapsack optimization-based schema linking model, and validated its effectiveness through experiments.
> > > - **Your suggestion of a metric-guided knapsack-like selection strategy is indeed an intriguing idea. In our opinion, it could be a reinforcement learning framework.** In reinforcement learning, feedback is provided by the environment when the model takes an action, and this feedback can be determined by some related metrics. Modeling schema linking as a reinforcement learning task and using a knapsack-like selection strategy for optimization, with $R_{miss}$ and $R_{redun}$ providing feedback, is an exciting concept. We think it makes sense and could be worth exploring in future research.
> > >   - However, it's important to note that, currently, $R_{miss}$ and $R_{redun}$ can only evaluate schema linking results after inference based on ground truth linking results, which are not available during inference. This presents a challenge for applying $R_{miss}$ and $R_{redun}$, or other metrics, in a reinforcement learning-based knapsack-like schema linking framework as a reward model to provide feedback.
> > >
> > > Thanks for sharing this idea. We believe that this metric-guided knapsack-like schema linking approach is very promising and holds great potential as a future research direction.

---

> > > ### Comment · Reviewer_dUxV · 2024-12-02
> > > **Thanks for engaging in the rebuttal process**
> > >
> > > Your clarifications will significantly improve the quality of writeup, as well as justify many of the design choices you made, so I increased the rating of the presentation. I will hold my overall rating at the same level, based on the perceived technical contribution. All the best!

---

> > > > ### Author Response · Authors · 2024-12-03
> > > > **Thanks for your reply**
> > > >
> > > > Thanks for your thoughtful review and feedback! We sincerely appreciate your valuable insights and are grateful for the opportunity to address your concerns. Thank you for your support of our paper.

---

> ### Author Response · Authors · 2024-11-24
> **Rresponse to Q2 (2/3)**
>
> > Q2.8 One can measure a large value of your redundancy measure, but it is not clear how much that damages SQL generation (compared to loss of schema recall). This is worth a separate study.
>
> - Thank you for suggesting this insightful approach. Investigating how variations in the redundancy measure affect SQL generation accuracy is indeed an enlightening idea. We have conducted related experiments, and the results are as follows. As shown in the table, adding redundant elements significantly negatively impacts text-to-SQL performance and increases the redundancy ratio.
>
> *Table Q2.a Randomly add 10%, 20%, and 30% redundant columns into all instances' schema linking results.*
>
> | BIRD-dev                                 | **Add 30% redundant columns** | **Add 20% redundant columns** | **Add 10% redundant columns** | Origin KaSLA |
> | ---------------------------------------- | ----------------------------- | ----------------------------- | ----------------------------- | ------------ |
> | Execution Accuracy (EX)                  | 56.38%                        | 58.34%                        | 60.82%                        | **63.75%**   |
> | Column Redundancy Ratio (less is better) | 54.32%                        | 48.17%                        | 45.58%                        | 42.24%       |
>
> > Q2.9 Eqn (2) is a measurement of loss of recall. "Redundancy rate" in eqn (3) is like one minus precision. It is confusing that redundancy is forced to zero if the recall is not perfect.
>
> - We addressed this concern in the response of W3, please refer to the above response.
>
> > Q2.10.1 The terminology in "nomination-guaranteed" gets even more confusing. About $I_{\text{nomi}}(s|q,S)$: while you have defined in eqn (5) the quantity Lnomi, I could not find an expression for $I _{\text{nomi}}(s|q,S)$ nearby. It seems there is no use of this score downstream. It is just an indicator for what schema elements StarCoder2 will generate.
>
> - Thanks for your concerns. $I_{nomi}(s \mid q,S) = 1$ if $s \in \widehat{S}_{nomi}$ and $I _{nomi}(s \mid q,S) = 0$ if $s \notin \widehat{S} _{nomi}$, indicating whether each schema element $s$ appears in the generation results $\widehat{S} _{nomi}$ of the StarCoder2-based nomination model. Actually, $I _{\text{nomi}}(s|q,S)$ is used in Eq. (4), $I({s \mid q,S}) = I _{nomi}{(s \mid q,S)} + \alpha I _{guar}{(s \mid q,S)}$, as a part of the important score.
>
> > Q2.10.2 Authors need to provide exact input and output specifications for finetuning StarCoder2.
>
> Below are examples of inputs and outputs for starcoder2-based nomination scoring model:
>
> - Input of starcoder2-based nomination scoring model
>
> ```
> database schema:
> table department, columns = [ department_id (int|primary key|values: 1) , name (text|values: State, Treasury), ranking (int|values: 1, 2)]
> table head ,columns = [ head_id (int|primary key|values: 1) , name (text|values: Tiger Woods) , age (real|values: 67.0)]
> table management, columns = [ department_id ( int|primary key|values: 2), head_id (int|primary key|values: 5), temporary_acting (text|values: Yes)]
> foreign keys: management.head_id = head.head_id, management.department_id = department.department_id
> Question:
> How many heads of the departments are older than 56 ? List the relevant columns in each table:
> ```
>
> - Output of starcoder2-based nomination scoring model
>
> ```
> "Relevant columns": {"department": null, "head": ["age"], "management": null}
> ```
>
> - Turn the output to the nomination score $I_{\text{nomi}}(s|q,S)$:
>
> ```
> {"department.department_id":0, "department.name":0, "department.ranking":0, "head.head_id":0, "head.name":0, "head.age":1, "management.department_id":0, "management.head_id":0, "management.temporary_acting":0}
> ```
>
> Thanks for your suggestions, we have updated the details about the input and output of StarCoder2 in our revision.
>
> > Q2.10.3  Also in eqn (5) how do you deal with the fact that S may be output in any order?
>
> - **Thanks for your question. We handle the fact that $S$ may be output in any order by aligning the order of elements in the linking results with those in the full schema input.** Our approach allows StarCoder2 to scan the full schema from beginning to end, selecting matching elements and placing them in the output in the same order. By aligning the output order with the input order, the absence of a specific order in schema linking does not affect the training process.

---

> ### Author Response · Authors · 2024-11-24
> **Rresponse to Q2 (3/3)**
>
> > Q2.11 And then, in L247, an importance value assigned by RoBERTa to each (surviving?) schema element. L271 "schemas with higher importance scores should be assigned higher values". Then we see Is,q which comes from eqn (4), combining inclusion by StarCoder2 and scoring by RoBERTa. This is confusing, because on the latter was called "imporance" earlier.
>
> We have revised this section in the updated version to clarify any misunderstandings.
>
> - **As shown in Equation 4, our importance score $I_{s,q}$ combines the high confident score from StarCoder2 ($I_{nomi}$ ) and the general score from RoBERTa ($I_{guar}$).** "Higher importance scores" refers to a higher $I_{s,q}$, instead of $I_{nomi}$ or $I_{guar}$.
>
> > Q2.12 L281 "schemas with higher importance should be assigned lower weights than those with lower importance scores". So the importance score figures twice: once in value, and again in weights. This seems not-elegant.
>
> Thank you for your concern. We have reformatted the equations in our paper for improved clarity.
>
> - **For not merging $\hat{V} _{s,q}$, $\hat{W} _{s,q}$ and $I _{s,q}$, the reason is that we aim to treat the importance score estimation and the knapsack factor estimation (value, weight, and capacity) as two separate components.** This makes the entire framework clearer and easier to develop, as the importance scoring model can be trained once and reused anytime.
>
> > Q2.13 L488 Rcorrect springs another surprise. Given the relatedness to recall and precision, why not just use F1?
>
> - We addressed this concern in the response of W3, please refer to the above response.
>
> > Q2.14 Success in Table 3 is more mixed.
>
> - Thanks for your feedback on the presentation of the experimental results. We have revised Table 3 in the updated version to better emphasize the most critical components.

---

> ### Author Response · Authors · 2024-11-24
>
> Thanks for your helpful suggestions and concerns. We've added the additional discussions to our revision. We welcome more discussions on any remaining issues.

---

> ### Comment · Reviewer_dUxV · 2024-11-27
> **Combining $R_{miss}$ and $R_{redun}$ via harmonic mean**
>
> I buy the argument that $R_{miss}$ and $R_{redun}$ are better tailored to schema retrieval than recall and precision, but that still does not explain why their combination cannot be via some form of harmonic mean. But I understand that once you justify moving away from recall and precision, there is less compulsion to restrict to standard combinations as well.

---

> ### Author Response · Authors · 2024-11-29
>
> > Basically an ablation replacing your measures with recall and precision. Does that make sense?
>
> Thanks for your question. We conducted an ablation study replacing $R_{miss}$ and $R_{redun}$ with $R _{recall}$ and $R _{precision}$ during evaluation, and the results are shown in the table below.
>
> From the comparison between $R _{recall}$ and $R _{miss}$ in Table 1, we can observe the following:
>
> - **$R _{recall}$ does not consistently correspond to SQL generation accuracy.** Since missing elements directly lead to the inability to generate correct SQL, which recall does not accurately capture.
>
>   - For example, DTS-SL ($R_{recall}$ = 77.73% ) and TA-SL ($R_{recall}$ = 77.74%) have nearly the same recall values, yet there are significant differences in their SQL generation accuracy.
>   - KaSLA shows significantly better SQL generation performance than CodeS-SL. However, it is difficult to distinguish between them based on recall ($R_{recall}$  = 97.71% and $R_{recall}$  = 96.43%).
>
> - **$R_{miss}$ can reflect actual SQL generation accuracy because $R_{miss}$ provides strict constraints for any element missing.**
>
>   - The evaluation of DTS-SL and TA-SL using $R_{miss}$ is more reasonable. DTS-SL ($R_{miss}$ = 35.77%) performs better than TA-SL ($R_{miss}$  = 43.80%) because it has fewer instances with missing elements.
>   - Using $R _{miss}$, the distinction between KaSLA and CodeS-SL becomes clear, with $R _{miss}$  = 5.83% for KaSLA and $R_ {miss}$  = 12.53 % for CodeS-SL.
>
> *Table 1. Evaluation with $ R _{recall}$ 和 $R _{miss}$ on Bird-dev for different schema linking models with CodeS-15B as the text-to-SQL model*
>
> |                         |                  | Pure-SL | DTS-SL | TA-SL  | CodeS-SL | KaSLA      |
> | ----------------------- | ---------------- | ------- | ------ | ------ | -------- | ---------- |
> | Column $R _{recall}$    | larger is better | 68.79%  | 77.73% | 77.74% | 96.43%   | **97.71%** |
> | Column $R_{miss}$       | less is better   | 57.54%  | 35.77% | 43.80% | 12.53%   | **5.83%**  |
> | Execution Accuracy (EX) | larger is better | 50.26%  | 55.22% | 53.13% | 60.30%   | **63.75%** |

---

> ### Author Response · Authors · 2024-11-29
>
> From the comparison between $R_{precision}$ and $R _{redun}$ in Table 2, we can observe the following:
>
> - **$R_{precision}$ is biased towards models with a severe element missing, which can mislead the evaluation because missing elements directly result in the inability to generate correct SQL.**
>   - For example, Pure-SL ($R_{miss}$ = 57.54%) and TA-SL ($R_{miss}$ = 43.80%) have significant element missing, yet they can easily achieve very high $R_{precision}$ scores ($R_{precision}$ = 80.06% and $R_{precision}$ = 84.92%, respectively), which is inconsistent with their SQL generation accuracy.
> - **$R_{redun}$ ensures that all necessary elements are included in the linking results when evaluating redundancy, thus preventing disruptions in accurate redundancy assessment.**
>   - By considering element missing, the evaluations by $R _{redun}$ for Pure-SL ($R _{redun}$ = 61.75 %) and TA-SL ($R _{redun}$ = 46.25 %) are more reasonable.
>   - KaSLA achieved the lowest $R_{redun}$ = 42.24% and the lowest $R_{miss}$ = 5.83%, resulting in the best SQL generation accuracy compared to other methods.
>
> *Table 2. Evaluation with $ R _{precision}$ 和 $R _{redun}$ on Bird-dev for different schema linking models with the same SQL generation model*
>
> |                         |                  | Pure-SL | DTS-SL | TA-SL      | CodeS-SL | KaSLA      |
> | ----------------------- | ---------------- | ------- | ------ | ---------- | -------- | ---------- |
> | Column  $R_{precision}$ | larger is better | 80.06%  | 15.15% | **84.92%** | 19.32%   | 54.44%     |
> | Column $R_{redun}$      | less is better   | 61.75%  | 83.04% | 46.25%     | 80.83%   | **42.24%** |
> | Execution Accuracy (EX) | larger is better | 50.26%  | 55.22% | 53.13%     | 60.30%   | **63.75%** |
>
> Thanks very much for raising further concerns and discussing them with us. We appreciate the opportunity to speak with you, and if you have other concerns, we can discuss them in depth.

---

> ### Author Response · Authors · 2024-12-01
> **Looking forward to your constructive feedback**
>
> Dear reviewer dUxV,
>
> Thanks again for taking the time to review our paper and for providing valuable feedback on our responses. We have carefully responded to your latest concerns. We discussed the intriguing idea of a metrics-based knapsack-like selection strategy thoroughly and provided an ablation study on traditional metrics, such as recall and precision, following your suggestions. We hope these responses can address your latest concerns and thanks for your support.
>
> As the author-reviewer discussion deadline approaches, we would be very grateful if you could share your valuable thoughts with us. If you have any further concerns, we look forward to further discussion with you.

---

### Official Review · Reviewer_jitH · 2024-11-05

**Soundness:** 2
**Presentation:** 3
**Contribution:** 2
**Rating:** 5
**Confidence:** 5

**Summary:**

The paper introduces a table (and column) retrieval/schema linking method for RAG-base text-to-SQL systems. The proposed approach called Knapsack Schema Linking Agent employ metrics for missing and redundant schema linkages explicitly optimizing the retrieval recall and precision. To constrain the redundancy in the table retrieval process, the paper proposes a capacity estimation method to limit the selection. The paper also proposes two benchmarks for training and evaluating the table retrieval model and task. Experimental evaluation demonstrates its superior performance on benchmarks.

**Strengths:**

S1. Table retrieval/schema linking is an important problem often overlooked or over simplified by the existing text-to-SQL systems. Real life  text-to-SQL use cases often deal with hundreds, thousands and even over hundreds of thousands tables (extreme case in data lake scenarios). The prompt can only contain a small amount of tables requiring high recall and high precision table retrieval/ranking method. This paper addresses this important problem although not at scale.

S2. The idea of explicitly measuring and trading off potential missing and redundant tables is interesting.

S3. The experimental evaluation demonstrates its superior performance against baselines on the proposed benchmarks and BIRD.

**Weaknesses:**

W1. As mentioned in S1, as the number of tables in a database where the text-to-SQL is performed against is growing, the problem of table retrieval becomes increasingly challenging. To really assess the impact of this work, it will be great if the paper can discuss and address the size of full schema |S| being realistically large.

W2. The proposed method requires a fair amount of fine-tuning efforts. The application scenario is unclear. Such an approach can be valid for competing in benchmark like BIRD where the dev and holdout test has similar distribution. It works when the type of questions and the schema design in the database is fairly stable over time. In the event of significant schema changes, does the training/fine-tuning need to happen again? The paper lacks of discussion on the exact text-to-SQL usecase.

W3. Text-to-SQL generation often deals with SQLs involving multiple tables (e.g. JOIN). Join keys can be semantically meaningless to the question (arbitrary id columns) but very important to the generation. It is unclear how is this taken into consideration when the proposed approach is performing column retrieval/selection.

W4. The paper lacks of details on the table representation. How is cell values considered?

W5. In real life application, table names and/or column names can be explicitly mentioned in the question. This motivates the use of sparse index such as inverted index. Can inverted index be used as a part of this approach to improve the recall/precision?

**Questions:**

Please respond to/comment on W1-W5.

---

> ### Author Response · Authors · 2024-11-24
> **Response to W1**
>
> Dear reviewer jitH,
>
> Thanks for your thoughtful suggestions and concerns. We have addressed each of your points carefully to alleviate any concerns and updated a revision.
>
> > W1. As mentioned in S1, as the number of tables in a database where the text-to-SQL is performed against is growing, the problem of table retrieval becomes increasingly challenging. To really assess the impact of this work, it will be great if the paper can discuss and address the size of full schema |S| being realistically large.
>
> Thanks for noting KaSLA's advantage in large-scale real applications. LLM-based text-to-SQL faces the following challenges: 1) super large schema sizes; 2) human-defined table or column names.
>
> **1) Super large schema size**
>
> - **KaSLA uses a hierarchical approach to shrink very super schema sizes.** The results below demonstrate that for large-sized schemas (with column numbers ≥ 50 or table numbers ≥ 8), KaSLA shows more significant improvements compared to baselines than for smaller schemas (with column numbers < 50 or table numbers < 8). Especially when there are many columns, KaSLA has a more pronounced advantage.
> - **The main reason is that KaSLA employs a hierarchical process for schema linking, first linking tables and then linking columns within those tables.** Compared to linking columns from all tables, this method is optimal because it reduces the selection space for columns, lowers time costs, and avoids noise and confusion from unlinked tables' columns.
>
> *Table W1.a Text-to-SQL performance on the BIRD-dev data with different table size*
>
> | Execution Accuracy (EX)                             | SuperSQL | CodeS  | KaSLA      |
> | --------------------------------------------------- | -------- | ------ | ---------- |
> | BIRD-dev with **table number >= 8** (753 instances) | 65.47%   | 64.01% | **66.67%** |
> | BIRD-dev with table number < 8  (781 instances)     | 51.98%   | 53.14% | **60.95%** |
>
> *Table W1.b Text-to-SQL performance on the BIRD-dev data with different column size*
>
> | Execution Accuracy (EX)                                | SuperSQL | CodeS  | KaSLA      |
> | ------------------------------------------------------ | -------- | ------ | ---------- |
> | BIRD-dev with **column number >= 50** (1140 instances) | 53.37%   | 52.70% | **59.06%** |
> | BIRD-dev with column number < 50  (394 instances)      | 69.56%   | 70.56% | **73.59%** |
>
> **2) Complex human-defined table or column names**
>
> - **KaSLA is adaptive to complex human-defined table and column names because the nomination and guaranteed scoring models provide robust score estimates.** These two scoring models deliver discriminative and generalized predictions to achieve robust importance score predictions, which is more robust in handling complex human-defined table or column names, enabling KaSLA to understand complex semantic information in real-world scenarios.
> - **In BIRD-dev, there are 6 databases (843 instances) with scenarios not present in the training data, meaning the semantic background of their table and column names was not covered during training.** As shown in the table, KaSLA demonstrates a clear advantage over baselines in dealing with both presented and non-presented development data.
>
> *Table W1.c Text-to-SQL performance of KaSLA on the presented dev data and nonpresented dev data of BIRD-dev*
>
> | Execution Accuracy (EX)                | SuperSQL | CodeS  | KaSLA      |
> | -------------------------------------- | -------- | ------ | ---------- |
> | Original BIRD-dev (1534 instances)     | 58.60%   | 60.30% | **63.75%** |
> | Presented BIRD-dev  (691 instances)    | 69.90%   | 67.58% | **71.92%** |
> | Nonpresented BIRD-dev  (843 instances) | 49.35%   | 51.01% | **57.06%** |

---

> ### Author Response · Authors · 2024-11-24
> **Responses to W2, W3**
>
> > W2. The proposed method requires a fair amount of fine-tuning efforts. The application scenario is unclear. Such an approach can be valid for competing in benchmark like BIRD where the dev and holdout test has similar distribution. It works when the type of questions and the schema design in the database is fairly stable over time. In the event of significant schema changes, does the training/fine-tuning need to happen again? The paper lacks of discussion on the exact text-to-SQL usecase.
>
> Thank you for bringing up this important question. We discussed the application of KaSLA in an exact text-to-SQL use case and found that the KaSLA trained on the public dataset can be directly applied to a new text-to-SQL scenario without additional training/finetuning.
>
> - **KaSLA trained on Spider shows strong performance on BIRD, and vice versa.** As shown in the table, even when trained on other datasets, KaSLA provides good schema linking performance.
>
> *Table W2.a Text-to-SQL performance of KaSLA trained on Spider-train but evaluated on Bird-dev*
>
> | Execution Accuracy (EX) | SuperSQL | CodeS  | KaSLA trained on Spider-train | KaSLA trained on BIRD-train |
> | ----------------------- | -------- | ------ | ----------------------------- | --------------------------- |
> | **BIRD**-dev            | 58.60%   | 60.30% | 63.10%                        | 63.75%                      |
>
> *Table W2.b Text-to-SQL performance of KaSLA trained on BIRD-train but evaluated on Spider-dev*
>
> | Execution Accuracy (EX) | SuperSQL | CodeS  | KaSLA trained BIRD-train | KaSLA trained on **Spider**-train |
> | ----------------------- | -------- | ------ | ------------------------ | --------------------------------- |
> | **Spider**-dev          | 87.04%   | 84.72% | 87.23%                   | 88.01%                            |
>
> - **BIRD already considers transferability evaluation.** As shown in the table, there are six dev databases with 843 instances not included in the training dataset. KaSLA performs well on both presented and non-presented dev data, demonstrating strong generalizability and transferability (as discussed in our response to W.1).
>
> *Table W2.c Text-to-SQL performance of KaSLA on the presented dev data and nonpresented dev data of BIRD-dev*
>
> | Execution Accuracy (EX)                | SuperSQL | CodeS  | KaSLA      |
> | -------------------------------------- | -------- | ------ | ---------- |
> | Original BIRD-dev (1534 instances)     | 58.60%   | 60.30% | **63.75%** |
> | Presented BIRD-dev  (691 instances)    | 69.90%   | 67.58% | **71.92%** |
> | Nonpresented BIRD-dev  (843 instances) | 49.35%   | 51.01% | **57.06%** |
>
> > W3. Text-to-SQL generation often deals with SQLs involving multiple tables (e.g. JOIN). Join keys can be semantically meaningless to the question (arbitrary id columns) but very important to the generation. It is unclear how is this taken into consideration when the proposed approach is performing column retrieval/selection.
>
> Thanks for highlighting the challenge of handling multi-table joins. We will describe how our framework addresses column selection in scenarios involving multiple tables (e.g., JOIN).
>
> - **KaSLA addresses multi-table join tasks by utlizing attention networks in two importance scoring models.** It employs a StarCoder2-based nomination scoring model and a RoBERTa-based guaranteed scoring model. During both training and inference, these models process all tables and their columns simultaneously. The attention mechanisms facilitate the connection of columns across different tables, enabling accurate linking when a query requires join operations.
> - The results from instances involving JOIN operations in BIRD-dev demonstrate KaSLA's advantages over the baseline.
>
> *Table W3.a Text-to-SQL performance of KaSLA on the BIRD-dev data involving and not involving the JOIN operation*
>
> | Execution Accuracy (EX) in BIRD-dev          | SuperSQL | CodeS  | KaSLA      |
> | -------------------------------------------- | -------- | ------ | ---------- |
> | BIRD-dev involving JOIN (1140 instances)     | 56.67%   | 56.75% | **62.89%** |
> | BIRD-dev not involving JOIN  (394 instances) | 64.21%   | 63.45% | **66.24%** |

---

> ### Author Response · Authors · 2024-11-24
> **Responses to W4, W5**
>
> > W4. The paper lacks of details on the table representation. How is cell values considered?
>
> Thank you for raising this important point.
>
> - **The inputs for both the StarCoder2-based nomination scoring model and the text-to-SQL model include cell values.** The reasoning ability of LLMs can help us to understand these cell values to achieve better nomination scores and SQL generation. We use the retrieval method from [1] to obtain the most representative cell value for each schema element, treating it as part of the element's description.
> - Below is the input format of the StarCoder2-based nomination scoring model. Each column has its corresponding cell value.
>
> ```
> database schema:
> table department, columns = [ department_id (int|primary key|values: 1) , name (text|values: State, Treasury), ranking (int|values: 1)]
> table head ,columns = [ head_id (int|primary key|values: 1) , name (text|values: Tiger Woods) , age (real|values: 67.0)]
> table management, columns = [ department_id ( int|primary key|values: 2), head_id (int|primary key|values: 5), temporary_acting (text|values: Yes)]
> foreign keys: management.head_id = head.head_id, management.department_id = department.department_id
> Question:
> How many heads of the departments are older than 56 ? List the relevant columns in each table:
> ```
>
> [1] CodeS: Towards Building Open-source Language Models for Text-to-SQL
>
> > W5. In real life application, table names and/or column names can be explicitly mentioned in the question. This motivates the use of sparse index such as inverted index. Can inverted index be used as a part of this approach to improve the recall/precision?
>
> Thanks for your insightful suggestions. We conducted experiments to evaluate the impact of the sparse index method (BM25 + inverted index) on the schema linking task.
>
> - **We used the sparse index retrieval method for preprocessing the full schema for KaSLA.** First, we collected schema element names and query words from the training dataset to construct a retrieval corpus mapping natural language words to schema element names. During inference on the dev dataset, we retrieved schema elements based on query words. For each schema entity, we calculated its score using BM25 and marked the top-1 element as the retrieved element. We then combined the full schema with the retrieved elements as inputs for KaSLA.
> - **Our experiments used the presented BIRD-dev data (843 instances with 6 databases not present in the training data) and the corresponding non-presented BIRD-dev data, as mentioned in our response to W1.** The results are shown in the table below. They indicate that the sparse index method did not enhance KaSLA, possibly because **(1)** although public text-to-SQL datasets are quite large, they are still too small for constructing an effective retrieval corpus, leading to suboptimal retrieval results with noise, which affects KaSLA's performance, and **(2)** KaSLA's StarCoder2-based nomination scoring model and RoBERTa-based guaranteed scoring model can effectively capture schema element names included in the questions through semantic information, making the sparse index introduction unnecessary for improvement.
>
> Thanks for proposing the inspiring idea of using the sparse index retrieval method in the text-to-SQL field. Although current text-to-SQL datasets may not yet support the construction of an effective retrieval corpus, limiting this idea's development, we still believe it is a very promising future research direction.
>
> *Table W5.a: Text-to-SQL Performance of KaSLA with sparse index on presented and non-presented dev data of BIRD-dev.*
>
> | Execution Accuracy (EX)                | KaSLA with sparse index (BM25 + inverted index) | Orign KaSLA |
> | -------------------------------------- | ----------------------------------------------- | ----------- |
> | Original BIRD-dev (1534 instances)     | 63.56%                                          | **63.75%**  |
> | Presented BIRD-dev  (691 instances)    | **72.21%**                                      | 71.92%      |
> | Nonpresented BIRD-dev  (843 instances) | 56.47%                                          | **57.06%**  |

---

> ### Author Response · Authors · 2024-11-24
>
> Thanks for your valuable feedback and concerns. We have addressed all additional discussions in our revision. We invite further discussion on any unresolved concerns.

---

> ### Author Response · Authors · 2024-12-01
> **Looking forward to your constructive feedback**
>
> Dear reviewer jitH,
>
> Thanks again for reviewing our work, and a special thanks for your constructive questions and suggestions on applying KaSLA in real-world scenarios. We have provided detailed responses to each of your concerns, emphasizing the challenges of the schema linking task, supplementing with experiments on cross-scenario transfer capability, and discussing the integration with index retrieval methods, among other detailed responses. We hope these responses can address your concerns and thanks for your support.
>
> As the author-reviewer discussion deadline approaches, we would be very grateful if you could share your valuable thoughts with us. If you have any further concerns, we look forward to further discussion with you.

---

### Official Review · Reviewer_BQ9c · 2024-11-06

**Soundness:** 4
**Presentation:** 3
**Contribution:** 3
**Rating:** 6
**Confidence:** 4

**Summary:**

This paper analyzes the famous schema linking problem in the field of LLM-based text-to-SQL. The main contribution is to formulate the selection of relevant tables/columns into a Knapsack problem. Each schema item is assigned a value, and the objective is to maximize the total value with the constraint that the total cost of all selected schema items should be less than a capacity. This idea is novel and interesting to me. And the authors present sound math proof (except minor issues, discussed below) in the main paper. The experiment results demonstrate that this pre-processing is useful on the dev sets of two well-known benchmarks, BIRD and Spider. Besides, this work also formally defines two intuitive metrics to evaluate/explain the "precision" and "recall" of schema linking from another perspective, namely, missing rate and redundancy rate.

**Strengths:**

1. The paper is well-written and easy to follow.

2. The experimental results verify the motivation.

3. The new formulation of schema linking problem sounds interesting and novel to me.

**Weaknesses:**

**[Updated]: Part of my major concern is resolved, and the minor concern is totally addressed. Thus, I am willing to raise my scores of both Soundness and Rating to the next level. About the Confidence score, I accidently choose 2 in the first round, while I originally intended to choose 4.**

1. Major concern: Although in Figure 1, the authors demonstrate the importance of schema linking (given the oracle set, there is still large margin), the problem that whether pre-extracting relevant schema items is a promising direction is still doubtful, especially considering the high cost of individually determining the scores/weights of each table/column. To the best of my knowledge, the majority of existing benchmarks have a relatively smaller database schema size, which can mostly afford to the prompt length. See the work below (also one missing reference):

_The Death of Schema Linking? Text-to-SQL in the Age of Well-Reasoned Language Models._

link: https://arxiv.org/pdf/2408.07702v2

It proves that, if the schema items can be fully inserted into the prompt, schema linking is not a necessity considering efficiency. In other words, this work circumvents this significant issue on the inference delay (including nomination model StarCoder2, guaranteed scorer RoBERTa-Large, extra statistics calculation and dynamic programming). I even did not see one limitation section discussing it. And a quantitive comparison on the inference time w/ and w/o your method is inevitable to trade-off the benefits and disadvantages.

2. Minor issue: some technical design or math is really confusing. Here are some questions:

a. in Eq.(4), you use two scores (both binary $I_{nomi}$ and float $I_{guar}$). How about directly use the soft score $I_{guar}$ as $I_{s,q}$? What is the performance difference? This is important that if a simpler calculation of $I_{s,q}$ can achieve similar expectations, we should always follow the Occam's Razor.

b. About Eq.(8) and (10), the authors define a little complicated equations to estimate the weight and capacity, respectively. I’m curious if there was any mathematical basis behind the design of these formulas, or you just come up with them all from intuition. If the answer is true, please cite them to make the equations more sound. Otherwise, it would be better to prove that this design is empirical and preferred by comparing multiple variants.

c. There are many hyper-parameters in the calculation, e.g., $\tau,\alpha,\gamma$. Could you explain in detail why each of them is needed? Introducing too many tunable parameters may lead readers to suspect that the experimental performance was achieved through careful hyperparameter tuning, which could indicate poor generalizability.

3. Other issues: (not affecting my final judgement at all, just kind suggestions)

i. Be more careful about the notation details, e.g., in Eq.(8), $f_V(s)$, $f_V(s, q)$, and $E_{I\ge \tau}(S)$, if the result is related to concrete input question $q$, please add the symbol $q$. Otherwise, the reviewers will wonder whether this is a benchmark-level statistics or instance-level score.

ii. Try to simplify some equations to avoid over-complicating the introduction. For example, the value estimation $V_{s,q}$ re-uses the importance score in Eq.(7), why not merge them into one? This will also alleviate issue i).

**Questions:**

All questions are introduced in the Weaknesses section.

**Details Of Ethics Concerns:**

None.

---

> ### Author Response · Authors · 2024-11-24
> **Response to W1 (1/2)**
>
> Dear reviewer BQ9c,
>
> Thank you very much for your insightful questions and valuable suggestions. We have addressed each point in detail to alleviate your concerns and updated a revision.
>
> > W1. Is schema linking task still important for text-to-SQL according to the following concerns？ Since the majority of existing benchmarks have a relatively smaller database schema size, which can mostly afford the prompt length of LLMs. In addition, "The Death of Schema Linking? Text-to-SQL in the Age of Well-Reasoned Language Models" claims that if the schema items can be fully inserted into the prompt, schema linking is not a necessity considering efficiency.
>
> Thanks for your comments. The purpose of schema linking is to increase the accuracy and the performance of the SQL generation. We can present the advantage of the KaSLA schema linking model from the following three points.
>
> ### The effect of KaSLA
>
> - **Accurate schema linking helps LLMs capture useful information and avoid potentially disruptive information.** We demonstrated the significant impact of schema linking on SQL generation in the table below.
> - **Adding KaSLA shows significant improvement over using the full schema**. As shown in the table, using KaSLA significantly outperforms using the full schema.
>
> *Table W1.a Comparison of full Schema and KaSLA with CodeS-15B as the Text-to-SQL Model*
>
> | Execution Accuracy (EX) | Full Schema with CodeS-15B | KaSLA with CodeS-15B |
> | ----------------------- | -------------------------- | -------------------- |
> | BIRD-dev                | 52.61%                     | 60.89%               |
> | Spider-dev              | 79.98%                     | 84.24%               |
>
> *Table W1.b Comparison of full Schema and KaSLA with CodeS-15B as the Text-to-SQL Model*
>
> | Execution Accuracy (EX) | Full Schema with StarCoder2-15B | KaSLA with StarCoder2-15B |
> | ----------------------- | ------------------------------- | ------------------------- |
> | BIRD-dev                | 55.54%                          | 63.75%                    |
> | Spider-dev              | 81.91%                          | 88.01%                    |
>
>
> ### The viewpoint of the provided paper
>
> Thank you for suggesting the paper. We have cited and discussed it in our revision. The paper presents two viewpoints that also support our findings.
>
> - **Normal schema linking helps moderately sized models.** The paper [1] found that schema linking can significantly enhance SQL generation with moderately sized LLMs, like Llama-3.1-8B (findings are in Section 4.2 of [1]). However, normal schema linking may not improve SQL generation with extremely large LLMs, such as GPT4-turbo, GPT-4o, and Llama 3.1-405b (findings are in Section 4.2 of [1]).
> - **Accurate schema linking helps extremely large LLMs. The reason schema linking does not improve performance with extremely large LLMs is due to the use of sub-optimal schema linking methods in [1], which limit the LLMs.** Since [1] didn’t release their code, we tested our KaSLA on the following SOTA models [2,3,4] with extremely large LLMs and still observed performance improvements.
>
> *Table W1.c Integrating KaSLA can further enhance extremely large text-to-SQL models*
>
> | Execution Accuracy (EX) | KaSLA  | DAIL-SQL (GPT-4-turbo) | DAIL-SQL (GPT-4-turbo) +KaSLA | CHESS (GPT-4-turbo) |  CHESS (GPT-4-turbo) +KaSLA | E-SQL (GPT-4o) | E-SQL (GPT-4o)  +KaSLA|
> | ----------------------- | ------ | --------------------- | ---------- | ------------------- | ---------- | -------------- | ---------- |
> | BIRD-dev                | 63.75% | 54.43%                | 57.89%     | 63.10%              | 63.89%     | 65.25%         | 65.91%     |
> | Spider-dev              | 88.01% | 83.08%                | 86.85%     | 87.14%              | 88.20%     | 88.30%         | 88.68%     |
>
> [1] The Death of Schema Linking? Text-to-SQL in the Age of Well-Reasoned Language Models
>
> [2] E-SQL: Direct Schema Linking via Question Enrichment in Text-to-SQL
>
> [3] Chess: Contextual harnessing for efficient sql synthesis
>
> [4] Text-to-sql empowered by large language models: A benchmark evaluation

---

> > ### Comment · Reviewer_BQ9c · 2024-11-26
> > **Good point about classifying the necessity of SL on different sized models.**
> >
> > I accept the point of discussing diferent cases since for many downstream applications, an extremely large LLM is not needed, or even too expensive. From this point, my major concern has been resolved.

---

> ### Author Response · Authors · 2024-11-24
> **Response to W1 (2/2) and W2 (1/4)**
>
> > W1.2  This work circumvents this significant issue on the inference delay (including nomination model StarCoder2, guaranteed scorer RoBERTa-Large, extra statistics calculation and dynamic programming). I even did not see one limitation section discussing it. And a quantitive comparison on the inference time w/ and w/o your method is inevitable to trade-off the benefits and disadvantages.
>
> Thank you for your suggestion regarding the trade-off between inference delay and accuracy. We provide the inference time on one 80G A100 GPU for the whole framework, both with and without KaSLA, as follows:
>
> *Table W1.2.a Inference time cost of each component in KaSLA*
>
> | Inference time  / instance | Schema  linking model | Nomination model | Guaranteed model | Factor estimation and dynamic programming | Text-to-SQL |
> | -------------------------- | --------------------- | ---------------- | ---------------- | ----------------------------------------- | ----------- |
> | BIRD-dev                   | Full schema           | \                | \                | \                                         | 14.35 s     |
> |                            | KaSLA                 | 12.87 s          | 0.12 s           | < 0.01 s                                  | 8.80 s      |
> | Spider-dev                 | Full schema           | \                | \                | \                                         | 9.69 s      |
> |                            | KaSLA                 | 9.17 s           | 0.06 s           | < 0.01 s                                  | 7.89 s      |
>
> We also provide the SQL generation accuracy for text-to-SQL, both with and without KaSLA, as follows:
>
> *Table W1.2.b Comparison of text-to-SQL accuracy with full Schema and KaSLA*
>
> | Execution Accuracy (EX) | Text-to-SQL with full schema | Text-to-SQL with KaSLA |
> | ----------------------- | ---------------------------- | ---------------------- |
> | BIRD-dev                | 55.54%                       | 63.75%                 |
> | Spider-dev              | 81.91%                       | 88.01%                 |
>
> As shown in these tables, we can conclude that:
>
> - **KaSLA introduces a significant improvement with a marginal inference delay.** This delay occurs because we use StarCoder2-15B to generate nomination scores for the full schema, aiming for better semantic understanding, which results in additional time consumption. Reducing the inference time for LLM-based nomination models during full schema processing might be a potential research direction.
>
> > W2: Minor issue: some technical design or math is really confusing. Here are some questions:
>
> > W2.1  in Eq.(4), you use two scores (both binary $ I _{\text{nomi}} $ and float $ I _{\text{guar}} $). How about directly use the soft score $ I _{\text{guar}} $ as $I _{s,q}$? What is the performance difference? This is important that if a simpler calculation of $I _{s,q}$ can achieve similar expectations, we should always follow the Occam's Razor.
>
> Thank you for this insightful question. We have provided the ablation study results of the importance scores in the table below.
>
> - **Only using the soft score $ I _{\text{guar}} $ and excluding the binary score $ I _{\text{nomi}} $ leads to sub-optimal performance.** This is because the RoBERTa-Large based guaranteed model offers only a general representation. This potential lack of obvious differentiation in scoring might introduce confusion, which the StarCoder2-based nomination model will address.
> - **Only using the binary score $ I _{\text{nomi}} $ also results in sub-optimal outcomes.** The Starcoder2-based nomination model tends to confirm elements with high confidence, leading to significant differences in scoring. This may result in missing elements that also match, but with medium confidence, which the RoBERTa-Large based guaranteed model should address.
>
> These results highlight the necessity of combining the binary score $ I _{\text{nomi}} $ and the soft score $ I _{\text{guar}} $ for effective importance score estimation.
>
> *Table W2.1.a Ablation study about $I_{\text{nomi}}$ and $I_{\text{guar}}$*
>
> | Execution Accuracy (EX) | $I _ {\text{nomi}}$ and $I _ {\text{guar}}$ | w/o $I _ {\text{guar}}$ | w/o $I _ {\text{nomi}}$ |
> | ----------------------- | --------------------------------------- | --------------------- | --------------------- |
> | BIRD-dev                | 63.75%                                  | 53.13%                | 57.63%                |
> | Spider-dev              | 88.01%                                  | 86.36%                | 85.40%                |

---

> > ### Comment · Reviewer_BQ9c · 2024-11-26
> > **A little question about the inference time and a little surprising about the soft/binary score.**
> >
> > For the inference time table, I disagree that the time delay is "marginal" (almost 1.5-2.0 times delay). As you sayed, you use StarCoder2-15B (a relatively powerful and large model) to generate nomination scores (which I also think this relatively large size is needed). But this size of LLM also shows to be time consuming. These overheads maybe unacceptable in tasks requiring timely responses. Thus, **please add a separate Limitation section or at least in the Appendix to showcase this extra burden. Each coin has two side and it's OK to leave this trade-off to real scenarios**. BTW, I am a little strange about the faster inference time on pure text-to-SQL caused by KaSLA. I can accept that it may be faster since fewer schema items need to be included, but considering the parallel encoding strategy, the gap should not be that large?
> >
> > For the ablation of soft/hard scores, it is interesting that, a hard decision $I_{nomi}$ proves to be much more effective than soft score on the tougher BIRD benchmark. And the combination of all these scores can be considered as a small multi-agent group where all agents make the decision together. That's interesting and make sense.

---

> > > ### Author Response · Authors · 2024-11-26
> > > **Response to "A little question about the inference time and a little surprising about the soft/binary score" (2/3)**
> > >
> > > > BTW, I am a little strange about the faster inference time on pure text-to-SQL caused by KaSLA. I can accept that it may be faster since fewer schema items need to be included, but considering the parallel encoding strategy, the gap should not be that large?
> > >
> > >
> > >
> > > Thanks a lot for sharing your further concerns about the time gap on SQL generation.
> > >
> > > - **We found that KaSLA significantly reduces the number of elements and the prompt length compared to the full schema, which explains the notable gap in SQL generation speed.** We compiled statistics on the average number of elements in the full schema and the schema linked by KaSLA, along with the prompt length. The results of column are presented in Table a and Table b below.
> > > - **As shown in Table, on BIRD-dev, KaSLA reduces the number of linked columns by 89.44% and the prompt length by 79.27%.** As you mentioned, since KaSLA provides fewer schema elements, this leads to a noticeable improvement in SQL generation inference speed.
> > >
> > > Thank you for introducing the discussion about SQL generation speed under the parallel encoding strategy.
> > >
> > > - **In a parallel setting, if the GPU memory constraint remains the same, KaSLA may increase the inference speed gap compared to the full schema.** KaSLA's shorter prompt length requires less GPU memory, allowing it to use a larger batch size to process more data, thereby further enhancing efficiency. In our implementation of SQL generation with the full schema, we found that we can only process 1-2 instances in parallel on one GPU (with the risk of memory overflow). Conversely, KaSLA shortens the overall prompt, allowing 3-4 instances to be processed in parallel, thus accentuating KaSLA's advantages.
> > > - **If we do not impose a GPU memory constraint and use the same batch size, we believe the gap will be smaller**, as the inference speed for both would improve.
> > >
> > >
> > >
> > > *Table a. Average column number of full schema and linking results of KaSLA.*
> > >
> > > | **Average column number** | **Full schema** | **Linking results of KaSLA** | **%Reduce** |
> > > | ------------------------- | --------------- | ---------------------------- | ----------- |
> > > | BIRD-dev                  | 75.56           | 7.98                         | - 89.44 %   |
> > > | Spider-dev                | 24.55           | 3.22                         | - 86.88 %   |
> > >
> > > *Table b. Average prompt length of full schema and linking results of KaSLA.*
> > >
> > > | Average prompt length (tokens) | **Full schema** | **Linking results of KaSLA** | **Reduce %** |
> > > | ------------------------------ | --------------- | ---------------------------- | ------------ |
> > > | BIRD-dev                       | 1751            | 363                          | - 79.27 %    |
> > > | Spider-dev                     | 662             | 177                          | - 73.26 %    |

---

> > > ### Author Response · Authors · 2024-11-26
> > > **Response to "A little question about the inference time and a little surprising about the soft/binary score" (3/3)**
> > >
> > > > For the ablation of soft/hard scores, it is interesting that, a hard decision Inomi proves to be much more effective than soft score on the tougher BIRD benchmark. And the combination of all these scores can be considered as a small multi-agent group where all agents make the decision together. That's interesting and make sense.
> > >
> > > - **Thank you so much for sharing your insight. We believe the significant impact of the hard decision $I _{\text{nomi}}$ is due to the ability of LLMs like StarCoder2 to effectively comprehend the complex linguistic environment of the challenging BIRD benchmark.** Consequently, it has a greater impact than the RoBERTa-based guaranteed model, which provides more general representations. Nonetheless, the RoBERTa-based guaranteed model remains very important in this context, **highlighting the value of multi-agent collaboration.**
> > >
> > > - **As you mentioned, studying the collaboration of several small or moderately sized models as multi-agents is indeed an intriguing idea.** It is possible that the collaboration of multiple small models could achieve results comparable to those of extremely large LLMs in the future. We appreciate the opportunity to discuss this impactful idea with you, and we are grateful for this thoughtful discussion.

---

> ### Author Response · Authors · 2024-11-24
> **Response to W2 (2/4)**
>
> > W2.2 About Eq.(8) and (10), the authors define a little complicated equations to estimate the weight and capacity, respectively. I’m curious if there was any mathematical basis behind the design of these formulas, or you just come up with them all from intuition. If the answer is true, please cite them to make the equations more sound. Otherwise, it would be better to prove that this design is empirical and preferred by comparing multiple variants.
>
>
>
> Thank you for your question. We will explain the definitions of Eq. (8) and Eq. (10) in detail and report on related empirical studies. To aid understanding, we'll use table linking as an example, where, in Eq. (8) and Eq. (10),  $q$ is a query, $s$ is a table and $S$ is the set of all tables.
>
>
>
> Eq. (8): $\widehat{W} _{s,q} = f _{W}(s,q)  = \lfloor {(I _{s,q} - \mathbb{E} _{I \geq \tau}(S, q) + 1)}^{-1} \rfloor$ represents the estimation of the weight for table $s$.
>
> **Detailed  description**
>
> There are several important components in Eq. (8):
>
> - $I _{s,q} $ denotes the importance score of table $s$ for query $q$.
> - $\mathbb{E} _{I \geq \tau}(S, q) = \mathbb{E}\left[ { {\lbrace I _{s,q} \mid I _{s,q} \geq \tau \rbrace } } _{s \in S} \right]$:
>   -  ${ {\lbrace I _{s,q} \mid I _{s,q} \geq \tau \rbrace } } _{s \in S}$ represents an important table set, including tables with an importance score greater than $\tau$, where $\tau=0.5$ is the threshold for determining importance.
>   - $\mathbb{E} _{I \geq \tau}(S, q)$ is the expected importance score of this important table set.
> - $I _{s,q} - \mathbb{E} _{I \geq \tau}(S, q)$ represents the relative importance of $s$. It represents how far the table $s$ is from the important table set. $I _{s,q} - \mathbb{E} _{I \geq \tau}(S, q) > 0$ indicates higher importance, while $I _{s,q} - \mathbb{E} _{I \geq \tau}(S, q) < 0$ indicates lower importance.
> - In Eq. (8):
>   - The inverse power emphasizes that the weight of $s$ is inversely proportional to its relative importance; the less important it is, the more it is potentially redundant and should have a larger weight.
>   - Adding 1 to the denominator prevents division by zero.
>   - The use of the floor function simplifies the complexity of the dynamic programming-based knapsack optimization.
>
>
> **Empirical Studies:**
>
> The detailed experimental results of multiple variants are as follows, we can see that Eq. (8) can help KaSLA achieve optimal performance.
>
> *Table W2.2.a Empirical studies of  multiple variants of Eq. (8)*
> | Variants                                                     | Description                                                  | Execution Accuracy (EX) |
> | ------------------------------------------------------------ | ------------------------------------------------------------ | ----------------------- |
> | $\widehat{W} _{s,q} = f _{W}(s,q)  = \lfloor {(I _{s,q})}^{-1} \rfloor$ | Simply defining the weight as inversely related to the importance score | 61.73%                  |
> | $\widehat{W} _{s,q} = f _{W}(s,q)  = \lfloor {(I _{s,q} - \mathbb{E} _{I}(S, q) + 1)}^{-1} \rfloor$ | Using the average of the importance scores of all tables ($\mathbb{E} _{I}(S, q) = \mathbb{E}\left[ {\{I _{s,q} \}} _{s \in S} \right]$)to calculate relative importance | 62.26%                  |
> | $\widehat{W} _{s,q} = f _{W}(s,q)  = \lfloor {(I _{s,q} - \mathbb{E} _{I \geq \tau}(S, q) + 1)}^{-1} \rfloor$ | (Eq. (8)) Using the average of importance scores of more important tables ($\mathbb{E} _{I \geq \tau}(S, q) = \mathbb{E}\left[ {\{I _{s,q} \mid I _{s,q} \geq \tau \}} _{s \in S} \right]$) to calculate relative importance. | 63.75%                  |

---

> ### Author Response · Authors · 2024-11-24
> **Response to W2 (3/4)**
>
> **(2) Eq. (10)**
>
> For query $q _d$ in the training data, we use ${S^{'}} _d$ to denote its ground truth schema linking results. We use different symbols ${S^{'}} _d$ here compared to our paper because markdown cannot display the original symbols.
>
> Eq. (10): $\text{Knap}(d) = {S^{'}} _d + { \lbrace s \mid I _{s,q _d} \geq \min { \lbrace I _{z,q _d}\rbrace } _{z \in {S^{'}} _d} \rbrace } _{s \in {S} _d \setminus {S^{'}} _d}$ represents a assumed full knapsack $\text{Knap}(d)$  defined for the query $q _d$. It reflects the actual schema linking effect of KaSLA and will be used in Eq. (11) to estimate capacity.
>
> - In practice, even an optimized KaSLA might not eliminate all redundant elements. Ideally, the schema linking results would include all matching elements (${S^{'}} _d$ in Eq. (10)) and a small number of high-importance score un-matching elements (${ \lbrace s \mid I _{s,q _d} \geq \min { \lbrace I _{z,q _d}\rbrace } _{z \in {S^{'}} _d} \rbrace } _{s \in {S} _d \setminus {S^{'}} _d}$ in Eq. (10)).
> - We use the lowest importance score among the ground truth linking results as a threshold, $\min { \lbrace I _{z,q _d} \rbrace } _{z \in {S^{'}} _d}$. Un-matching elements whose importance score exceeds this threshold are likely to be incorrectly included in the schema linking results by KaSLA.
>
> In summary, Eq. (10) accounts for potential redundant linking in KaSLA in practical application, helping Eq. (11) estimate a realistic and robust capacity.
>
> **Empirical Studies:**
>
> The detailed experimental results of multiple variants are as follows, we can see that Eq. (10) can help KaSLA achieve optimal performance.
>
> *Table W2.2.b Empirical studies of  multiple variants of Eq. (10)*
> | Variants                                                     | Description                                                  | Execution Accuracy (EX) |
> | ------------------------------------------------------------ | ------------------------------------------------------------ | ----------------------- |
> | $\text{Knap}(d) = {S^{'}} _d $                               | Using only ground truth linking elements introduces overly strict constraints. | 61.15%                  |
> | $\text{Knap}(d) = {S^{'}} _d + { \lbrace s   \rbrace } _{s \in {S} _d \setminus {S^{'}} _d}$ | Using all schema elements introduces overly loose constraints. | 55.74%                  |
> | $\text{Knap}(d) = {S^{'}} _d + { \lbrace s \mid I _{s,q _d} \geq \min { \lbrace I _{z,q _d}\rbrace } _{z \in {S^{'}} _d} \rbrace } _{s \in {S} _d \setminus {S^{'}} _d}$ | **(Eq. (10)** Using ground truth linking elements combined with high-importance un-matching elements introduces appropriate constraints. | 63.75%                  |
>
>
>
>
> > W2.3 There are many hyper-parameters in the calculation, e.g., $ \tau $,$ \alpha $,$ \gamma $. Could you explain in detail why each of them is needed? Introducing too many tunable parameters may lead readers to suspect that the experimental performance was achieved through careful hyperparameter tuning, which could indicate poor generalizability.
>
> Thank you for your suggestions regarding hyperparameters. We explained the three hyper-parameters in the KaSLA framework: $ \tau $,$ \alpha $, and $ \gamma $ in detail as follows.
>
> The usages of these hyper-parameters:
>
> - $ \alpha $ in the estimation function of importance score: $I({s \mid q,S}) = I _{nomi}{(s \mid q,S)} + \alpha I _{guar}{(s \mid q,S)}$, $\alpha>0$
> - $ \tau $ in the expectation function of high importance scores: $\mathbb{E} _{I \geq \tau}(S, q) = \mathbb{E}\left[ {\{I _{s,q} \mid I _{s,q} \geq \tau \}} _{s \in S} \right]$, $0<\tau<1$
> - $ \gamma $ in the estimation function of capacity: $\widehat{C} _q = f _{C}(q) = \gamma \cdot \max  \left[  \{ \sum _{s \in \text{Knap}(d)} \widehat{W} _{s,q} \}\right] _{d \in D}$, $\gamma>0$
>
> ### Explanations:
>
> - **We use $ \alpha $ to control the magnitude of the contribution from the guaranteed scoring model.** A larger $ \alpha $ biases the estimation of the importance score towards the guaranteed score, resulting in a more stable and generalized score estimation. Conversely, a smaller $ \alpha $ shifts the bias towards the nomination score, achieving a more discriminative score estimation.
> - **We use $ \tau $ to determine whether an importance score is truly important.** Elements with an importance score exceeding $ \tau $ are considered truly important, and their expectation is used for weight estimation.
> - **We employ $ \gamma $ to adjust the scaling of the estimated capacity.** In practical applications, scaling to a larger capacity can include more potential matching elements in the linking results, but it may also overly relax constraints, leading to redundancy. We use $ \gamma $ to manage this trade-off.

---

> ### Author Response · Authors · 2024-11-24
> **Responses to W2 (4/4) and W3**
>
> ### Implementation details:
>
> - **KaSLA has good generalizability and does not require careful hyperparameter tuning.** We simply set these parameters as: $ \alpha = 1$, $ \tau = 0.5$, $ \gamma = 1$ in the paper.
> - **We also provided results considering a range of these hyperparameters in the table below.** The results show that the hyperparameter values only slightly affect KaSLA's performance and offer the potential for the most optimal results. This demonstrates the generalization ability of KaSLA.
>
> *Table W2.3.a Ablation study of hyperparameters*
>
> | $ \alpha \in \{0.1, 0.5, 1, 5\}$, $ \tau = 0.5$, $ \gamma = 1$ | $\alpha = 0.1$ | $\alpha =0.5$ | $\alpha =1$  | $\alpha =5$  |
> | ------------------------------------------------------------ | -------------- | ------------- | ------------ | ------------ |
> | BIRD-dev                                                     | 63.10%         | 63.69%        | **63.75%**   | 61.15%       |
> | Spider-dev                                                   | 87.72%         | **88.10%**    | 88.01%       | 86.27%       |
> | $ \alpha = 1$, $ \tau \in \{0.1, 0.5, 0.7, 0.9\}$, $ \gamma = 1$ | $\tau = 0.1$   | $\tau = 0.5$  | $\tau = 0.7$ | $\tau = 0.9$ |
> | BIRD-dev                                                     | 63.10%         | **63.75%**    | 63.43%       | 63.10%       |
> | Spider-dev                                                   | 87.43%         | **88.01%**    | 87.81%       | 87.52%       |
> | $ \alpha = 1$, $ \tau = 0.5$, $ \gamma = \in \{0.5, 1, 1.5, 2\}$ | $\gamma = 0.5$ | $\gamma = 1$  | $\gamma = 1.5$ | $\gamma = 2$ |
> | BIRD-dev                                                     | 49.02%         | **63.75%**    | 63.10%       | 62.84%       |
> | Spider-dev                                                   | 57.25%         | **88.01%**    | 87.72%       | 87.14%       |
>
> > W3.1 Be more careful about the notation details, e.g., in Eq.(8), fV(s), fV(s,q), and EI≥τ(S), if the result is related to concrete input question q, please add the symbol q. Otherwise, the reviewers will wonder whether this is a benchmark-level statistics or instance-level score.
>
> Thanks for identifying this issue. We have refined the equations and notations in the revision.
>
> - For $\mathbb{E} _{I \geq \tau}(S)$, since its inputs should involve $q$, we have updated its notation from $\mathbb{E} _{I \geq \tau}(S)$ to $\mathbb{E} _{I \geq \tau}(S, q)$ in the revised version. Since $f _{V}(s,q)$, $f _{W}(s,q)$, and $f _{C}(q)$ are function definitions, their format remains stable, with $q$ and $s$ as inputs. Therefore, we maintain their format in the revision.
>
> > W3.2 Try to simplify some equations to avoid over-complicating the introduction. For example, the value estimation Vs,q re-uses the importance score in Eq.(7), why not merge them into one? This will also alleviate issue i).
>
> Thanks for your concern. We have reformatted the equations in our paper for improved clarity.
>
> - **For not merging $\hat{V} _{s,q}$ and $I _{s,q}$, the reason is that we aim to treat the importance score estimation and the knapsack factor estimation (value, weight, and capacity) as two separate components.** This makes the entire framework clearer and easier to develop, as the importance scoring model can be trained once and reused anytime. While we use a simple positive correlation for value estimation $\widehat{V} _{s,q} = f _{V}(s,q) = I _{s,q}$, we encourage further exploration of optimized estimation methods.

---

> > ### Comment · Reviewer_BQ9c · 2024-11-26
> > **W.r.t. the math formulation**
> >
> > Very rigorous attitude. The authors:
> > 1. presents a detailed explanation regarding different variants of the weight and Knap design to showcase the superior performances. Although I prefer more succinct math formulations, I can accept that in many cases, we have to introduce more odds and ends for better performances in the real-world (for example, the disgusting "high-importance un-matching elements" filter).
> > 2. gives extra experiments to demonstrate that the three hyperparameters are not sensitive, thus decreasing the possibility that we have to devote much time into deliberating tuning parameters for different benchmarks.

---

> > > ### Author Response · Authors · 2024-11-26
> > > **Rresponse to "W.r.t. the math formulation"**
> > >
> > > Special thanks for your recognition of our paper and responses. We are truly grateful for the opportunity to address your concerns about math formulation and the above. Engaging in this discussion with you has been crucial in improving the quality of our work.
> > >
> > > If you have any further concerns, we would be grateful for the chance to discuss them with you.

---

> ### Author Response · Authors · 2024-11-24
>
> Thanks for your insightful suggestions and concerns. We have incorporated all additional discussions into our revision. We are also open to further discussions on any remaining concerns.

---

> ### Author Response · Authors · 2024-11-26
> **Response to "Good point about classifying the necessity of SL on different sized models"**
>
> Thanks so much for your response and comments. We sincerely appreciate your recognition of the discussion regarding the various cases of schema linking with different scale LLMs, especially considering diverse downstream applications. It is an honor to engage in this discussion with you.

---

> ### Author Response · Authors · 2024-11-26
> **Response to "A little question about the inference time and a little surprising about the soft/binary score" (1/3)**
>
> > For the inference time table, I disagree that the time delay is "marginal" (almost 1.5-2.0 times delay). As you sayed, you use StarCoder2-15B (a relatively powerful and large model) to generate nomination scores (which I also think this relatively large size is needed). But this size of LLM also shows to be time consuming. These overheads maybe unacceptable in tasks requiring timely responses. Thus, **please add a separate Limitation section or at least in the Appendix to showcase this extra burden. Each coin has two side and it's OK to leave this trade-off to real scenarios**.
>
> Thanks so much for your comments and for further expressing your concerns.
> - In light of your suggestions, **we have updated our revised PDF. Specifically, on page 10 (from Line 509 to Line 524), we included a separate limitations section just before the conclusion section to discuss our limitation regarding inference delay.** As you rightly pointed out, while we strive to deliver more accurate results, minimizing inference delay remains an important task. Your understanding and consideration are deeply appreciated, and we have discussed inference delay in detail in the paper.
> - **In the future, we hope to reduce the time usage of the nomination model while processing the entire schema while maintaining accuracy.** Thanks again for this insightful feedback.

---

> ### Author Response · Authors · 2024-12-01
> **Looking forward to your constructive feedback**
>
> Dear reviewer BQ9c,
>
> Thanks again for reviewing our work and for your constructive suggestions and comments. To address your latest concerns, we have included a limitations section in the revision to discuss KaSLA's inference delay, and we have provided detailed responses regarding the SQL generation time reduced by KaSLA and discussed the multi-agent group insight about soft/binary score. We hope these responses can address your latest concerns, and thanks for your support.
>
> As the author-reviewer discussion deadline approaches, we would be very grateful if you could share your valuable thoughts with us. If you have any further concerns, we look forward to further discussion with you.

---

### Official Review · Reviewer_WDRq · 2024-11-08

**Soundness:** 1
**Presentation:** 1
**Contribution:** 2
**Rating:** 3
**Confidence:** 4

**Summary:**

The paper formulates the schema linking for text2sql as a knapsack problem. It defines many concepts, like redundancy, schema missing rate, score functions, as well as the weight, value and the capacity for the knapsack formulation. The paper is hard to read, in general. Some definitions are intuitive, such as redundancy. Models are fine-tuned for the scoring functions, and most formulas are per query. The paper does not convince me that schema linking is a knapsack problem; as the weight and value are both defined in terms of the importance of the schema element, and capacity is a forced parameter that is introduced. The experimental results look promising, but there are various concerns about them, which are explained below.

**Strengths:**

1. The paper provides extensive experiments.
2.

**Weaknesses:**

1. Knapsack formulation is not intuitive, as the capacity constraints is superfluous. If the capacity was formulated depending on the context size limitations, it might have made sense.
2. The scoring model is fine-tuning with ground-truth. This requires having a ground truth set, training models, which will only work for benchmark datasets like BIRD and Spider, but is not solution in practice.
3. Experiments compare multiple things at the same time, so it is difficult to judge whether the accuracy savings are fully attributable to the schema linking process introduced in this paper.
4. Why introduce new metrics like missing schema and redundancy, while precision and recall of schema linking can fully cover these concepts?
5. The experiments do not include two of the SOTA schema linking works, E-SQL and CHESS, both are on the BIRD leadership. Especially, E-SQL focuses mainly on schema linking and does not use many of the other techniques, like self-consistency, to achieve high accuracy. Experiments should have included at least E-SQL.
A. CHESS: Contextual Harnessing for Efficient SQL Synthesis:  Shayan Talaei, Mohammadreza Pourreza, Yu-Chen Chang, Azalia Mirhoseini, Amin Saberi
B. E-SQL: Direct Schema Linking via Question Enrichment in Text-to-SQL: Hasan Alp Caferoğlu, Özgür Ulusoy

**Questions:**

1. Redundancy is a confusing term; do you include the same table/column multiple times in the context? I understand that you do not. This is supposed to capture irrelevant schema elements; i.e. precision. How is this different?
2. Similarly, how is missing schemas different than recall?
3. In the paper, when you say schema it is confusing whether you mean table or column or both. It is not until 4.3 that it gets some clarification. Do you mix schema errors or solve two separate knapsack problems, one for tables and one for columns? The models in the score function, are they trained for tables and columns separately? Column selection is conditional, you choose columns from the selected tables; how does that impact the score functions?
4. How are the scoring functions trained? Ground truth for which dataset? Dev or train?

---

> ### Author Response · Authors · 2024-11-24
> **Response to W1**
>
> Dear reviewer WDRq,
>
> Thanks very much for your insightful suggestions and concerns. We have provided a detailed response to each concern and made improvements in the revised vision.
>
> > W1: Knapsack formulation is not intuitive, as the capacity constraints is superfluous. If the capacity was formulated depending on the context size limitations, it might have made sense.
>
> Thanks for raising this important question. We address it from three angles: the motivation for formulating schema linking as a knapsack problem, the necessity of a capacity constraint, and content-size capacity.
>
> - **Schema linking requires no missing elements and low redundant elements.** Any missing schema element can cause SQL generation to fail directly, while redundant elements can confuse the SQL generation process and lead to incorrect outputs.   As shown in the table, missing tables or columns and adding redundant ones significantly negatively impact text-to-SQL performance.
> - **These requirements align with the knapsack optimization goal of maximizing value while minimizing weight within a capacity constraint.** Therefore, we formulate it as a knapsack problem to address both requirements.
>
> *Table W1.a Randomly drop 1, 2, and 3 columns from all instances' schema linking results.*
>
> |  Execution Accuracy (EX) | Drop 3 columns | Drop 2 columns | Drop 1 column | Origin KaSLA |
> | ----------------------- | ---------------------- | ---------------------- | ---------------------- | ------------ |
> | BIRD-dev                | 27.57%                 | 33.38%                 | 46.09%                 | **63.75%**   |
>
> *Table W1.b Randomly add 10%, 20%, and 30% redundant columns into all instances' schema linking results.*
>
> |  Execution Accuracy (EX) | **Add 30% columns** | **Add 20% columns** | **Add 10% columns** | Origin KaSLA |
> | ----------------------- | --------------------------------------- | --------------------------------------- | --------------------------------------- | ------------ |
> | BIRD-dev                | 56.38%                                  | 58.34%                                  | 60.82%                                  | **63.75%**   |
>
> - **Without a capacity constraint, schema linking models tend to link any possible element, leading to redundancy and decreased performance.** As demonstrated in the table, giving unlimited capacity increases redundancy, thereby reducing text-to-SQL performance.
>
> *Table W1.c Text-to-SQL performance of KaSLA with and without capacity*
>
> | Execution Accuracy (EX) | Without capacity | Origin KaSLA |
> | ----------------------- | ---------------- | ------------ |
> | BIRD-dev                | 55.54%           | **63.75%**   |
>
> - **Content-size capacity also works; however, KaSLA outperforms it by considering the character of specialized queries.** We compared the content-size KaSLA with the original KaSLA. The results show that the original KaSLA significantly outperforms both the baselines and content-size KaSLA because it evaluates capacity according to the query.
>
> *Table W1.d Text-to-SQL performance of content-size KaSLA and origin KaSLA*
>
> | Execution Accuracy (EX) | SuperSQL | CodeS  | content-size KaSLA | Origin KaSLA |
> | ----------------------- | -------- | ------ | ------------------ | ------------ |
> | BIRD-dev                | 58.60%   | 60.30% | 61.73%             | **63.75%**   |
>
> - KaSLA achieves optimal performance by attaining both the lowest column missing ratio and the lowest column redundancy ratio, demonstrating the effectiveness of the proposed knapsack optimization framework.
>
> *Table W1.e Performance on Bird-dev with execution accuracy (EX) for text-to-SQL evaluation and missing ratio and redundancy ratio for schema linking evaluation. The SQL generation model is a fine-tuned StarCoder2-15B*
>
> |                         | Pure-SL | DTS-SL | TA-SL  | CodeS-SL | KaSLA      |
> | ----------------------- | ------- | ------ | ------ | -------- | ---------- |
> | Column missing ratio    | 57.54%  | 35.77% | 43.80% | 12.53%   | **5.83%**  |
> | Column redundancy ratio | 61.75 % | 83.04% | 46.25% | 80.83%   | **42.24%** |
> | Execution Accuracy (EX) | 50.26%  | 55.22% | 53.13% | 60.30%   | **63.75%** |

---

> > ### Comment · Reviewer_WDRq · 2024-11-26
> >
> > Thanks for the detailed response. I understand that there is a trade-off between recall (making sure all needed schema elements are selected) and precision (there is no irrelevant schema element).  But there is no natural capacity requirement. One might argue different models get confused with extra information in the context. But, recent long context models are shown to be robust, and capacity requirement is not needed. There is, of course, a latency and cost implication, as the context size grows.
> >
> > BTW, I am still very confused with redundancy. Do you mean the same schema element is included multiple times in the context, hence redundant? I interpreted redundancy as irrelevancy.

---

> > > ### Author Response · Authors · 2024-11-28
> > >
> > > > One might argue different models get confused with extra information in the context. But, recent long context models are shown to be robust, and capacity requirement is not needed. There is, of course, a latency and cost implication, as the context size grows.
> > >
> > > Thank you for bringing up this insightful discussion.
> > >
> > > - **KaSLA can reduce context size obviously to reduce the latency and cost implications.** As shown in Tables 4 and 5, KaSLA reduces the number of linked columns by 89.44% on the BIRD-dev dataset and shortens the prompt length by 79.27%.
> > >
> > > *Table 4. Average column number of full schema and linking results of KaSLA.*
> > >
> > > | **Average column number** | **Full schema** | **Linking results of KaSLA** | **%Reduce** |
> > > | ------------------------- | --------------- | ---------------------------- | ----------- |
> > > | BIRD-dev                  | 75.56           | 7.98                         | - 89.44 %   |
> > > | Spider-dev                | 24.55           | 3.22                         | - 86.88 %   |
> > >
> > > *Table 5. Average prompt length of full schema and linking results of KaSLA.*
> > >
> > > | Average prompt length (tokens) | **Full schema** | **Linking results of KaSLA** | **Reduce %** |
> > > | ------------------------------ | --------------- | ---------------------------- | ------------ |
> > > | BIRD-dev                       | 1751            | 363                          | - 79.27 %    |
> > > | Spider-dev                     | 662             | 177                          | - 73.26 %    |
> > >
> > > - **Reducing content size by KaSLA can also enhance SQL generation accuracy.** As shown in the Table 6 below, irrelevant elements will confuse the SQL generation model. By introducing a capacity constraint, KaSLA effectively minimizes irrelevant elements, providing more accurate SQL generation results.
> > >
> > > *Table 6. Randomly add 10%, 20%, and 30% irreverent columns into all instances' schema linking results.*
> > >
> > > | Execution Accuracy (EX) | **Add 30% irreverent columns** | **Add 20% irreverent columns** | **Add 10% irreverent columns** | Origin KaSLA |
> > > | ----------------------- | ------------------------------ | ------------------------------ | ------------------------------ | ------------ |
> > > | BIRD-dev                | 56.38%                         | 58.34%                         | 60.82%                         | **63.75%**   |

---

> ### Author Response · Authors · 2024-11-24
> **Responses to W2 and W3**
>
> > W2: The scoring model is fine-tuning with ground-truth. This requires having a ground truth set, training models, which will only work for benchmark datasets like BIRD and Spider, but is not solution in practice.
>
> Thanks for your comments. With the explicit definition of the knapsack framework, KaSLA demonstrates strong generalizability and transferability.
>
> - **KaSLA trained on Spider shows strong performance on BIRD, and vice versa.** As shown in the table, even when trained on other datasets, KaSLA provides solid schema linking performance.
>
> *Table W2.a Text-to-SQL performance of KaSLA trained on Spider-train but evaluated on Bird-dev*
>
> | Execution Accuracy (EX) | SuperSQL | CodeS  | KaSLA trained on Spider-train | KaSLA trained on BIRD-train |
> | ----------------------- | -------- | ------ | ----------------------------- | --------------------------- |
> | **BIRD**-dev            | 58.60%   | 60.30% | 63.10%                        | 63.75%                      |
>
> *Table W2.b Text-to-SQL performance of KaSLA trained on BIRD-train but evaluated on Spider-dev*
>
> | Execution Accuracy (EX) | SuperSQL | CodeS  | KaSLA trained on BIRD-train | KaSLA trained on **Spider**-train |
> | ----------------------- | -------- | ------ | ------------------------ | --------------------------------- |
> | **Spider**-dev          | 87.04%   | 84.72% | 87.23%                   | 88.01%                            |
>
> - **BIRD already considers transferability evaluation.** As shown in the table, there are six dev databases with 843 instances not present in the training dataset. KaSLA performs well on both the presented dev data and the non-presented dev data.
>
> *Table W2.c Text-to-SQL performance of KaSLA on the presented dev data and non-presented dev data of BIRD-dev*
>
> | Execution Accuracy (EX)                | SuperSQL | CodeS  | KaSLA      |
> | -------------------------------------- | -------- | ------ | ---------- |
> | Original BIRD-dev (1534 instances)     | 58.60%   | 60.30% | **63.75%** |
> | Presented BIRD-dev  (691 instances)    | 69.90%   | 67.58% | **71.92%** |
> | Nonpresented BIRD-dev  (843 instances) | 49.35%   | 51.01% | **57.06%** |
>
>
> > W3: Experiments compare multiple things at the same time, so it is difficult to judge whether the accuracy savings are fully attributable to the schema linking process introduced in this paper.
>
> Thanks for your comments. We provide the following two ablation studies to further verify the improvements of our KaSLA model:
>
> - **Can KaSLA enhance general models? Yes.** KaSLA consistently enhances existing text-to-SQL frameworks by improving their schema linking components.
>
> *Table W3.a Text-to-SQL performance of integrating KaSLA into SOTA baselines*
>
> | Execution Accuracy (EX) | DTS-SQL | DTS-SQL  +KaSLA         | DAIL-SQL |  DAIL-SQL +KaSLA         | CHESS  |   CHESS  +KaSLA       | E-SQL  |  E-SQL  +KaSLA        |
> | ----------------------- | ------- | ---------- | -------- | ---------- | ------ | ---------- | ------ | ---------- |
> | BIRD-dev                | 55.22%  | 58.41%     | 54.43%   | 57.89%     | 63.10% | 63.89%     | 65.25% | 65.91%     |
> | Spider-dev              | 85.11%  | 87.43%     | 83.08%   | 86.85%     | 87.14% | 88.20%     | 88.30% | 88.68%     |
>
> - **Does KaSLA outperform other schema linking models? Yes.** KaSLA outperforms schema linking baselines using the same SQL generation model, a pre-trained CodeS-15B.
>
> *Table W3.b Text-to-SQL performance of KaSLA and other schema linking baselines with CodeS-15B as the text-to-SQL model*
>
> | Execution Accuracy (EX) | Pure-SL | DTS-SL | TA-SL  | CodeS  | KaSLA  |
> | ----------------------- | ------- | ------ | ------ | ------ | ------ |
> | BIRD-dev                | 52.61%  | 51.76% | 56.13% | 58.08% | 60.89% |
> | Spider-dev              | 79.98%  | 82.50% | 84.33% | 84.04% | 87.52% |

---

> ### Author Response · Authors · 2024-11-24
> **Response to W4 (1/2)**
>
> > W4: Why introduce new metrics like missing schema and redundancy, while precision and recall of schema linking can fully cover these concepts?
> >
> > Q1: Redundancy is a confusing term; do you include the same table/column multiple times in the context? I understand that you do not. This is supposed to capture irrelevant schema elements; i.e. precision. How is this different?
> >
> > Q2: Similarly, how is missing schemas different than recall?
>
> Thanks for noticing this. We first illustrate the limitations of precision and recall in the schema linking task and then explain why we extend them to missing and redundancy metrics for better evaluation.
>
> **1) Limitations of precision and recall in Schema**
>
> Precision: $R_{precision}=\frac{1}{|B|} \sum_{(q,S) \in B}\frac{|\hat{S}_q \cap S_q|}{|\hat{S}_q|}$
>
> Recall: $R_{recall} = \frac{1}{|B|} \sum_{(q,S) \in B} \frac{|\hat{S}_q \cap S_q|}{|S_q|}$
>
> ($B$ is the whole dev dataset, $\hat{S}_q$ is the predicted schema linking results, and $S_q$ is the ground truth.)
>
> They are both general evaluation metrics; however, they have drawbacks that hinder their usage in schema linking.
>
>
> - **The same recall metric values can correspond to different SQL generation accuracies:** In schema linking, missing a single table or column (recall < 100%) can significantly decrease text-to-SQL generation accuracy. Averaging recall cannot indicate how many instances are correct. We provide two examples in the table below.
> - **As shown in the table**, in example 1, the average recall is 80%. However, since 3 out of 5 instances have missing elements, only 2 instances can generate correct SQL. In example 2, the average recall is also 80%, but with missing elements in every instance, not a single instance can generate correct SQL. This demonstrates that average recall does not accurately represent actual schema linking results.
>
> *Table W4.a  Two examples of using recall as a column linking metric for 5 instances—high average recall does not equal high correct linking*
>
> | Recall    | ${Recall} _{1}$ | ${Recall} _{2}$ | ${Recall} _{3}$ | ${Recall} _{4}$ | ${Recall} _{5}$ | Average recall | Correct instance |
> | --------- | --------------- | --------------- | --------------- | --------------- | --------------- | -------------- | ---------------- |
> | Example-1 | 100%            | 100%            | 66.7%           | 66.7%           | 66.7%           | 80%            | 2                |
> | Example-2 | 80%             | 80%             | 80%             | 80%             | 80%             | 80%            | 0                |
>
> - **Higher precision metric values may result in lower SQL generation accuracy:** Precision is not decisive in schema linking tasks. For instance, if the precision is 100% but the recall is below 100%, there will be 0 correct SQL generation. We provide two examples in the table below.
> - **As seen in the table**, in example 3, all 5 instances have no redundant elements in their linking results, yet all instances still have missing elements. Precision is 100%, but no correct SQL can be generated. In example 4, there are no missing elements, but redundancy exists; therefore, precision is low, but all instances can potentially generate correct SQL. This indicates that precision does not clearly correlate with the success of schema linking results.
>
> *Table W4.b  Two examples of using precision and recall as column linking metrics for 5 instances—precision cannot reflect linking results accurately*
>
> | (Precision, Recall) | (${Precision} _{1}$, ${Recall} _{1}$) | (${Precision} _{2}$, ${Recall} _{2}$) | (${Precision} _{3}$, ${Recall} _{3}$) | (${Precision} _{4}$, ${Recall} _{4}$) | (${Precision} _{5}$, ${Recall} _{5}$) | Average precision | Correct instance |
> | ------------------- | ------------------------------------- | ------------------------------------- | ------------------------------------- | ------------------------------------- | ------------------------------------- | ----------------- | ---------------- |
> | Example-3           | (100%, 80%)                           | (100%, 80%)                           | (100%, 80%)                           | (100%, 80%)                           | (100%, 80%)                           | 100%              | 0                |
> | Example-4           | (20%, 100%)                           | (20%, 100%)                           | (20%, 100%)                           | (20%, 100%)                           | (20%, 100%)                           | 20%               | 5                |

---

> > ### Comment · Reviewer_dUxV · 2024-11-27
> > **Thanks for the clarification about recall and precision**
> >
> > Your example helps clarify why standard recall and precision are not ideal.
> > It would be nice if you could demonstrate that if  standard recall and precision were used to guide a knapsack-like selection strategy, as close as possible to your method, they result in suboptimal choices.
> > Basically an ablation replacing your measures with recall and precision.
> > Does that make sense?

---

> ### Author Response · Authors · 2024-11-24
> **Response to W4 (2/2)**
>
> **(2) Necessary to propose missing ratio and redundancy ratio**
>
> Schema missing ratio: $R _{miss} =\frac{1}{|B|} \sum _{(q,S) \in B} \unicode{x1D7D9}(S _q \nsubseteq \widehat{S} _q)$
>
> Schema redundancy ratio: $R _{redun} = \frac{1}{|B|} \sum _{(q,S) \in B} \underbrace{{(\frac{|\hat{S} _q \setminus S _q|}{|\hat{S}_q|})}^{\sigma}} _{\text{redundancy ratio}}, \underbrace{\sigma = \unicode{x1D7D9}(S _q \subseteq \widehat{S} _q)} _{\text{Non-missing Indicator}}$
>
> Due to the above drawbacks, precision and recall cannot be used in schema linking evaluation. Then we propose the missing ratio and redundancy ratio to evaluate the schema linking accurately.
>
> - **Our schema missing ratio can reflect the actual impact of schema linking results to SQL generation by a strict evaluation.** It considers an instance a failure if any element is missing, acknowledging success only when no elements are missing. This aligns with the fact that any missing element leads to incorrect SQL generation.
> - **In Table W4.a,** the two examples show a missing ratio of 60% and 100% (lower is better), accurately reflecting the difficulty in generating correct SQL from these schema linking results.
> - **Lower missing ratio will have higher SQL generation accuracy**: The table below shows the column missing ratio and SQL generation accuracy using the same text-to-SQL model with different schema linking models. The results demonstrate that the missing ratio accurately reflects the accuracy of schema linking and directly affects SQL generation. KaSLA achieves the highest SQL generation accuracy with the lowest missing ratio.
>
> *Table W4.c Performance on Bird-dev with execution accuracy (EX) for text-to-SQL evaluation and missing ratio for schema linking evaluation. The SQL generation model is a fine-tuned StarCoder2-15B*
>
> |                         | Pure-SL | DTS-SL | TA-SL  | CodeS-SL | KaSLA      |
> | ----------------------- | ------- | ------ | ------ | -------- | ---------- |
> | Column missing ratio    | 57.54%  | 35.77% | 43.80% | 12.53%   | **5.83%**  |
> | Execution Accuracy (EX) | 50.26%  | 55.22% | 53.13% | 60.30%   | **63.75%** |
>
> - **Our schema redundancy ratio can provide a meaningful evaluation by calculating redundancy only when all matching elements are present in the linking results.** The main difference between precision and our redundancy ratio is that the prediction with element missing will be judged as failure because, in such cases, redundancy is not the sole reason for incorrect SQL generation, thus making its evaluation meaningless.
> - **In Table W4.b**, the two examples show redundancy ratios of 100% and 20% (lower is better), correctly reflecting that the first schema linking result struggles to generate correct SQL while the second one can.
> - **Lower redundancy and missing ratios will have higher SQL generation accuracy.** Using the same experimental setup, the table below shows the column redundancy ratio and corresponding SQL generation accuracy. The results in Tables W4.c and W4.d indicate that only KaSLA achieves the lowest redundancy and missing ratios, resulting in the highest accuracy of SQL generation.
>
> *Table W4.d Performance on Bird-dev with execution accuracy (EX) for text-to-SQL evaluation and missing ratio for schema linking evaluation. The SQL generation model is a fine-tuned StarCoder2-15B*
>
> |                         | Pure-SL | DTS-SL | TA-SL  | CodeS-SL | KaSLA      |
> | ----------------------- | ------- | ------ | ------ | -------- | ---------- |
> | Column redundancy ratio | 61.75 % | 83.04% | 46.25% | 80.83%   | **42.24%** |
> | Execution Accuracy (EX) | 50.26%  | 55.22% | 53.13% | 60.30%   | **63.75%** |

---

> ### Author Response · Authors · 2024-11-24
> **Responses to W5, Q1, Q2**
>
> > W5: The experiments do not include two of the SOTA schema linking works, E-SQL and CHESS, both are on the BIRD leadership. Especially, E-SQL focuses mainly on schema linking and does not use many of the other techniques, like self-consistency, to achieve high accuracy. Experiments should have included at least E-SQL. A. CHESS: Contextual Harnessing for Efficient SQL Synthesis: Shayan Talaei, Mohammadreza Pourreza, Yu-Chen Chang, Azalia Mirhoseini, Amin Saberi B. E-SQL: Direct Schema Linking via Question Enrichment in Text-to-SQL: Hasan Alp Caferoğlu, Özgür Ulusoy
>
> Thank you for providing these papers. We have included their citations and discussions in our revision. We reproduced E-SQL and CHESS based on the released codes and compared their performance with and without KaSLA.
>
> - **Integrating KaSLA can further enhance state-of-the-art (SOTA) text-to-SQL models like E-SQL and CHESS by improving their schema linking.** As shown in the table, integrating KaSLA provides a stable improvement for current SOTA models, highlighting its plug-and-play nature and robust schema linking capability.
> - **Implementation details: We enhanced their schema linking component with KaSLA.** CHESS [1] uses a retrieval-augmented chain-of-thought prompting method for schema linking, while E-SQL [2] employs a filtered schema correction strategy. Since CHESS uses GPT-4 and a fine-tuned DeepSeek Coder model, but did not release the latter’s checkpoint, we reproduced each component in CHESS using GPT-4. In line with [2], we used GPT-4o as the base LLM for E-SQL.
>
> *Table W5.a Text-to-SQL performance of integrating KaSLA into SOTA baselines (CHESS and E-SQL)*
>
> | Execution Accuracy (EX) | KaSLA  | CHESS (GPT-4) |  CHESS (GPT-4) +KaSLA       | E-SQL (GPT-4o) |   E-SQL (GPT-4o) +KaSLA       |
> | ----------------------- | ------ | ------------- | ---------- | -------------- | ---------- |
> | BIRD-dev                | 63.75% | 63.10%        | 63.89%     | 65.25%         | 65.91%     |
> | Spider-dev              | 88.01% | 87.14%        | 88.20%     | 88.30%         | 88.68%     |
>
> [1] CHESS: Contextual Harnessing for Efficient SQL Synthesis
>
> [2] E-SQL: Direct Schema Linking via Question Enrichment in Text-to-SQL
>
>
> > Q1: Redundancy is a confusing term; do you include the same table/column multiple times in the context? I understand that you do not. This is supposed to capture irrelevant schema elements; i.e. precision. How is this different?
> >
> > Q2: Similarly, how is missing schemas different than recall?
>
> We have responded to them together with our response to W3; please refer to the above response to W3.

---

> > ### Comment · Reviewer_WDRq · 2024-11-26
> >
> > Thanks for including the comparison. I appreciate the effort.
> > Could you provide more details on how E-SQL+KaSLA and CHESS+KaSLA implemented? Both of them have their own schema linking steps. Did you replace with yours? Execution accuracy of CHESS and CHESS+KaSLA and E-SQL and E-SQL + KaSLA are almost the same, suggesting their schema linking works as effectively as KaSLA.

---

> > > ### Author Response · Authors · 2024-11-29
> > >
> > > Thanks very much for your suggestions and concerns.
> > >
> > > > Could you provide more details on how E-SQL+KaSLA and CHESS+KaSLA implemented? Both of them have their own schema linking steps. Did you replace with yours? Execution accuracy of CHESS and CHESS+KaSLA and E-SQL and E-SQL + KaSLA are almost the same, suggesting their schema linking works as effectively as KaSLA.
> > >
> > > - **We added KaSLA as an additional prompt to implement CHESS + KaSLA and E-SQL + KASLA.** E-SQL proposed a question enrichment method to provide descriptions and used the enriched question as the prompt for SQL generation. CHESS proposed a retrieval method to provide cell values for the adaptive selection and use the question and filtered elements as prompts for SQL generation. We kept their respective prompts and added KaSLA as addition prompts.
> > > - **KaSLA outperforms their schema linking in both metrics and fair comparisons.** We compared the schema linking results of E-SQL on missing elements and redundancy, as well as their SQL generation performance with StarCoder2-15B. Table 1 shows that KaSLA has lower element missing and redundancy ratios. Table 2 demonstrates that KaSLA achieves better SQL generation performance with StarCoder-15B.
> > >
> > > *Table 1 Missing rate and Redundancy rate on Bird-dev of different schema linking methods*
> > >
> > > |                    | **Pure-SL** | **DTS-SL** | **TA-SL** | **CodeS-SL** | **E-SQL-SL** | **KaSLA**  |
> > > | ------------------ | ----------- | ---------- | --------- | ------------ | ------------ | ---------- |
> > > | Column  $R_{miss}$ | 57.54%      | 35.77%     | 43.80%    | 12.53%       | 77.03%       | 5.83%  |
> > > | Column $R_{redun}$ | 61.75%      | 83.04%     | 46.25%    | 80.83%       | 77.84%       | 42.24% |
> > > |                    |             |            |           |              |              |            |
> > >
> > > *Table 2 Performance on Bird-dev of fine-tuned StarCoder-15B with different schema linking methods*
> > >
> > > | **Execution Accuracy (EX)** | **Pure-SL** | **DTS-SL** | **TA-SL** | **CodeS** | **E-SQL-SL** | **KaSLA** |
> > > | --------------------------- | ----------- | ---------- | --------- | --------- | ------------ | --------- |
> > > | BIRD-dev                    | 50.26%      | 55.22%     | 53.13%    | 60.30%    | 56.26%       | 63.75%    |
> > >
> > > - **Though simply adding prompts, KaSLA still improves these two models.**  E-SQL and CHESS utilize tailored techniques to enhance the best-performed LLMs (GPT-4 and GPT-4o) to generate SQL. However, simply adding KaSLA as addition prompts can also improve them, verifying the generalisability and effectiveness of KASLA.
> > >
> > > Thanks again for engaging in this discussion to address any remaining concerns. We appreciate your time and further discussions.

---

> ### Author Response · Authors · 2024-11-24
> **Responses to Q3, Q4**
>
> > Q3.1: In the paper, when you say schema it is confusing whether you mean table or column or both. It is not until 4.3 that it gets some clarification.
>
> - **Since the definitions of knapsack schema linking, metrics, and score and factor estimation are the same for both tables and columns,** we did not distinguish them before Section 4.3 to avoid repetition. Such unified definitions aid in the development of KaSLA. Thanks to your suggestion, we have revised the related text for clarity and updated it in the revision.
> - **We distinguish between tables and columns in Section 4.3, the hierarchical KaSLA strategy**, because we treat them differently in a hierarchical process. We first perform table linking, followed by linking columns within the selected tables.
>
> > Q3.2 Do you mix schema errors or solve two separate knapsack problems, one for tables and one for columns?
>
> - **KaSLA addresses two separate knapsack optimization problems: one for tables and another for columns within the linked tables.** After estimating importance scores, KaSLA employs a hierarchical process for schema linking. It starts with table linking using proposed knapsack optimization method, then links columns within the linked tables, guided by the same method.
> - **This approach is optimal because it reduces the selection space of columns, lowers time costs, and avoids noise and confusion from the columns of unlinked tables.** The pseudocode for this process is provided in Appendix C.
>
> > Q3.3 The models in the score function, are they trained for tables and columns separately?
>
> - **The models in the score function are trained simultaneously for both tables and columns, rather than separately.** During the training and inference of the scoring models, addressing both table and column scores simultaneously helps the model integrate semantic understanding and establish relationships across different tables through the attention networks in the language model.
> - **This simultaneous approach enhances KaSLA's ability to address the challenging multi-table JOIN tasks.** The table below shows results from instances involving or not involving the JOIN operation in BIRD-dev. By simultaneously addressing tables and columns, KaSLA outperforms baselines in multi-table JOIN tasks.
>
> *Table Q3.3.a Text-to-SQL performance of KaSLA on the BIRD-dev data involving and not involving the JOIN operation*
>
> | Execution Accuracy (EX) in BIRD-dev           | SuperSQL | CodeS  | KaSLA      |
> | --------------------------------------------- | -------- | ------ | ---------- |
> | BIRD-dev involving JOIN (1140 instances)      | 56.67%   | 56.75% | **62.89%** |
> | BIRD-dev not involving  JOIN  (394 instances) | 64.21%   | 63.45% | **66.24%** |
>
> > Q3.4 Column selection is conditional, you choose columns from the selected tables; how does that impact the score functions?
>
> - **Choosing columns from selected tables is more efficient and accurate than choosing from all tables because it reduces the selection space, lowers time costs, and avoids noise and confusion from unlinked tables' columns.** As shown in the table below, the performance of choosing columns from selected tables is better than choosing from all tables in both datasets.
>
> *Table Q3.4.a Text-to-SQL performance of KaSLA with choosing columns from selected tables and all tables*
>
> | Execution Accuracy (EX) | Choose column from all tables | Choose columns from the selected tables (KaSLA) |
> | ----------------------- | ----------------------------- | ----------------------------------------------- |
> | BIRD-dev                | 62.58%                        | 63.75%                                          |
> | Spider-dev              | 87.23%                        | 88.01%                                          |
>
>
> > Q4: How are the scoring functions trained? Ground truth for which dataset? Dev or train?
>
> - **The two scoring functions are trained on the training dataset using its ground truth labels.** We only use the dev dataset for evaluations.

---

> ### Author Response · Authors · 2024-11-24
>
> Thanks for your valuable suggestions and concerns. We have incorporated all the additional discussions in our revision. We also welcome further discussion on any remaining concerns.

---

> ### Author Response · Authors · 2024-11-28
>
> Thanks so much for your response and for discussing your concerns with us further. We truly appreciate the opportunity to continue this discussion and are eager to address your questions and concerns. We will proceed step by step to provide detailed answers to each concern you've raised.
>
>
>
> > BTW, I am still very confused with redundancy. Do you mean the same schema element is included multiple times in the context, hence redundant? I interpreted redundancy as irrelevancy.
>
> Thanks for your deep thinking. Redundancy does not refer to the same schema element being included multiple times. For irrelevancy, the following are the detailed clarifications.
>
> ### **Concepts clarification**
>
> While interpreting redundancy as irrelevancy is a good direction, our use of the term "redundancy" differs from irrelevancy in a subtle but important way.
>
> - Redundancy has a mandatory requirement of the inclusion of all the needed schema elements:
>   - Do not include all necessary schema elements—-Redundancy Ratio is 1.
>   - Includes the necessary schema elements—-Redundancy Ratio is the irrelevancy ratio.
> - **Low irrelevancy cannot ensure generating SQL correctly:** A low irrelevancy ratio cannot guarantee that it isn’t missing necessary elements. Thus, even with a low irrelevancy ratio, the text-to-SQL model may also generate incorrect SQL.
>
> ### **Toy example**
>
> We illustrate the difference between redundancy and irrelevancy with the following toy example. Assuming **a**, **b**, and **c** are the correct schema linking elements. Observing from this table:
>
> - Low redundancy ratios will lead to correct SQLs.
> - High redundancy ratios will lead to incorrect SQLs.
>
> The conclusion is that irrelevancy ratio does not have stable correlation with SQL accuracy.
>
> |      | Linked results              | Correct SQL? | Redundancy Ratio **(Less is better)** | Irrelevancy **(Less is better)** |
> | ---- | --------------------------- | ------------ | ---------------- | -------------------------------- |
> | 1    | ***a, b, c,** d*            | Yes          | 25%              | 25%                              |
> | 2    | ***a, b, c**, d, e, f, … z* | Maybe not    | 88.46%           | 88.46%                           |
> | 3    | ***a, b***                  | No           | 100%             | 0%                               |
> | 4    | ***a,** d, e*               | No           | 100%             | 67%                              |
>
> ### **Redundancy formulation clarification**
>
> We implement the redundancy ratio with a non-missing indicator to determine if it contains all necessary elements.
>
> ($B$ is the whole dev dataset, $\hat{S} _q$ is the predicted schema linking results, and $S _q$ is the ground truth.)
>
> - **Redundancy ratio:**
>
>   $$ R _{redun} =\frac{1}{|B|} \sum _{(q,S) \in B}  {(\underbrace{\frac{|\hat{S} _q \setminus S _q|}{|\hat{S} _q|}} _{\text{irrelevancy}})}^{\sigma}$$
>   where $\sigma$ is the Non-missing Indicator, $\sigma = 1$ if $\widehat{S} _q \text{ contains all elements in } S _q$, and $\sigma = 0$ if  $\widehat{S} _q \text{ doesn't contain all elements in } S _q$.
>
> - **Irrelevancy ratio:**
>
>   $$R_{irrelevancy}=\frac{1}{|B|} \sum _{(q,S) \in B}\frac{|\hat{S} _q \setminus S _q|}{|\hat{S} _q|}$$
>
> The redundancy ratio will be 100% if the schema linking does not contain all necessary schema elements, which is meaningless for generating SQL correctly.

---

> ### Author Response · Authors · 2024-11-28
>
> > Thanks for the detailed response. I understand that there is a trade-off between recall (making sure all needed schema elements are selected) and precision (there is no irrelevant schema element). But there is no natural capacity requirement.
>
> Thanks for your careful reading. Schema linking capacity is the core trade-off “hyper-” parameter that impacts missing and redundancy (irrelevancy). We utilize a toy example to illustrate the capacity.
>
> Assuming that **a, b, and c** are the accurate schema-linking elements, then we have the following cases:
>
> - Over-large capacities will cause redundancy: If the capacity is set as 26, then the schema linking set will contain 23 irrelevant elements, resulting in a serious redundancy.
> - Over-small capacity will cause missing: If the capacity is set as 2, then the schema linking set will miss at least one necessary element.
> - Suitable capacities can better trade off the missing and redundancy to achieve a higher accuracy in SQL generation.
>
> *Table 2. Examples of the impacts of different capacities for SQL generation*
>
> |            | **Linked results**        | **Capacity** | Missing Ratio | **Redundancy Ratio**       |
> | ---------- | ------------------------- | ------------ | ------------- | -------------------------- |
> | Instance-1 | **a, b, c,** *d, e, …, z* | 26           | 0/3           | 23/26 = 88%                |
> | Instance-2 | **a, b**                  | 2            | 1/3           | 100% (judged as a failure) |
> | Instance-3 | ***a, b, c***             | 3            | 0/3           | 0/3 = 0%                   |

---

> ### Author Response · Authors · 2024-12-01
> **Looking forward to your constructive feedback**
>
> Dear reviewer WDRq,
>
> Thanks again for taking the time to review our paper and providing valuable feedback on our responses. We have carefully responded to your latest concerns, providing detailed explanations to clarify the definition of redundancy and demonstrating the importance of capacity in schema linking. We have also showcased KaSLA's significant role in reducing content length and provided implementation details on its improvements over the two latest baselines. We hope these responses can address your concerns and thanks for your support.
>
> As the author-reviewer discussion deadline approaches, we would be very grateful if you could share your valuable thoughts with us. If you have any further concerns, we look forward to further discussion with you.

---

### Author Response · Authors · 2024-12-04
**Thanks for taking the time to review our paper and providing valuable feedback**

Dear AC and Reviewers,

We sincerely appreciate your valuable time and insightful suggestions. Our paper offers new potential in text-to-SQL generation, which can assistt and facilitate real-world applications, particularly for complicated schemas and enhancing existing models (both local LLMs and closed-source LLMs). Below, we highlight the key contributions of our KaSLA model:

1. **Providing Suitable Metrics and Benchmarks for Schema Linking:** As discussed with Reviewers WDRq and dUxV, traditional metrics such as AUC, F1, Precision, and Recall are inadequate for evaluating the performance of schema linking models. To address this, KaSLA introduces tailored **missing ratio** and **redundancy ratio** metrics, where **lower missing and redundancy ratios** directly correlate with higher SQL generation performance.

2. **Introducing a New Paradigm for Schema Linking and Text-to-SQL:** KaSLA discovers that optimal schema linking results should simultaneously satisfy low missing ratios and low redundancy ratios. By **formulating schema linking as a knapsack problem**, KaSLA provides accurate schema-linking results that retain all necessary schema elements while avoiding redundancy.

3. **Providing Significant Improvements for Popular Text-to-SQL Models:** As discussed with Reviewers WDRq, BQ9c and dUxV, the schema linking results produced by KaSLA can generally **enhance the performance of existing text-to-SQL models**, including the mentioned CHESS and E-SQL. Moreover, KaSLA demonstrates remarkable **transferability across different datasets**, as discussed with Reviewers WDRq and jitH.

During the rebuttal period, we engaged in productive discussions with the reviewers and are pleased to report that Reviewers WDRq, BQ9c, dUxV and QByZ have accepted and approved the insights of KaSLA, as well as our additional experiments and responses. The key points of our discussion include:

1. **Verifying the Transferability of KaSLA:** We conducted cross-scenario transfer experiments to demonstrate that KaSLA exhibits strong transferability by adapting effectively to different scenarios without requiring further fine-tuning.

2. **Confirming Improvements over SOTA Baselines:** We compared KaSLA with the latest baselines, and the results consistently show that KaSLA can improve their performance without any tailored techniques.

3. **Comparing Different Design Versions:** We conducted ablation studies on the key components and hyperparameters in KaSLA. The results demonstrate that each component in KaSLA is well-organized and essential, and the overall framework exhibits good generalizability and robustness.

4. **Enhancing Figure Illustrations:** We have improved the main figures to clearly illustrate the crucial differences between KaSLA and traditional schema linking methods. We have also reformatted the equations and provided more details on KaSLA for clearer presentation.

5. **Analyzing Inference Time Cost:** We detailed the inference time cost of each component in KaSLA and discussed potential time delays. Additionally, we provided an analysis of the low time complexity of the dynamic programming component in KaSLA, which benefits from its hierarchical approach.

6. **Discussing Related Literature:** We provided a detailed discussion of related works in schema linking to clarify current challenges and highlight the impact of our work.

We thank all the reviewers and AC for their valuable feedback and insightful discussions.  We believe that improvements and clarifications made during the rebuttal period, along with the support from the reviewers, demonstrate the significance and novelty of our KaSLA model.

Thank you once again for your consideration.

Sincerely,

The Authors

---

### Meta-Review · Area_Chair_nEd8 · 2024-12-23

**Metareview:**

This paper proposes modeling schema linking in text-to-SQL generation into a Knapsack problem. The proposed formulation can effectively model the tradeoff between precision and recall during schema linking. Experiments on two datasets demonstrate the effectiveness of the proposed approach.

**Strengths**:

* This paper highlights the importance of schema linking in text-to-SQL, and provides a suitable benchmark for this domain.

* The proposed method significantly outperformed popular existing text-to-SQL methods.

**Weaknesses**:

* Many reviewers find that the paper is hard to understand (WDRq, jitH, dUxV), with missing details and overall a lack of clarity. In particular, there are a lot of new technical terms and concepts introduced in the paper, which are often not quite intuitive. As raised by dUxV, the authors should provide more explanations, motivating examples, along with ablations to motivate the design of the certain component of the system, such as using custom metrics instead of relying on precision/recall.

* A major concern is the motivation of formulating schema linking as a knapsack problem (WDRq, dUxV). Specifically, it remains unclear whether some elements in the knapsack problem formulation, such as capacity constraints, are sound and appropriate. This is also partially due to issues with the writing as some design choices lack proper motivation and explanation.

While I really appreciate the authors for spending tremendous amounts of time and effort to revise the submission while providing additional clarifications and extensive additional experimental results, the amount of revision required to make this paper much better would go beyond the scope of the author response period. Therefore, I believe the submission would benefit from another round of major revision in order to fully address these issues. Thanks for submitting your work.

**Additional Comments On Reviewer Discussion:**

There are a lot of clarifications with additional experiments offered by the authors during the rebuttal period. This also reflects the amount of revision required in order to make this paper reach the level of acceptance. The remaining issue around whether modeling schema linking as knapsack problem is a natural choice left unaddressed.

---

### Decision · Program_Chairs · 2025-01-22

Reject